# Plasma proteome variation and its genetic determinants in children and adolescents

Lili Niu [1,2,3], Sara Elizabeth Stinson [4], Louise Aas Holm [4,5], Morten Asp Vonsild Lund [5,6], Cilius Esmann Fonvig [4,5,7], Leonardo Cobuccio[1], Jonas Meisner[1], Helene Bæk Juel[4], Joao Fadista[3], Maja Thiele[8,9], Aleksander Krag[8,9], Jens-Christian Holm [4,5,7], Simon Rasmussen [1,10] ✉, Torben Hansen [4] ✉ & Matthias Mann [1,2] ✉

Our current understanding of the determinants of plasma proteome variation during pediatric development remains incomplete. Here, we show that genetic variants, age, sex and body mass index significantly influence this variation. Using a streamlined and highly quantitative mass spectrometry-based proteomics workflow, we analyzed plasma from 2,147 children and adolescents, identifying 1,216 proteins after quality control. Notably, the levels of 70% of these were associated with at least one of the aforementioned factors, with protein levels also being predictive. Quantitative trait loci (QTLs) regulated at least one-third of the proteins; between a few percent and up to 30-fold. Together with excellent replication in an additional 1,000 children and 558 adults, this reveals substantial genetic effects on plasma protein levels, persisting from childhood into adulthood. Through Mendelian randomization and colocalization analyses, we identified 41 causal genes for 33 cardiometabolic traits, emphasizing the value of protein QTLs in drug target identification and disease understanding.

The global prevalence of pediatric obesity has increased markedly over the past four decades, with affected children and adolescents facing elevated risks of prediabetes, metabolic syndrome, asthma and fatty liver disease[1]. Studying childhood obesity is vital for understanding its health consequences and formulating effective prevention and treatment strategies[2].

The concentration of some proteins in the blood varies with growth, especially during puberty, as a result of hormonal and metabolic changes, immune maturation and tissue development. Examples include insulin-like growth factor 1 (IGF1), leptin, growth hormone, sex hormones, insulin and C-reactive protein[3].

The levels of specific proteins in human blood are the most commonly used indicators of potential health issues[4]. Understanding the genetic and other determinants of the plasma proteome can support biomarker research and drug development[5–8]. Factors such as genetics, age, sex, body mass index (BMI), growth and development including puberty affect circulating protein levels[9–12], and this is important to investigate at a large scale in children and adolescents, covering pre-pubertal and post-pubertal stages.

Affinity-based proteomics can infer the relationship between blood protein levels and these factors at a large scale[8,13–16]. In comparison, mass spectrometry (MS)-based proteomics provides much higher

[1]Novo Nordisk Foundation Center for Protein Research, University of Copenhagen, Copenhagen, Denmark. [2]Department of Proteomics and Signal Transduction, Max Planck Institute of Biochemistry, Martinsried, Germany. [3]Novo Nordisk A/S, Copenhagen, Denmark. [4]Novo Nordisk Foundation Center for Basic Metabolic Research, University of Copenhagen, Copenhagen, Denmark. [5]The Children's Obesity Clinic, accredited European Centre for Obesity Management, Department of Pediatrics, Copenhagen University Hospital Holbæk, Holbæk, Denmark. [6]Department of Biomedical Sciences, University of Copenhagen, Copenhagen, Denmark. [7]The Faculty of Health and Medical Sciences, University of Copenhagen, Copenhagen, Denmark. [8]Odense Liver Research Centre, Department of Gastroenterology and Hepatology, Odense University Hospital, Odense, Denmark. [9]Department of Clinical Research, University of Southern Denmark, Odense, Denmark. [10]The Novo Nordisk Foundation Center for Genomic Mechanisms of Disease, Broad Institute of MIT and Harvard, Cambridge, MA, USA. ✉e-mail: srasmuss@sund.ku.dk; torben.hansen@sund.ku.dk; mmann@biochem.mpg.de

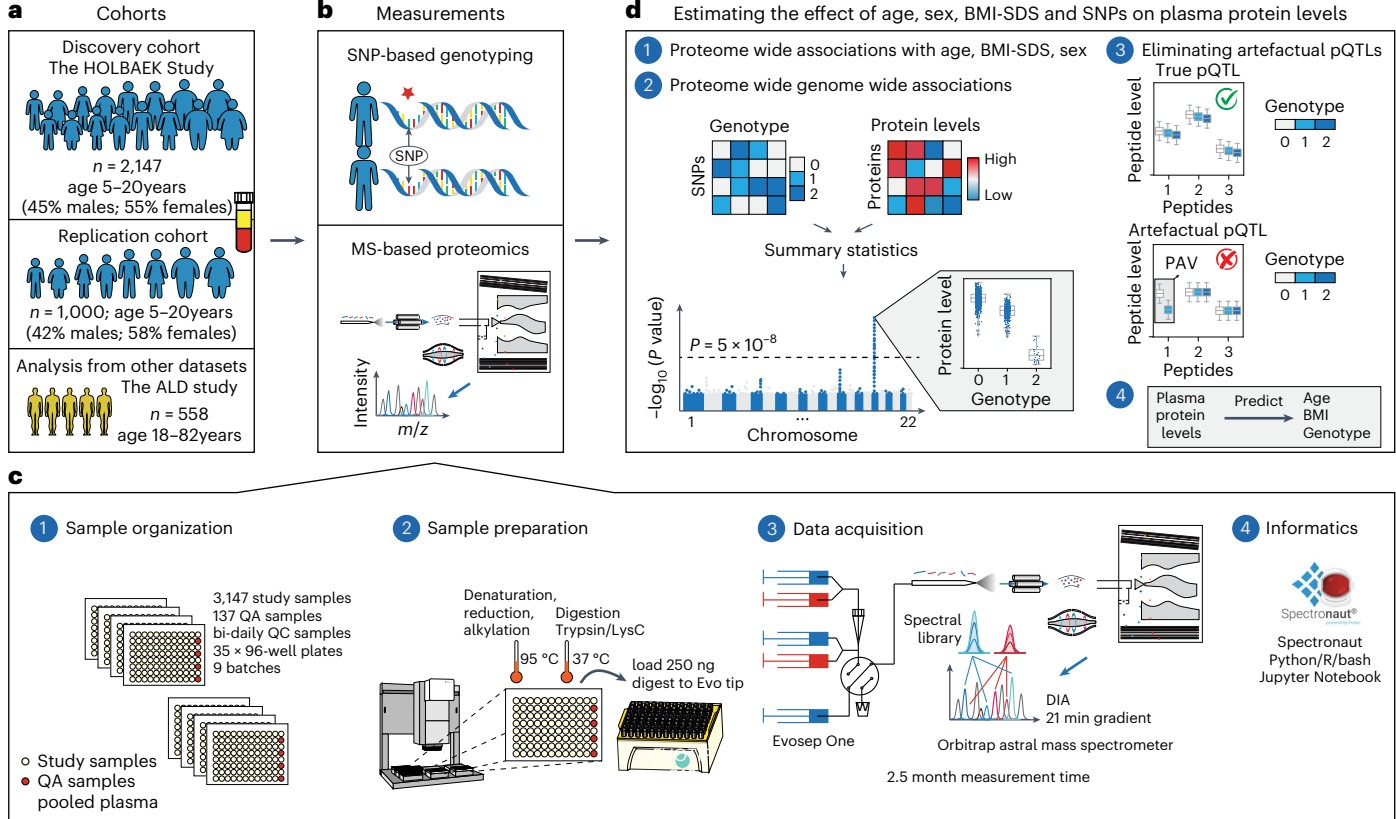

**Fig. 1 | Study overview and proteomics workflow. a**, Discovery and replication cohorts used in this study. **b**, MS-based plasma proteome profiling and SNP-based genotyping were performed on the discovery and replication cohorts. **c**, Proteome profiling workflow and the computational tools used for processing of proteomics data, including (1) sample organization, (2) sample preparation, (3) data acquisition and (4) informatics. **d**, Schematic representation of (1 and 2) the association analysis with (3) a quality control step to eliminate artefactual pQTLs as described in the main text and (4) prediction of age, BMI and genotype based on plasma protein levels. ALD, alcohol-related liver disease; $m/z$, mass to charge ratio; QA, quality assessment; QC, quality control.

specificity in identifying and quantifying proteins[17–19], as MS measures the mass of peptides and their fragments down to several parts per million. However, MS-based proteomics has been limited by lower throughput and fewer quantified proteins because of constraints of MS sensitivity and other factors[20–23]. In bottom-up MS-based proteomics, peptide signals are measured, quantified and assembled into proteins using a reference proteome database. Depending on the number of peptides mapping to each protein, this generally yields several data points on the identity and quantitative changes of the proteins in the plasma proteomes of the study participants. These can be used for better protein-level estimates or the removal of outliers. By contrast, binder-based methods typically probe a single epitope of plasma proteins or two closely spaced ones, as in the Olink proximity extension assay. Here, we aimed to alleviate the limitations of MS-based proteomics, enabling us to investigate larger cohorts with greater proteome depth than previously possible, including replication cohorts.

## Results

### Discovery and replication cohorts

Our discovery cohort included 2,147 children and adolescents aged 5–20 years from the HOLBAEK Study, 45% of whom were from the general population and 55% from the Children's Obesity Clinic in Holbæk, Denmark (Fig. 1a). For replication of the genetic effects on plasma protein levels, we included 1,000 additional children and adolescents from the HOLBAEK Study and 558 adults with alcohol-related liver disease aged 19–82 years, matched with healthy controls recruited from the Region of Southern Denmark[5]. Baseline characteristics for all cohorts are provided in Tables 1 and 2, with the discovery cohort further stratified

by sex in Supplementary Note 1. We conducted single-nucleotide polymorphism (SNP)-based genotyping and MS-based plasma proteome profiling (Fig. 1b and Methods). We acquired plasma proteome profiles of all participants ($n$ = 2,147) with a data-independent acquisition (DIA) strategy and a single-run workflow, using a liquid chromatography system designed for robust clinical use[24] coupled online to a recently released Orbitrap Astral mass spectrometer[25] (Fig. 1c). We performed sample preparation and liquid chromatography–MS analysis using equal plasma volumes, with precursor-level MS signal normalization across samples using Spectronaut's cross-run normalization method (Methods and Supplementary Note 2)[26]. After stringent quality control, 1,216 proteins remained with 91% data completeness. Analyzing 94 quality assessment samples over a 6 week measurement period revealed a 33% median coefficient of variation for the entire workflow (Supplementary Fig. 1a–d). We estimated the effect of age, sex, BMI-standard deviation score (BMI-SDS) adjusted for age and sex[27] and 5.2 million SNPs on plasma protein levels (Fig. 1d).

### Impact of demographic and health factors on plasma proteome

To the best of our knowledge, this is the first MS-based proteomics study on neat plasma with substantial depth (>1,000 proteins), largely owing to the use of the Orbitrap Astral mass spectrometer[25] for this purpose. We investigated the biological processes represented by the quantified proteins. The most prevalent processes included complement and coagulation cascades, metabolism and inflammatory response, reflecting the key roles of plasma proteins in immunity, blood clotting and transport (Fig. 2a).

## Table 1 | Participant characteristics in the discovery cohort

|  | General population (n=951) | Obesity clinic (n=1,179) |
|---|---|---|
| Age, mean (s.d.), years | 12 (3) | 12 (3) |
| Female sex, no. (%) | 530 (56) | 641 (54) |
| Male sex, no. (%) | 421 (44) | 538 (46) |
| BMI, mean (s.d.), kg m$^{-2}$ | 17.7 (2.3) | 27.1 (5.6) |
| BMI-SDS, mean (s.d.) | −0.05 (0.81) | 2.77 (0.75) |
| Tanner stage 1/2–5, no. (%) | 233/474 (33/67) | 327/569 (36/64) |
| ALT, U l$^{-1}$ | 19 (16–23) | 24 (19–31) |
| AST, U l$^{-1}$ | 26 (22–31) | 24 (20–29) |
| GGT, U l$^{-1}$ | 16 (12–19) | 17 (15–21) |
| Glucose, mmol l$^{-1}$ | 5 (4.7–5.2) | 5 (4.8–5.3) |
| Insulin, pmol l$^{-1}$ | 51 (36.3–68.4) | 79 (55.1–117.8) |
| HbA1c, mmol mol$^{-1}$ | 34 (32–35) | 34 (32–36) |
| Triglycerides, mmol l$^{-1}$ | 0.6 (0.5–0.8) | 0.9 (0.7–1.3) |
| Total cholesterol, mmol l$^{-1}$ | 3.9 (3.5–4.3) | 4 (3.6–4.6) |
| LDL cholesterol, mmol l$^{-1}$ | 2 (1.7–2.4) | 2.3 (1.9–2.8) |
| HDL cholesterol, mmol l$^{-1}$ | 1.5 (1.3–1.8) | 1.2 (1–1.4) |

Abbreviations: ALT, alanine aminotransferase; AST, aspartate aminotransferase; GGT, gamma-glutamyl transferase; HbA1c, hemoglobin A1c; LDL, low-density lipoprotein; HDL, high-density lipoprotein. Data are shown as median (interquartile range) unless otherwise noted.

## Table 2 | Participant characteristics in the replication cohorts

| Sub-group | Children replication cohort | | Adult replication cohort |
|---|---|---|---|
|  | General population (n=407) | Obesity clinic (n=590) | GALA–ALD/HP (n=558) |
| Age, mean (s.d.), years | 12 (3) | 12 (3) | 56 (10) |
| Female sex, no. (%) | 259 (64) | 320 (54) | 148 (73) |
| Male sex, no. (%) | 148 (36) | 270 (46) | 410 (27) |
| BMI, mean (s.d.), kg m$^{-2}$ | 17.7 (2.3) | 26.9 (5.4) | 27.2 (5) |
| BMI-SDS, mean (s.d.) | −0.08 (0.8) | 2.76 (0.75) | NA |
| Tanner stage 1/2–5, no. (%) | 83/214 (28/72) | 149/259 (37/63) | NA |
| ALT, U l$^{-1}$ | 20 (17–23) | 22 (18–30) | 28 (21–42) |
| AST, U l$^{-1}$ | 25 (21–32) | 26 (21.7–31) | 30 (24–46) |
| GGT, U l$^{-1}$ | 16 (13–19) | 16 (13–20) | 48 (25–136) |
| Glucose, mmol l$^{-1}$ | 5 (4.7–5.2) | 5.1 (4.8–5.4) | 6 (5.5–6.6) |
| Insulin, pmol l$^{-1}$ | 52.9 (35.4–69.3) | 78.5 (53.8–124.1) | NA |
| HbA1c, mmol mol$^{-1}$ | 34 (31–35) | 34 (32–36) | 36 (33–39) |
| Triglycerides, mmol l$^{-1}$ | 0.6 (0.5–0.8) | 0.9 (0.6–1.4) | 1.2 (0.9–1.8) |
| Total cholesterol, mmol l$^{-1}$ | 3.9 (3.4–4.3) | 4.2 (3.7–4.7) | 5 (4.4–5.9) |
| LDL cholesterol, mmol l$^{-1}$ | 2 (1.7–2.4) | 2.4 (2–2.9) | 3 (2.3–3.6) |
| HDL cholesterol, mmol l$^{-1}$ | 1.5 (1.3–1.7) | 1.2 (1–1.4) | 1.3 (1.1–1.7) |
| Abstaining from alcohol at time of inclusion, no. (%) | NA | NA | 197 (35) |
| Statin use prior time of inclusion, no. (%) | NA | NA | 97 (17) |
| Steatosis 0/1/2/3, no. | NA | NA | 373/79/70/36 |
| Inflammatory activity 0/1/2/3/4/5, no. | NA | NA | 290/90/78/ 50/28/22 |
| Fibrosis stage 0–1/2/3/4, no. | NA | NA | 367/102/26/63 |

GALA–ALD/HP, gut and liver axis–alcohol-related liver disease/healthy participants. Data are shown as median (interquartile range) unless otherwise noted.

To assess the impact of age, sex and BMI-SDS[27] on the plasma proteome, we performed multiple linear regression analysis (Fig. 2b and Methods). In total, 58% of the quantified plasma proteins were associated with at least one of these factors (40% with age, 32% with sex and 22% with BMI-SDS), with 8% linked to all three based on the current dataset and our modeling (Fig. 2c). Proteins most strongly associated with age included known age-related proteins such as F9 (ref. 28), RBP4 (ref. 29) and COL1A1 (ref. 9) as well as others not previously linked to age, such as GPLD1, APCS and IGFALS (Fig. 2d and Supplementary Table 1). The age association of IGFALS aligns with evidence that defects or low expression of IGFALS can cause pubertal delay in children[30]. In addition to individual proteins, our dataset reveals insights into three critical biological processes that occur during pediatric development. Firstly, we observed an age-related increase in IGF1 receptor signaling, with IGF1 levels peaking at ages 12–13 years in female adolescents and 14–15 years in male adolescents and then declining (Extended Data Fig. 1a,b), consistent with the literature[31,32]. IGFBP5 mirrored IGF1, although the levels of IGFBP1 and IGFBP2 declined from childhood through adolescence (Extended Data Fig. 1c–e). IGFBP1 and IGFBP2 were consistently reduced in children with obesity, along with adiponectin (Extended Data Fig. 1d–f). Similarly, SHBG, A2M and CRP all showed obesity-dependent levels (Extended Data Fig. 1g–i). Secondly, we captured post-pubertal declines in essential bone development proteins (ACAN, COL1A1, COL1A2, THBS4, COMP and POSTN), reflecting sex-specific differences in growth plate closure during skeletal maturation (Extended Data Fig. 1j–m). Aggrecan (ACAN) is a major proteoglycan in cartilage, and mutation in this protein causes early growth cessation, resulting in a severely reduced adult stature[31]. Our data documents a more than tenfold decline in ACAN levels starting at age 12–13 years for female adolescents and 13–14 years for male adolescents, offering the potential for early growth disorder diagnostics. Thirdly, proteins involved in angiogenesis and cellular adhesion (ANGPTL3, CDH5, ITGB1, ICAM1, VCAM1 and ACE) showed an age-related decline (Extended Data Fig. 1n–q). This analysis identified proteins not previously associated with childhood obesity, to the best of our knowledge, including

A2M, PON3, ADAMTSL4, HSPG2 and MAGEB6B, all of which showed decreased levels in children with obesity.

Similarly, we recapitulated known sex differences in protein levels, such as PZP and BCHE, and identified proteins without previously reported sex-specific differences, including CD5L (Fig. 2e and Supplementary Table 1). Notably, CD5L and IGHM emerged as the top proteins associated with sex, aligning with the observation that females have higher IgM levels than males[33]. This finding is further supported by a recent report showing that CD5L is an obligate member of circulating IgM[34].

We included an interaction term between BMI-SDS and obesity status to assess whether protein association with BMI-SDS differed between subgroups. Of the 240 BMI-SDS-associated proteins, 163 were specific to the obesity group and 32 showed differing effect sizes between groups (Fig. 2c). This suggests that a general population of similar size would have yielded fewer BMI-SDS-associated proteins. Inflammatory proteins showed the strongest associations

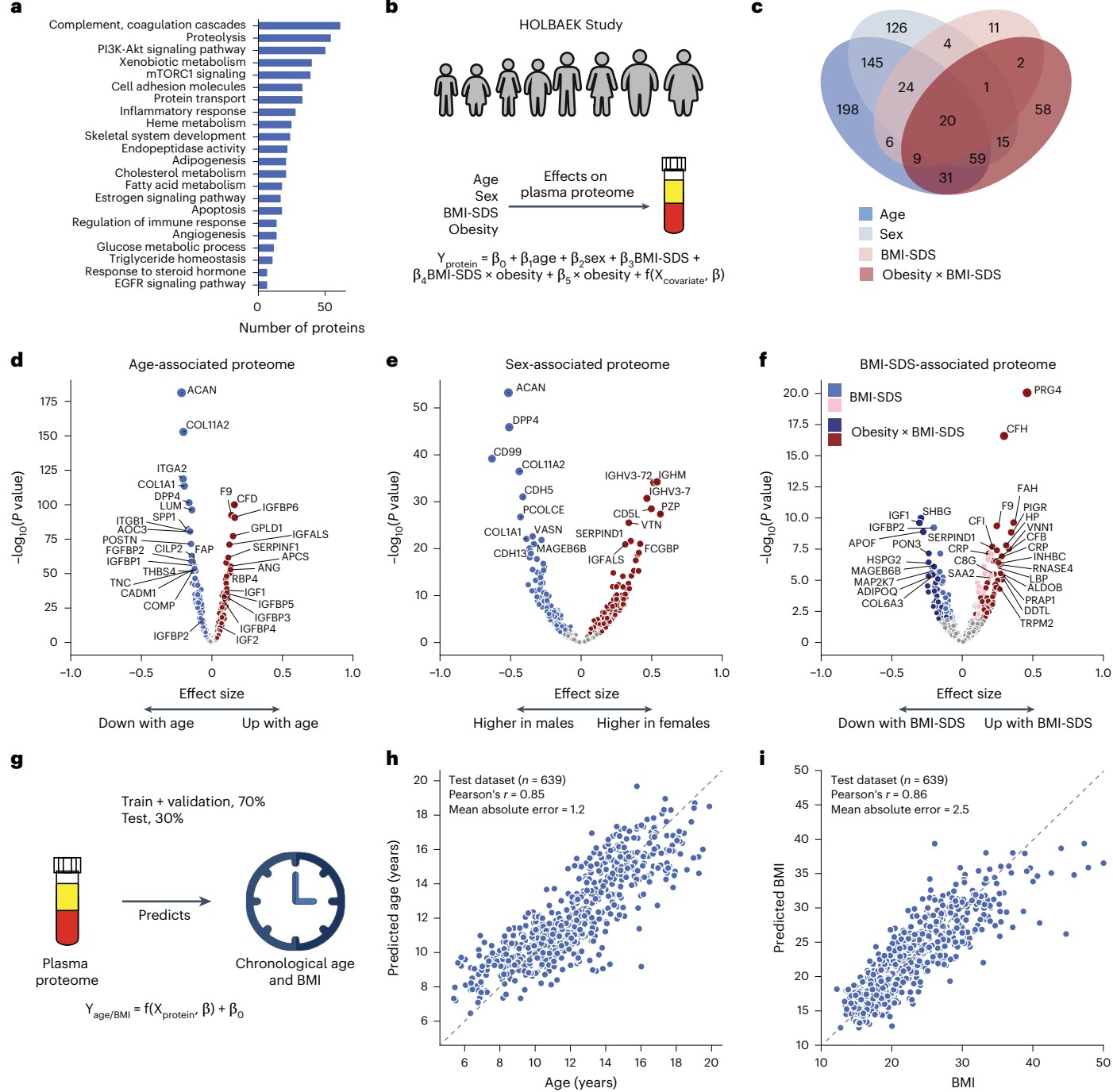

**Fig. 2 | Age-associated, sex-associated and BMI-SDS-associated plasma proteins. a**, Biological processes represented by all identified proteins after quality control. **b**, Schematic representation of linear modeling of protein levels using various factors. **c**, Number of proteins associated with age, sex, BMI-SDS and the interaction term between obesity status and BMI-SDS; $n = 1,601$ biologically independent samples. **d**–**f**, Volcano plots showing proteins associated with age (**d**), sex (**e**) and BMI-SDS (**f**), highlighting strongly associated proteins. For **c**–**f**, multiple linear regression was used to test for association, with beta coefficients estimated using ordinary least squares regression. Two-sided $P$ values were approximated using a $t$-distribution with significance set at Benjamini–Hochberg-corrected $P < 0.05$. **g**, Schematic representation of linear modeling of age and BMI using plasma proteome. **h**,**i**, Prediction of age (**h**) and BMI (**i**) in the test set. Pearson's correlation coefficients between predicted and real values are indicated; $n = 639$ biologically independent samples.

with BMI-SDS, including CRP, complement system proteins (C3, CFH, CFI) and acute phase proteins (A2M, APCS, SAA1, LBP)[35] (Fig. 2f). Most of these were also statistically significant in the normal weight group, albeit with smaller effect sizes, indicating that elevated inflammatory protein levels with increasing BMI-SDS are not exclusive to obesity (Supplementary Table 1). Among the BMI-SDS-associated proteins is ANGPTL3, which has shown variable associations with BMI and obesity

in previous studies (Supplementary Note 3). Notably, we observed that PRG4 decreased with weight loss[35], consistent with its positive association with BMI-SDS in this dataset. Interestingly, PRG4 deficiency protects mice against glucose intolerance and fatty liver disease, suggesting the therapeutic potential of the proteins identified here[36].

Further exploring age, sex and BMI-SDS interactions revealed 24 proteins with interaction effects between age and BMI-SDS, 18 between

sex and BMI-SDS and 149 between age and sex, including PZP, AGT, SHBG and the above-mentioned bone development proteins (Supplementary Table 2).

Plasma protein levels can serve as a 'biological clock' in adults[9]. We extended this concept to children and adolescents, accurately estimating age within ±1.2 years using the 50 most predictive proteins (Pearson's $r = 0.85$ between predicted and actual age) in a subset of 639 individuals not included in model training (Fig. 2g,h, Supplementary Table 3 and Methods). Likewise, a panel of 50 proteins consistently indicated BMI (Fig. 2i). Age-predictive proteins primarily regulated IGF1 receptor signaling, cartilage and skeletal development, fibroblast growth factor response and cell–cell adhesion. Notably, the top five to ten proteins alone predicted age nearly as well (mean absolute error, 1.5–1.6 vs 1.2 years for all 50). BMI-predictive proteins included established obesity markers and obesity-related proteins, including adiponectin, CRP, IGFBP1, IGFBP2, PRG4, SHBG, apolipoproteins (APOA4, APOF) and inflammatory response proteins (A2M, APCS, LBP, HSPG2, HP, AOC3, ITGB1, VNN1).

### Effect of SNPs on the plasma proteome

We tested 5.2 million SNPs for association with plasma levels of 1,216 proteins in 1,909 individuals (Methods). We defined the primary protein QTL (pQTL) of a protein as the most significant variant in linkage disequilibrium ($r^2 > 0.2$) within ±1 Mb of the protein-coding gene (Supplementary Note 4)[37,38].

In addition, we performed genome-wide association analysis on nearly 10,000 peptides after quality control. Protein-altering variants can lead to alterations in amino acid sequences and modify the binding surfaces in the case of affinity-based technologies, potentially introducing biases in proteomics studies[17,22,39]. To address this source of potential bias, we developed a framework leveraging peptide-level data to exclude artefactual pQTLs and categorize pQTLs into confidence tiers based on peptide-level evidence (Methods and Extended Data Fig. 2a–d).

Applying a study-wide significance level of $P < 4.1 \times 10^{-11}$ ($5 \times 10^{-8}$ adjusted for 1,216 proteins tested) yielded 1,252 primary pQTLs for 327 proteins (Supplementary Table 4). For downstream analysis, we adopted the conventional genome-wide association study (GWAS) significance threshold of $P < 5 \times 10^{-8}$, identifying 1,947 primary pQTLs for 443 proteins (Fig. 3a,b). Approximate conditional analysis revealed 733 conditionally independent pQTLs for 443 proteins (Methods and Supplementary Table 5). Genomic inflation was well controlled (median lambda$_{GC}$ = 1.002, s.d. = 0.004; see Supplementary Note 5 for quantile–quantile plots). Comparison with our previous dataset, which was generated using an older generation of MS[40], reveals that our methodology produces highly consistent protein quantification across time points and instrumentations (Extended Data Fig. 3a–h, Supplementary Table 6 and Supplementary Note 6). This confirms the robustness of our findings and suggests that our approach can reliably detect biological variation despite technological advancements or delays in sample analysis.

These pQTLs are primarily located in the non-coding regions, with only 3% representing missense and 1% synonymous variants (Fig. 3c, Supplementary Table 4 and Methods). The dominance of non-coding variants among the identified pQTLs is consistent with previous studies reporting 86% and 98% of them[37,41].

Genetic variations normally affect the expression level of the entire protein. Therefore, all peptides identifying the same protein should generally show the same fold change between genotypes. Our peptide-level analysis showed that 77% of reported pQTLs had at least two supporting peptides (Fig. 3d and Supplementary Table 4). Notably, in 94% of these cases, all peptides exhibited the same direction of effect, indicating highly consistent quantitative information at the peptide level (Extended Data Fig. 4a–c). The peptide-level data also helps quantify protein variants affected by amino acid substitutions,

a limitation in affinity-based proteomics as demonstrated by the influence of rs9898 on histidine-rich glycoprotein abundance[42]. We identified the association between rs9898 and circulating histidine-rich glycoprotein levels, which was successfully replicated in both the pediatric and adult cohorts, with a 62% protein sequence coverage and 26 supporting peptides (Extended Data Fig. 5a–d). Importantly, protein quantification was unaffected by the missense mutation (Pro204Ser), which was not identified, probably because it would only produce a four-amino-acid sequence (NCPR) but would have been an outlier if it had. This example illustrates an important advantage of MS-based proteomics in navigating the complexities of protein quantification across variants.

Remarkably, 62% of discovered pQTLs were in *cis* and 60% of the proteins had at least one *cis*-pQTL associated, implying pervasive local regulation (Fig. 3e,f). The MS-based proteomics data recapitulated that the same genomic locus can regulate multiple proteins and that one protein can be regulated by multiple genomic loci. Specifically, 25% of the pQTLs were associated with more than one protein, while 64% of the proteins had multiple pQTLs associated with them and 26% of these had pQTLs located on different chromosomes (Fig. 3g,h).

We reasoned that higher plasma protein abundance would manifest in a higher number and signal of identifying peptides, increasing the likelihood of finding genetic associations. Indeed, as abundance increased, so did the proportion of proteins with genetic associations, a pattern also seen with increased technical reproducibility and peptide count per protein after quality control (Fig. 3i–k).

### Decomposition of variance in plasma protein levels

Having established the quality of the pQTLs, we performed variance decomposition to understand the relative contribution of genetic variation and demographic factors to plasma protein levels (Methods). This revealed that independent pQTLs accounted for 1% to 66% of the variance in protein levels (average, 11%), and for 63% of proteins, pQTLs contributed more variance than age, sex, BMI-SDS and obesity combined (Fig. 4a and Supplementary Table 7). However, some proteins were primarily affected by other factors: SHBG by age and obesity; PRG4 by obesity; and IGF1 and RBP4 by age.

To address the important question of how stable the genetic influences are across pediatric development, we segmented the cohort into 5–9, 10–14 and 15–20-year-olds. This revealed a remarkable stability in genetic influences, with Pearson correlations between 0.95 and 0.98 (Fig. 4b–d). Our results establish that MS-based proteomics can provide unique insights into factors determining protein-level variance throughout pediatric development.

### Characterization of pQTL effect sizes

Next, we investigated the effect size using beta statistics derived from the association tests and allelic fold change calculated on data before normalization[43] (Methods). Although they were mostly mild, 143 pQTLs for 71 proteins exceeded twofold differences in protein levels (absolute $\log_2$(fold change) of >1) between homozygous reference (0/0) and alternative (1/1) genotypes (Supplementary Fig. 2a,b and Supplementary Table 4). These large genetic effect sizes led us to reason that protein levels could predict genotype, which was confirmed for eight proteins with a balanced accuracy of ≥0.8 (Supplementary Table 8 and Methods). Local (*cis*-pQTL) regulation generally had the largest effect sizes (Supplementary Fig. 2c,d). Furthermore, missense mutations, variants in the 5′ and 3′ untranslated regions and transcription factor binding sites had greater average effect sizes than intronic and intergenic variants (Supplementary Fig. 2e), supporting an important role of these regions in transcriptional regulation. Large effect sizes for proteins like MST1, PROCR, BST1, IL1RAP, APOE and LPA may have important implications for clinical and biomarker research (Fig. 5a–f). Notably, the rs2232613-T missense mutation reduced levels of LBP fourfold (Supplementary Table 4). Given the important role of this protein in

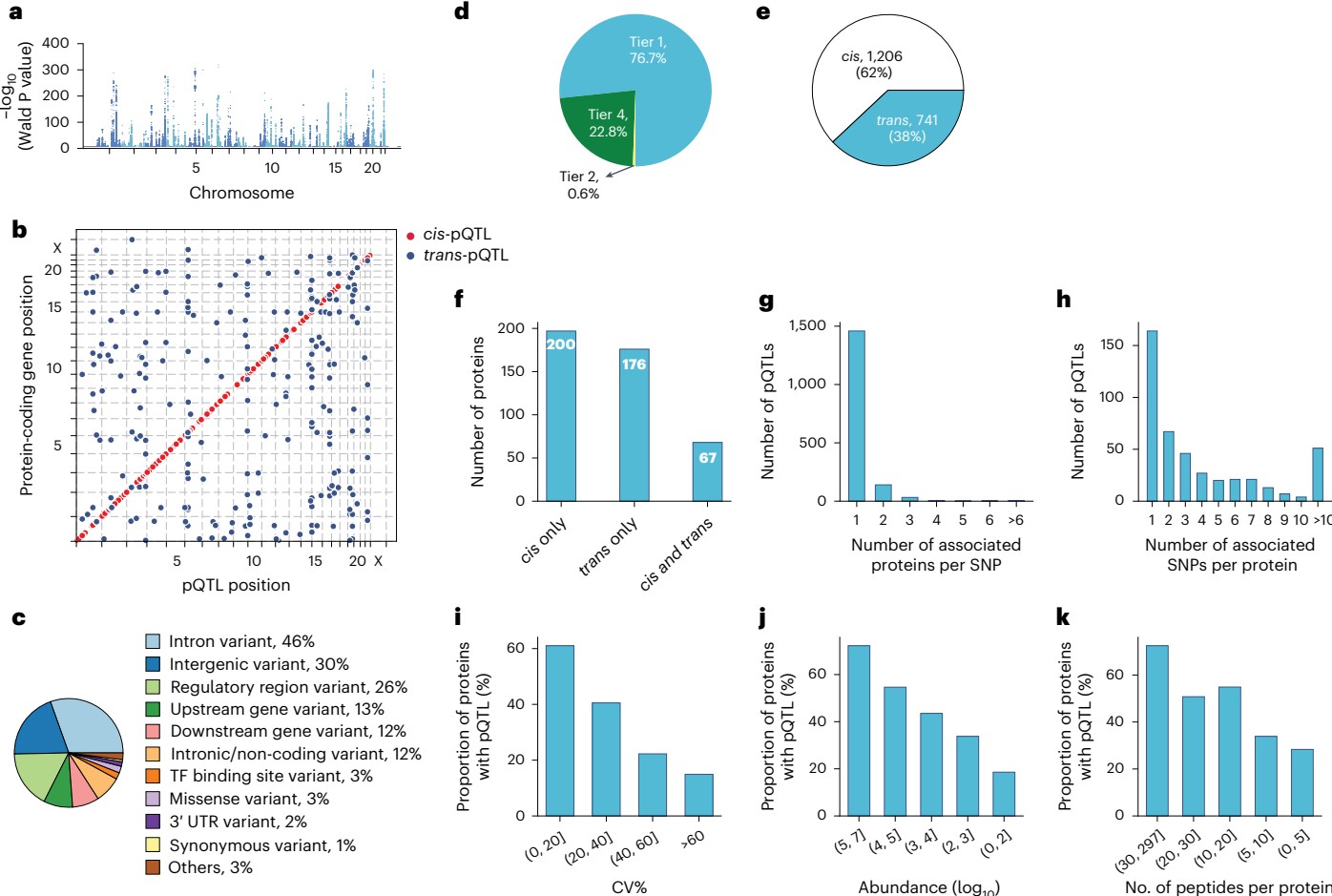

**Fig. 3 | Characterization of pQTLs. a**, Primary pQTLs across the genome (two-sided Wald test in a linear mixed model with genome-wide significance $P < 5 \times 10^{-8}$). **b**, Primary pQTLs against the locations of the transcription start site of the gene coding the protein target. **c**, Variant annotation. **d**, Classification of pQTLs based on peptide-level evidence. **e**, Number of *cis*-pQTLs and *trans*-pQTLs. **f**, Number of proteins that are associated with *cis* only, *trans* only and both *cis* and *trans*-pQTLs. **g**, Distribution of the number of associated proteins per SNP. **h**, Distribution of the number of associated SNPs per protein. **i–k**, Proportion of proteins with genetic associations when stratifying proteins into buckets based on technical variation (**i**), median abundance (**j**) and number of identified peptides after quality control per protein (**k**). TF, transcription factor; UTR, untranslated region; CV, coefficient of variation.

innate immunity, we speculate that individuals with the mutant form have compromised immunity, which has indeed been reported[44]. The observed substantial genetic effects emphasize the importance of considering pQTL information when interpreting findings from clinical and biomarker research, particularly for proteins under strong influence of genetic variation.

**Integrating pQTLs with variant–trait associations**

To investigate whether our MS-based study had discovered previously unreported pQTLs, we compared it to 35 published studies (Extended Data Fig. 6a, Supplementary Table 9 and Methods). This revealed 643 such pQTLs regulating 213 proteins, of which 140 proteins had no previously reported genetic regulation (Supplementary Table 4). Of the remaining pQTLs, 55% were replicated in at least five studies, with *cis*-pQTLs showing higher replication rates than *trans*-pQTLs, indicating stronger and more consistent evidence for *cis*-pQTLs across studies (Extended Data Fig. 6b,c).

GWAS provide increasingly detailed associations between genomic loci and phenotypes but often lack mechanistic insights into the proteins mediating the effects[43,45,46]. To close this gap, we investigated whether our novel pQTLs were associated with phenotypic traits. Mapping our protein-associated variants to the GWAS Catalog revealed 162 such cases (Supplementary Table 10 and Methods).

Many connections were immediately biologically plausible; for example, rs73001065 has been associated with a decreased risk for non-alcoholic fatty liver disease, which our data indicates may happen through reduced APOB levels. Individuals with the bone mineral density-lowering allele rs13469-T in *POLDIP2* had reduced levels of SPP2, a protein crucial for bone metabolism healing[47]. Similarly, children with the type 2 diabetes risk allele rs529565-C in *ABO* had higher MASP1 levels in our data, consistent with elevated MASP1 levels in patients with type 2 diabetes[48,49]. Colocalization analysis supported all these associations (Supplementary Table 11).

Our analysis identified 24 proteins associated with missense variants in various genes, including 11 linked to *APOE*, a key gene in Alzheimer's and cardiovascular disease. Specifically, the APOE-ε4 allele (rs7412-C and rs429358-C), a major Alzheimer's disease risk factor, was associated with lower plasma levels of SPTBN4, a brain-expressed protein implicated in neurodevelopmental disorder[50]. Therefore, we speculated that there is a downregulation of SPTBN4 in Alzheimer's disease, which is indeed supported by previous reports and its epigenetic silencing in patients with Alzheimer's disease[51,52]. Additionally, our data suggests that APOE-ε4 allele's link to depression[53] may involve elevated HTR1D levels, a target of approved treatments for depression and anxiety. These observations offer additional hypotheses for understanding disease mechanisms and potentially alternative therapeutic strategies.

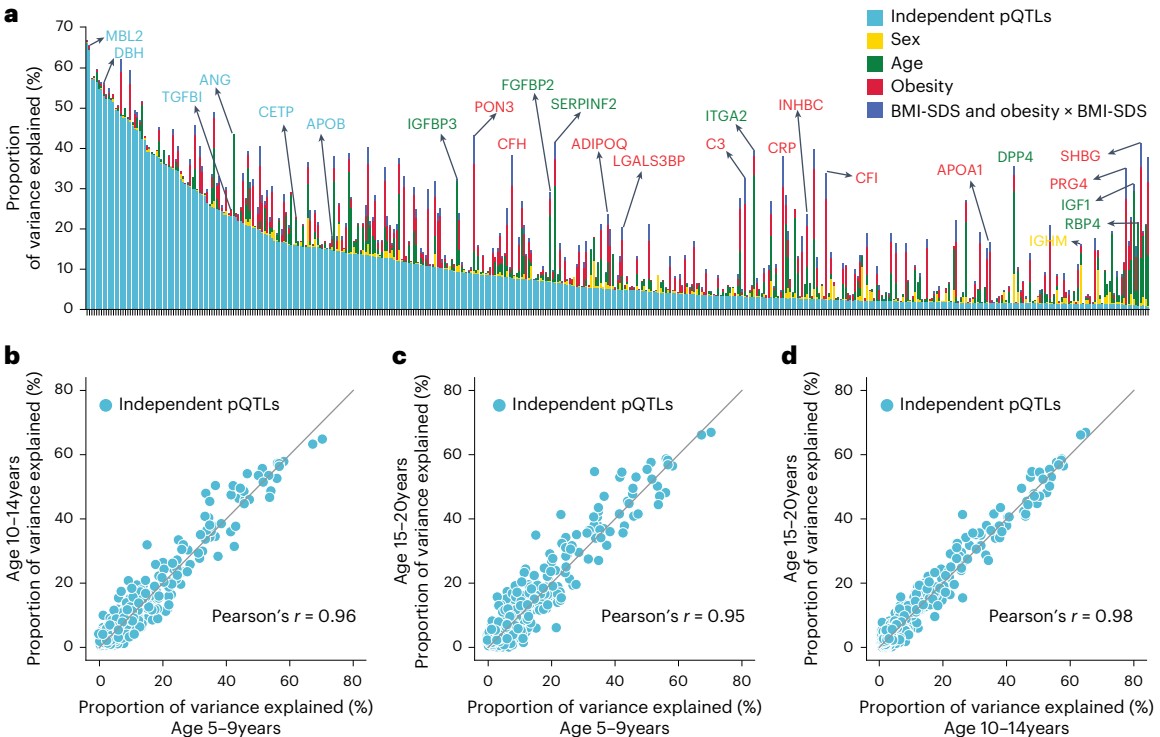

**Fig. 4 | Variance in plasma protein levels explained by various factors.**
**a**, Proportion of variance explained by conditionally independent pQTLs, age, sex, obesity and BMI-SDS (summed variance from BMI-SDS and its interaction with obesity status). Proteins are ordered by decreasing variance attributable to independent pQTLs. **b–d**, Pairwise comparisons of variance explained by independent pQTLs across three age groups: 10-14 years vs. 5-9 years (**b**), 15-20 years vs. 5-9 years (**c**) and 15-20 years vs. 10-14 years (**d**); Pearson correlation coefficients are also shown.

In addition, we performed systematic two-sample Mendelian randomization using the top *cis*-pQTLs from 267 proteins on 47 cardiometabolic GWAS outcomes including obesity, diabetes, atherosclerotic cardiovascular disease, metabolic dysfunction-associated steatohepatitis, Alzheimer's disease and chronic kidney disease (Methods and Supplementary Note 7). This analysis reported 345 causal relationships between 106 proteins and 36 traits related to these six highly prevalent diseases ($P < 2.3 \times 10^{-6}$) (Extended Data Fig. 7a and Supplementary Table 12). Of these, 101 (29%) causal relationships between 41 genes and 33 traits were further validated by colocalization (posterior probability above 70%) (Fig. 5g,h, Supplementary Table 11 and Methods). Interestingly, our data causally connect SHH to height, which may be connected to its crucial role in embryonic development. These results illustrate how integrating high-confidence pQTLs with GWAS results helps to understand the molecular mechanisms between variant–disease and variant–trait associations.

#### Highly replicated pQTLs in children and adults

We assessed the replication rate of the pQTLs in 1,000 children and adolescents and 558 adults, respectively. Around 90% of the pQTLs were eligible for replication owing to independent quality control on proteomics and genomics data (Supplementary Note 8). Of these, we successfully replicated 97% of pQTLs in children (99% of the *cis*, 92% of the *trans* and 92% of the novel) and 91% in adults (92% of the *cis*, 88% of the *trans* and 90% of the novel) with nominal significance ($P < 0.05$) (Supplementary Fig. 3a,b and Supplementary Tables 13 and 14). The high replication rate in adults suggests that the vast majority of detected pQTLs are not life-stage-specific. To detect such potential pQTLs would probably require larger cohorts with similar sizes and health status. Furthermore, the direction and magnitude of effects aligned well between the discovery and replication cohorts (Pearson's $r = 0.97$ and 0.98; Fig. 6a,b), with larger effects observed in the replicated pQTLs (Fig. 6c).

We next asked whether pQTL information could improve biomarker performance, building on our previously reported biomarkers for liver fibrosis, inflammation and steatosis[5]. For half of these biomarkers, including TGFBI and LBP, which showed the largest genetic effect sizes, we identified and replicated pQTLs (Fig. 6d,e). Our data revealed that TGFBI protein levels shifted depending on its corresponding pQTL in both disease and control groups (Fig. 6f). Incorporating genotype information further improved the accuracy of classifying patients with fibrosis stages F0–F1 versus F2–F4 using TGFBI (Extended Data Fig. 8a). Similarly, we observed a marked decrease in LBP levels associated with the rs2232613 variant (Fig. 6g and Extended Data Fig. 8b). These findings suggest that pQTLs should be integrated into biomarker research, especially for proteins with strong genetic effects, although their impact on classification performance should be evaluated individually.

#### Discussion

In this large MS-based plasma proteomics study, we analyzed the plasma proteomes from over 3,000 children and adolescents, mapping highly specific and quantitative protein trajectories throughout childhood and adolescence, including puberty-related differences between male and female adolescents. Prior studies have focused on infancy[11], childhood[10] or adulthood[9]; our study fills a gap by providing insights into the pre-pubertal and post-pubertal stages.

Variance decomposition revealed varying contributions from genetic variants, demographic and health factors to pediatric plasma proteome variation. Notably, about 70% of quantified proteins were significantly regulated, with over one-third influenced by SNPs. These associations enabled accurate predictions of age, BMI or the genotype for a few SNPs from the plasma proteome. Our study provides a valuable resource for pediatric research and biomarker studies, and it is available for further exploration at proteomevariation.org.

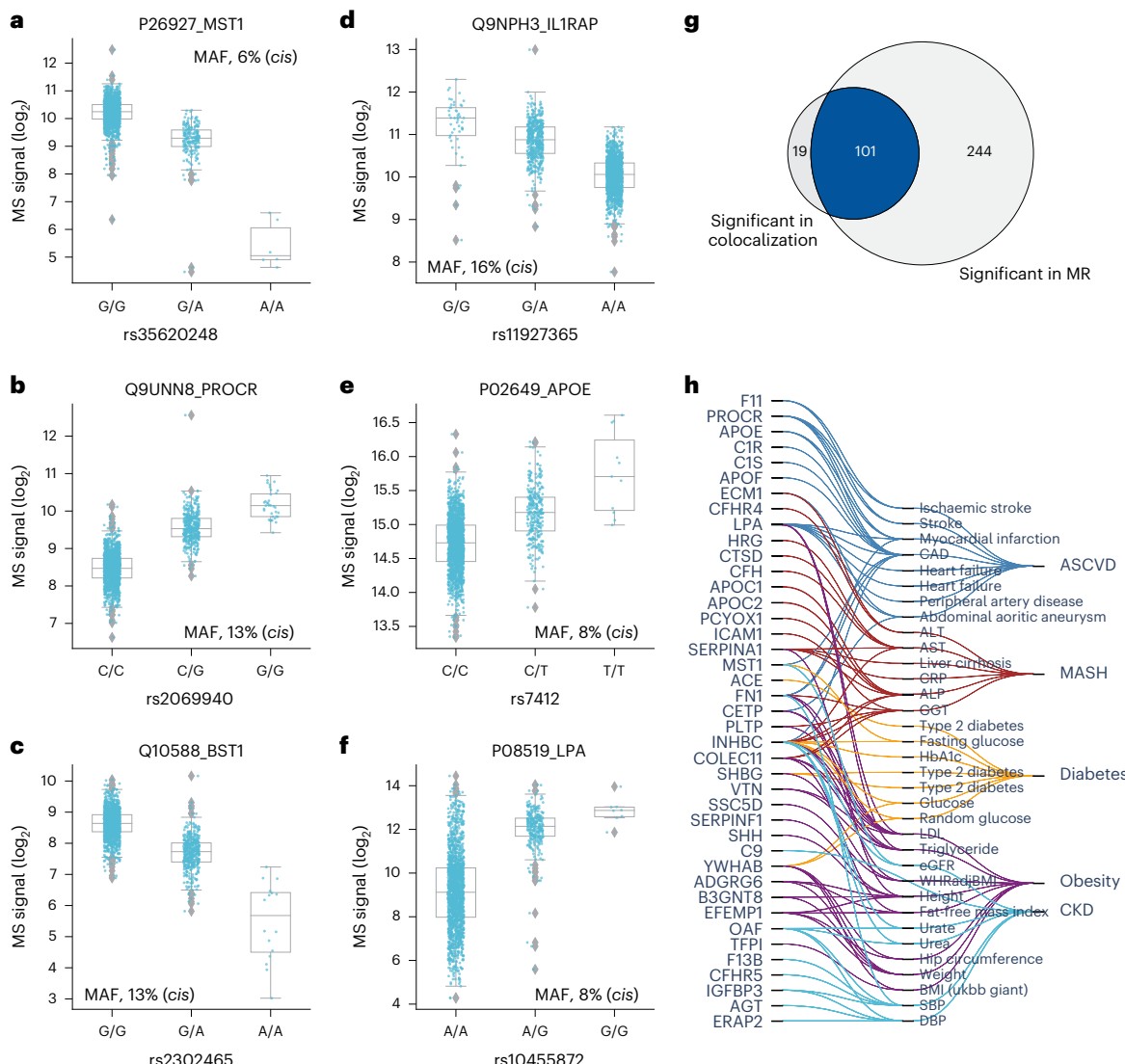

**Fig. 5 | Effect sizes and integration of pQTLs with known variant–trait associations. a–f,** Distribution of log$_2$ intensity values of the top six proteins with the highest absolute beta value in genome-wide association analysis. The gray line in the middle of the box is the median, the top and bottom of the box represent the upper and lower quartile values of the data and the whiskers represent the upper and lower limits for consideration of outliers (Q3 + 1.5 × IQR, Q1 – 1.5 × IQR); IQR, interquartile range (Q3 – Q1); MAF, minor allele frequency. For genotype 0/0:0/1:1/1, the numbers of biological replicates are $n$ = 1278:328:6, 1708:191:0, 1490:399:23, 1225:410:35, 1637:260:15 and 1227:606:79, respectively. Only non-imputed values are shown. **g,** Venn diagram showing the number of protein–outcome pairs that are significant in colocalization analysis (HyPrColoc

method) and two-sample Mendelian randomization (MR), using a two-sided Wald ratio test implemented in the twoSampleMR package with significance defined as $P < 2.5 × 10^{-6}$ (correcting for the number of protein-coding genes). **h,** Protein–trait pairs that are colocalized and with supporting evidence for causation from MR. CAD, coronary artery disease; ALT, alanine aminotransferase; AST, aspartate aminotransferase; CRP, C-reactive protein; ALP, alkaline phosphatase; GGT, gamma-glutamyl transferase; HbA1c, hemoglobin A1c; LDL, low-density lipoprotein; eGFR, estimated glomerular filtration rate; SBP, systolic blood pressure; DBP, diastolic blood pressure; WHRadjBMI, waist-to-hip ratio adjusted for BMI; ASCVD, atherosclerotic cardiovascular disease; MASH, metabolic dysfunction-associated steatohepatitis; CKD, chronic kidney disease.

We identified proteins that to the best of our knowledge were not previously linked to childhood obesity, including decreased levels of brain-enriched OLFM1, heart muscle-enriched NEBL, intestine and pancreas-enriched ANPEP and skeletal muscle-enriched FHL3. UMOD, a kidney-specific protein typically abundant in urine, was also reduced in children with obesity. Although its physiological role in serum is unclear, UMOD may have anti-inflammatory effects[54] and is inversely associated with metabolic syndrome and type 2 diabetes in the older population[55,56]. We speculate that lower UMOD levels may contribute to the chronic low-grade inflammation observed in obesity. These findings facilitate hypothesis generation and provide insights into early metabolic dysregulation involving extrahepatic organs.

Prior studies have demonstrated the importance of large sample sizes in pQTL studies[15]. As far as we know, ours is the largest of its kind, although it remains moderate compared to affinity-based proteomics studies. Nevertheless, we identified pQTLs for over one-third of the quantified plasma proteome with robust peptide-level evidence, including hundreds of novel pQTLs. Our findings suggest that technological improvements in MS-based plasma proteomics will probably enhance the genetic associations detected, even within current studies. Thus, expanding proteome depth without compromising throughput or accuracy is crucial, which may be achievable with emerging workflows combining depletion, multiplexing and further improved MS acquisition schemes.

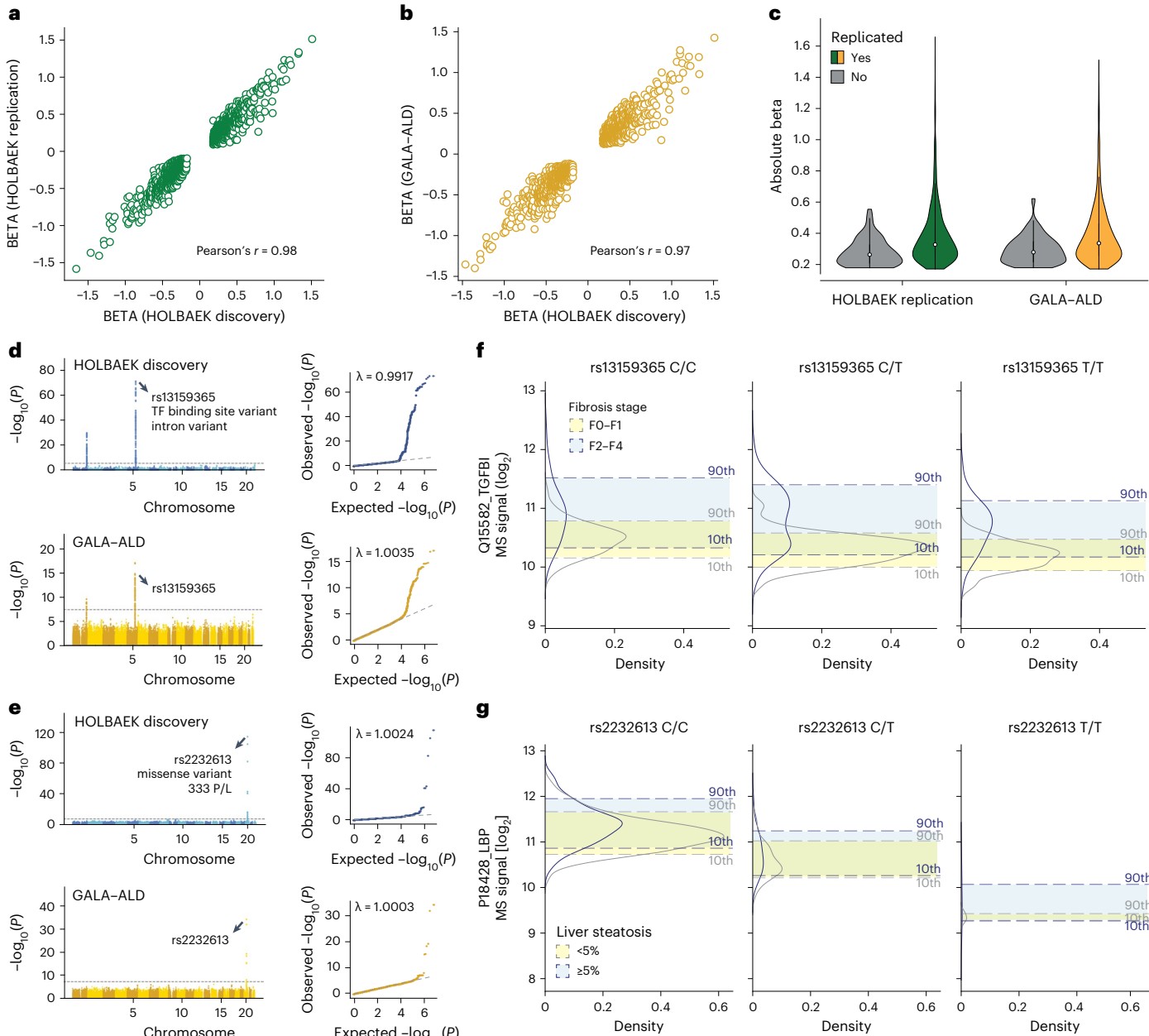

**Fig. 6 | Replication of pQTLs in children and adults. a**, Correlation of beta coefficient for replicated pQTLs in the children replication cohort. **b**, Correlation of beta coefficient for replicated pQTLs in the adult cohort. **c**, Distribution of absolute beta coefficient of replicated and non-replicated pQTLs. **d**, Manhattan plot of association between SNPs and plasma levels of TGFBI in the discovery cohort (upper panel) and adult replication cohort (lower panel) with the lead variant annotated. **e**, Manhattan plot of association between SNPs and plasma levels of LBP in the discovery cohort (upper panel) and adult replication cohort

(lower panel) with the lead variant annotated. **f**, Distribution of plasma levels of TGFBI stratified by the genotype of its lead-associated variant and fibrosis stage in the adult replication cohort. For genotype 0/0:0/1:1/1, n = 96:181:90 and 55:93:43 for fibrosis stage F0–F1 and F2–F4, respectively. **g**, Distribution of plasma levels of LBP stratified by the genotype of its lead-associated variant and steatosis stage in the adult replication cohort. For genotype 0/0:0/1:1/1, n = 318:52:3 and 157:25:3 for steatosis <5% and ≥ 5%, respectively. GALA–ALD, gut and liver axis–alcohol-related liver disease.

Focusing on the Danish population, this study minimized confounding effects from population stratification, genetic admixture and environmental variation, potentially enhancing the identification and replication of pQTLs. It will be interesting to extend this work to other populations.

We replicated known pleiotropy of the *ABO* and *APOE* locus, identifying both known and novel *trans*-pQTLs for 13 and 10 proteins, respectively. Variants in the *ABO* locus regulated proteins critical for homeostasis, including VWF for platelet adhesion, ACE for blood pressure regulation, adhesion proteins like PTPRM, ICAM2, CDH1, CDH5 and

CDH13 as well as MBL2 and MASP1 involving innate immunity. We also identified previously unreported pQTLs in the *ABO* locus for PTPRM, CDH13 and MASP1 along with another potential pleiotropic locus in *AKNA*, which regulated proteins involved in inflammation and immune response, including ORM1, ORM2, S100A8 and S100A and TRAV17.

According to Medawar's mutation accumulation theory, genetic regulation is more robust in early life owing to stronger selective pressures[57,58]. This aligns with our observation that the proportions of variance explained by pQTLs remained relatively stable across the three age groups. The high replication rate in adults further suggests

that pQTLs are largely stable between children and adults, despite influences like disease. However, it would be valuable to investigate how aging affects pQTL detection in the older population as environmental factors and age-related processes become more prominent. Prior studies have shown a 4.7% decline in detected expression QTLs in the blood of patients aged 70–80 years[57] and that aging impacts the predictive power of expression QTLs differently across tissues[59]. This indicates that pQTL stability observed in younger populations may be altered as individuals age.

Our findings emphasize the need to tailor biomarker reference levels according to their associated pQTLs. We found that incorporating pQTL data improved the performance of diagnostic biomarkers, a benefit that could extend to prognostic and predictive biomarkers.

One of the limitations of our study is that it uses a cross-sectional design. A longitudinal approach would better capture age-dependent trajectories of plasma protein levels, reducing inter-individual variability. In addition, the participants' ages follow a normal distribution with a mean of 12 years, and there is limited representation ($n < 30$) at the younger (age 5 years) and older (age 20 years) extremes. This may affect the accuracy of the age-dependent protein abundance trajectories at these age extremes. Furthermore, stringent quality control excluded nearly 90% of SNPs owing to low minor allele frequency or imputation quality, potentially omitting low-frequency pQTLs. A larger sample size could help lower the minor allele frequency threshold.

Existing large-scale pQTL studies have predominantly used affinity-based proteomics platforms optimized for body fluids. Although these platforms report the quantification of thousands of proteins, results often lack consistency across studies, and large-scale validation of binding reagent specificity, ideally by orthogonal methods such as MS[17–19], is still needed. Furthermore, up to one-third of affinity proteomics pQTLs may be affected by epitope effects[60]. Although such artefactual pQTLs can be estimated through conditional and colocalization analysis, they cannot be directly assessed[8,14].

By contrast, MS-based proteomics is highly specific and agnostic to sample type and species, allowing for direct elimination of artefactual pQTLs using peptide-level data. For instance, our peptide-based framework excluded artefactual pQTLs for four proteins: IL6ST (rs2228043), FLT4 (rs34221241), LYZ (rs1800973) and OSMR (rs34675408). The association between FLT4 and missense variant rs34221241 arose from a single peptide, flanking the variant amino acid Asn149Asp. Similarly, LYZ's association with rs1800973 was caused by a peptide flanking Thr88Asn. These data helped flag artefactual pQTLs identified in affinity-based studies; for example, rs11739016 for IL6ST and rs34221241 for FLT4 (ref. 15).

Additionally, our data resolved discrepancies in pQTL effect direction, such as for APOE. The T-allele of rs7412 was negatively associated with APOE levels in an aptamer-based study of the Icelandic population[61] but positively associated in an MS-based study of Han Chinese[23], which aligns with our data in the Danish population. These discrepancies underscore the value of MS-based proteomics in validating and extending pQTL discoveries.

MS-based proteomics is not necessarily immune to the 'epitope effect'[17]. A prior study using exome sequencing and MS found that nearly half of the associations were technical artefacts when using both protein and peptide data for discovery[22]. Our approach addresses this issue by using protein-level data for discovery and peptide-level data for verification, classifying pQTLs into confidence tiers. Alternatively, modifying the protein sequence database upfront can also reduce artefactual pQTLs[62], although it may miss variants in linkage disequilibrium with untyped protein-altering variants. Our method complements these approaches by providing a comprehensive framework for pQTL validation.

To date, affinity-based studies have identified tens of thousands of pQTLs for thousands of circulating proteins, but the search remains incomplete, especially at the tissue and cell-type levels. Many proteins only leak into plasma and primarily function in tissues, where MS-based proteomics can quantify nearly complete proteomes. A few pioneering MS-based proteomics studies with a few hundred samples have already identified pQTLs in the brain and liver[63–65]. We propose expanding these efforts by developing deep and high-throughput workflows and ensuring access to genotyped samples. Analyzing specific cell types in tissues could also be envisioned[66]. Future research should also explore the validation of affinity-based pQTLs using MS-based methods. These efforts will enable fruitful downstream applications like colocalization and Mendelian randomization, aiding in causal inference and drug candidate identification[67,68].

## Online content

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

## Methods

### Ethical approval

The study protocol was approved by the ethics committee for the Region Zealand (protocol no. SJ-104) and is registered at the Danish Data Protection Agency (REG-043-2013). The HOLBAEK Study, including the obesity clinic cohort and the population-based cohort, is also registered at ClinicalTrials.gov (NCT00928473). The study was conducted according to the principles of the Declaration of Helsinki, and oral and written informed consent was obtained from all participants. An informed oral assent was given by the participant if the participant was younger than 18, and the parents gave informed written consent.

### Study participants

We included 2,147 children and adolescents (55% females, 45% males, with 17 missing values because of survey failure) aged 5–20 years from The HOLBAEK Study for pQTL discovery, and an additional 1,000 matched by age, sex and obesity status for replication (58% females, 42% males). The participants were recruited from two groups: the Children's Obesity Clinic, Centre of Obesity Management offering the multidisciplinary childhood obesity management program at Copenhagen University Hospital Holbæk[69], and a population-based cohort from schools across 11 municipalities in Zealand, Denmark[70], as part of a cross-sectional study conducted between January 2009 and April 2019. Eligibility criteria for the obesity clinic group included individuals aged 5–20 years with a BMI above the 90th percentile (BMI-SDS ≥ 1.28) according to Danish reference values[27]. Exclusion criteria for both groups were diagnosed type 1 or type 2 diabetes, treatment with medications including insulin, liraglutide and/or metformin or meeting type 2 diabetes criteria[71] based on the blood sample taken for this study (fasting plasma glucose of >7.0 mmol l$^{-1}$ and/or hemoglobin A1c (HbA1c) > 48 mmol mol$^{-1}$). Tanner stage[72,73] was evaluated by a pediatrician for individuals recruited at the obesity clinic and self-evaluated using a questionnaire with picture pattern recognition for individuals in the population-based group. The Danish population in this study was defined based on residency in Denmark.

### Plasma proteomics

We prepared 2,241 plasma samples in the discovery cohort (including 94 quality assessment samples (pooled plasma samples)) across six batches, and 1,043 plasma samples in the replication cohort (including 43 quality assessment samples) across four batches. Samples were randomized before proteomics sample preparation on an automated liquid handling system (Agilent Bravo) in a 96-well plate format (Supplementary Note 2)[5,74]. Peptides were partially eluted from Evotips with <35% acetonitrile and analyzed using the Evosep One liquid chromatography system (Evosep Biosystems) coupled online to an Orbitrap Astral mass spectrometer (Thermo Fisher Scientific)[25,75]. Eluted peptides were separated on an 8 cm-long PepSep column (150 μm inner diameter packed with 1.5 μm of Reprosil-Pur C18 beads (Dr. Maisch)) using a 21-min gradient and electrosprayed with a stainless emitter (30 μm inner diameter) at 1.9 kV. Data were acquired in DIA mode with full scan and tandem mass spectrum acquired in parallel by the Orbitrap and Astral mass analyzer, respectively. Full scans (380–980 $m/z$) were acquired at 240,000 resolution with a normalized automatic gain control target of 500% and 3 ms maximum injection time. Tandem mass spectra (150–2,000 $m/z$) were acquired at 80,000 resolution with automatic gain control of 500% and a maximum injection time of 5 ms. We used 200 3-Th isolation-window scanning from 380 to 980 $m/z$ and fragmented ions using high-energy collision dissociation with 25% normalized collision energy.

Protein quantification was performed at the MS2 level using the Quant 2.0 algorithm in Spectronaut (v.18), with the 'proteotypicity' filter set to 'only protein group specific'[26]. Default settings were used unless otherwise noted. Data filtering was set to 'Qvalue'. 'Cross-run normalization' was enabled with the strategy of 'local normalization' based on rows with 'Qvalue complete'. False discovery rate was set to 1% at both protein and peptide precursor levels. A spectral library generated using direct DIA from a subset of the samples in this study was used in the targeted analysis of DIA data against the human reference canonical proteome database (2023.05 release). We excluded samples with a protein count lower than the median minus three times the standard deviation across all runs ($n = 2$). The plasma proteomics dataset was filtered for 60% valid values across all samples (proteins with >40% missing values were excluded from downstream statistical analysis), log$_2$ transformed with the remaining missing values imputed by drawing random samples from a normal distribution with downshifted mean by 1.8 and scaled standard deviation (0.3) relative to that of abundance distribution of all proteins in one sample. Specifically, a total of 2,498 proteins were quantified, filtering for 60% valid values across all samples, resulting in a dataset of 1,216 proteins with a data completeness of 91%. The resulting dataset was then corrected for sample preparation batches using ComBat (v.0.3.2)[76].

Tissue specificity annotation was according to the Human Protein Atlas classification, which is based on transcriptomics data[77]. Biological processes and functions represented by the quantified proteins were mapped using Gseapy (v.1.1.1).

### Association of plasma proteome with age, sex and BMI-SDS

Plasma protein levels were normalized by rank-based inverse normal transformation with a Python implementation (https://github.com/edm1/rank-based-INT). The default Blom offset of $c = 3/8$ was adopted. We used the following equation:

$$Y_i^\ell = \Phi^{-1}\left(\frac{r_i - c}{N - 2c + 1}\right)$$

where $r_i$ is the rank of the $i$th observation among the total number of $N$, and $\Phi^{-1}$ denotes the quantile function (or percent point function) implemented in SciPy (v.1.7.1; https://scipy.org/citing-scipy). We used multiple linear regression implemented in Pingouin[78] (v.0.4.0) to estimate the effect of age, sex, BMI-SDS[27], obesity and the interaction of obesity with BMI-SDS on the protein level. Associations were considered significant if the Benjamini–Hochberg-corrected $P$ values were below 0.05. We observed that the largest variation in the plasma proteome is driven by platelet markers, and further investigation revealed unequal distribution of platelet marker levels across the sample collection period (Supplementary Fig. 1e–g), suggesting a systematic bias related to storage duration, although sampling-related technical factors cannot be ruled out. We included pubertal status and time to analysis (plasma sample storage) as covariates in the regression model. We further adjusted for principal component 1 to eliminate potential confounding. Pubertal status was dichotomized according to Tanner stage (Tanner 1, pre-pubertal; Tanner 2–5, pubertal or post-pubertal). The normality of the residuals was tested using the Shapiro–Wilk test implemented in the Python package Pingouin (v.0.5.4), which showed that residuals were normally distributed for 90% of the proteins after rank-based inverse normalized transformation. Linearity between protein levels and predictors was assumed for all proteins as a blanket approach but was not formally tested. Variance inflation factor analysis implemented in the Python package Statsmodels (v.0.13.0) showed no serious multicollinearity among predictors of primary interest (age, sex and BMI-SDS), with all variance inflation factors below ten and those for age and sex below five. Samples with missing values of variables in the regression analysis were excluded ($n = 1,601$ remained including 646 males and 955 females with a median age of 12 years).

### Prediction of age and BMI using plasma proteins

We used the linear regression model from scikit-learn[79] (v.1.0) to predict age and BMI (here we used the BMI without adjusting for age and sex). We split the dataset into training (70%) and test sets (30%), with the training set further divided into training and validation sets (at

a ratio of 70:30). Features were ranked by their absolute correlation coefficients with the outcome. We trained models with an increasing number of features (1–200) and evaluated them on the validation set to determine the optimal number of features based on an incremental decrease in mean squared error. The final model with the selected features was then evaluated on the held-out test set. Mean absolute error and Pearson's *r* between the predicted values and real values in the test set were calculated to indicate prediction accuracy.

### Hierarchical cluster analysis of protein trajectories
Unsupervised hierarchical clustering of age-associated proteins was performed using the Python seaborn package (v.0.12.2). Proteins that passed the Benjamini–Hochberg-corrected *P* value with an absolute coefficient above 0.06 were included. Row clustering was based on median $\log_2$-intensity after *Z*-score normalization across ages.

### Genotyping and imputation in the discovery cohort
Participants in this study were genotyped in three batches on the Infinium HumancoreExome12 (v.1.0) and HumancoreExome24 (v.1.1) Beadchips (Illumina). Genotypes were called using the Genotype module of the GenomeStudio (Illumina). Before imputation, datasets from the three different batches were merged after quality control (only variants present on both chip versions were kept), and monomorphic variants and batch-associated variants were removed (Fisher's exact test, $P < 1 \times 10^{-7}$). We used the Sanger imputation server to phase the genotype data using EAGLE2 (v.2.0.5) and impute it using PBWT with the HRC1.1 panel (GRCh37). We excluded individuals with more than 5% missing genotypes, with heterozygosity that was too high or too low (inbreeding coefficient $\mathrm{abs}(F) > 0.2$), duplicated measurements (keeping the one with higher quality) as well as individuals of non-European descent as determined using principal component analysis based on ancestry informative markers. All study samples whose Euclidean distance from the center fell outside a radius of >1.5× the maximum Euclidean distance of the European reference samples (the 1000 Genomes dataset) were considered non-European. We excluded SNPs with a call rate of <95% and actionable variants. We conducted additional quality control steps for genetic association analysis according to the guidelines[80] using PLINK (v.1.90b6.24) and custom R scripts. In brief, we checked for sex discrepancy based on the X chromosome inbreeding coefficient and none had mismatches. We removed SNPs with an imputation INFO score of <0.7, minor allele frequency of <0.05 and SNPs that deviated from Hardy–Weinberg equilibrium ($P < 1 \times 10^{-6}$) as well as SNPs on the sex chromosomes. We removed individuals with high or low heterozygosity rates (individuals who deviate ±3 s.d. from the sample's heterozygosity rate mean), resulting in a final dataset of 5,242,958 SNPs and 1,924 individuals (846 males, 1,078 females).

### Genome-wide association analysis
For each protein, we adjusted rank-based inverse normal transformed levels for age, sex, BMI-SDS, binary obesity status, obesity × BMI-SDS interaction, PC1 and plasma sample storage time. We standardized the residuals again using rank-based inverse normal transformation and used the standardized values as phenotypes and genotyping arrays as covariates for genome-wide association testing using a univariate linear mixed model implemented in GEMMA[81] (v.0.98.5). We calculated the centered relatedness matrix to control for cryptic relatedness and population stratification. In total, data from 1,909 individuals passed the genotype data quality control and included proteomics and covariate data. We used the Wald test to compute all *P* values. The genomic inflation factor from GWAS results was calculated for each protein using the open-source Python script compute_lambda.py (v.2.0).

### Annotation of SNPs
SNP effects were annotated using the Ensemble Variant Effect Predictor with the RefSeq transcript database. The nearest gene of the SNPs is annotated using GeneLocator (v.1.1.2), a Python package that returns a list of genes or overlapping genes in which the SNP is included, or the gene whose start or end is closest to the specified coordinates in the case a SNP does not fall within any genes.

### Eliminating artefactual pQTLs
Protein-altering variants can create artefactual pQTLs in both affinity-based and bottom-up MS-based proteomics. In bottom-up MS-based proteomics, this occurs because the reference proteome database lacks variant protein versions[17]. Consequently, peptides containing variant amino acids are undetected, and only the reference version is quantified in individuals with the reference allele, creating artificial differences in observed peptide abundances, potentially biasing protein quantification if the reference peptide is used.

Bottom-up MS-based proteomics provides multiple measurements at the tryptic peptide level per protein, offering greater granularity for protein quantification. We leveraged this by assessing pQTLs using peptide-level data and categorizing genuine pQTLs into confidence tiers based on the number of supporting peptides. We performed genome-wide association analysis on approximately 10,000 quality-controlled and protein-group-specific tryptic peptides; that is, peptides unique to a protein group in the context of the canonical human protein sequence database. A peptide is considered significantly associated with a variant if the *P* value is below the threshold after adjusting for the number of primary variant–protein associations ($2.4 \times 10^{-5}$).

- Tier 1: a variant–protein association with at least two supporting peptides showing directionally concordant associations;
- Tier 2: a variant–protein association with only one supporting peptide, the SNP is non-synonymous and does not reside within the protein-coding gene;
- Tier 3: a variant–protein association with only one supporting peptide, the SNP is non-synonymous and resides within the protein-coding gene. However, the SNP-encoded single amino acid variant should not be part of the supporting peptide;
- Tier 4: a variant–protein association with only one supporting peptide, and the SNP is synonymous.

For tiers 2–4, the total number of peptides available for testing must not exceed three, as the SNP–protein associations in these tiers are supported by only one peptide.

### Decomposition of variance in pediatric plasma protein levels
We estimated the variance explained by various predictors using a stepwise addition approach in a multiple regression model. Predictors were added in this order: independent pQTLs, sex, age, obesity status, BMI-SDS and the interaction between BMI-SDS and obesity status. Incremental sums of adjusted squares were calculated at each step to determine the additional variance explained.

### Effect size of pQTLs
In addition to beta statistics derived from the association test, we determined pQTL effect sizes by calculating the ratio of mean protein abundance in heterozygotes (1/0) and homozygotes (1/1) relative to wild type (0/0), without data transformation.

### Prediction of genotype based on protein levels
We used QLattice[82], a symbolic regression method implemented in the Feyn Python module (v.3.0.1) to predict the presence of variant alleles based on protein levels in the derivation cohort. The model was allowed to include age and sex as predictors. The derivation cohort was randomly split into a training set (70%) and a validation set (30%), stratified by genotype. Classification performance was evaluated on the validation set, which was excluded from training.

## Comparison with previous pQTL studies

To assess the novelty of the identified pQTLs, we compared our results at genome-wide significance level to 35 previously published pQTL studies, encompassing over 100,000 pQTLs for 5,000 distinct genes from various proteomics platforms, primarily SomaScan and Olink panels. For all studies, we retained the pQTLs at the reported significance levels. We defined novelty if no variants within ±1 Mb of our primary pQTLs had been previously reported for the corresponding protein. Replication was defined otherwise. Linkage disequilibrium was not considered in this analysis. A comparison was done at the protein level by matching the reported gene name from each study. Gene names are mapped based on Uniprot IDs through Uniport ID mapping in case of missing values. Genome coordinates based on GRCh38 were converted to GRCh37 using pyliftover (v.0.4).

## Mapping pQTLs to GWAS catalog results

We mapped the primary pQTLs to GWAS Catalog results (v.1.0.2) through SNP IDs to identify any associations with disease traits. Linkage disequilibrium was not considered in this analysis.

## Two-sample Mendelian randomization

We performed systematic Mendelian randomization on 47 cardiometabolic GWAS outcomes spanning obesity, diabetes, atherosclerotic cardiovascular disease, metabolic dysfunction-associated steatohepatitis, Alzheimer's disease and chronic kidney disease. Mendelian randomization was calculated with the Wald ratio method (TwoSampleMR v.0.5.6)[83] to test whether the top *cis*-pQTLs (variant with the lowest *P* value) were causally linked to any GWAS outcomes, with significance defined as Wald ratio Mendelian randomization $P < 2.5 \times 10^{-6}$ (correcting for the number of protein-coding genes).

## Colocalization analysis

We performed pairwise colocalization analysis using HyPrColoc (R package hyprcoloc v.1.0.0)[84]. The prior probability of a SNP being causal to one trait in a region was set to $1 \times 10^{-4}$ by default. A colocalization signal was defined by a regional probability of ≥0.6 and a posterior probability of ≥0.7 with non-uniform priors.

## Statistics and reproducibility

Sample size was not predetermined statistically, as this depends on the proteomics workflow throughput, but our sample sizes are similar to those reported in previous publications[38,85,86]. Samples failing proteomics quality control (*n* = 2; Methods) were excluded to minimize potential analytical biases. Additionally, we excluded participants with genomic data not meeting quality control standards and non-European ancestry (Methods) for population homogeneity. Plasma samples were randomized, and experimenters were blinded to participants' genotype data but not age, BMI or sex.

## Reporting summary

Further information on research design is available in the Nature Portfolio Reporting Summary linked to this article.

## Data availability

The GWAS summary statistics generated in this study can be downloaded from the GWAS Catalog (https://www.ebi.ac.uk/gwas), under accession IDs GCST90452968 to GCST90454170: https://ftp.ebi.ac.uk/pub/databases/gwas/summary_statistics/GCST90452001-GCST90453000/; https://ftp.ebi.ac.uk/pub/databases/gwas/summary_statistics/GCST90453001-GCST90454000/; https://ftp.ebi.ac.uk/pub/databases/gwas/summary_statistics/GCST90454001-GCST90455000/. Accession IDs or download links for publicly available GWAS summary statistics datasets used in this study are listed in Supplementary Note 7. The canonical human reference proteome database (2023.05 release) was downloaded from the European Bioinformatics Institute database (https://ftp.ebi.ac.uk/pub/databases/reference_proteomes); tissue specificity annotation of proteins was downloaded from the Human Protein Atlas database (https://www.proteinatlas.org/about/download). The GWAS Catalog (v.1.0.2) was downloaded from https://www.ebi.ac.uk/gwas/docs/file-downloads; protein transcription start sites were extracted from BioMart (accessed on 23 November 2023) (https://grch37.ensembl.org/info/data/biomart/index.html); the gene sets used for mapping biological processes and functions can be accessed through Gseapy (v.1.1.1) using the identifiers 'MSigDB_Hallmark_2020' and 'GO_Biological_Process_2023'. All analysis results are available as supplementary tables. Searchable results are publicly accessible at proteomevariation.org. The study protocol is also available upon request to Jens-Christian Holm (jhom@regionsjaelland.dk). Owing to GDPR regulations, individual-level clinical metadata, genomics and proteomics data generated in this study cannot be made publicly available but are available upon request to the corresponding authors. The time frame for response to requests from the authors is within 1 month. When processing data, certain restrictions apply: a data processing agreement must be signed between the data controller and processor; data must not be processed for purposes other than statistical and scientific studies; and personal data must be deleted, anonymized and destroyed at the end of investigation and must not be passed on to a third party or individuals who are not authorized to access the data. Source data are provided with this paper.

## Code availability

The software used in this study can be accessed as follows: EAGLE2 (v.2.0.5), https://alkesgroup.broadinstitute.org/Eagle; Spectronaut (v18), https://biognosys.com/software/spectronaut; Python (v.3.9.0 and v.3.8.11), https://www.python.org; Combat (v.0.3.2), https://pypi.org/project/combat; Gseapy (v.1.1.1), https://gseapy.readthedocs.io; Seaborn (v.0.12.2), https://seaborn.pydata.org; QLattice implemented in the Feyn Python module (v.3.0.1), https://www.abzu.ai/qlattice; Compute-lambda.py (v.2.0), https://github.com/pgxcentre/lambda; Bcftools (v.1.14), https://samtools.github.io/bcftools; Gemma (v.0.98.5), https://github.com/genetics-statistics/GEMMA; Plink (v.1.90b6.24), https://www.cog-genomics.org/plink; GCTA-COJO (v.1.93.3), https://yanglab.westlake.edu.cn/software/gcta; variant effect predictor was performed online on 2 March 2024 with the RefSeq transcript database (http://grch37.ensembl.org/Homo_sapiens/Tools/VEP); Scikit-learn (v.1.0), https://scikit-learn.org/stable/whats_new/v1.0.html; INT-transformation (v.2019.08.21), https://github.com/edm1/rank-based-INT; Scipy (v.1.7.1), https://scipy.org; Pingouin (v.0.4.0 and v.0.5.4), https://pingouin-stats.org/build/html/index.html; Statsmodels (v.0.13.0), https://www.statsmodels.org/stable/index.html; Pyliftover (v.0.4), https://pypi.org/project/pyliftover; GeneLocator (v.1.1.2), https://pypi.org/project/GeneLocator; Hyprcoloc (v.1.0.0), https://github.com/cnfoley/hyprcoloc; and TwoSampleMR (v.0.5.6), https://mrcieu.github.io/TwoSampleMR. The custom scripts can be accessed through GitHub (https://github.com/llniu/pQTL_Holbaek-Study) and Zenodo (https://doi.org/10.5281/zenodo.14474112)[87].

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

## Acknowledgements

Our gratitude goes to all participants and their families from The HOLBAEK Study. We appreciate the staff at The Children's Obesity Clinic for their assistance in clinical studies, especially G. Holløse and T. Larsen for providing the state of the acquisition, storage and maintenance of biological samples and clinical data. J. Bork-Jensen and J. V. T. Lominchar at the Phenomics Platform of Novo Nordisk Foundation Center for Basic Metabolic Research deserve recognition for their genomics data processing contributions. We also acknowledge the members of the Clinical Proteomics Group at Novo Nordisk Foundation Center for Protein Research and the Department of Proteomics and Signal Transduction (Max Planck Institute of Biochemistry). Special thanks to L. Drici, V. Albrecht and A. Brunner for their technical assistance. J. Madsen and M. Wierer, Director of Proteomics Research Infrastructure (PRI), provided valuable technical support as well. Our appreciation extends to V. Gudmundsdottir at the University of Iceland for kindly sharing the pQTL results she had collected from 20 existing pQTL studies, and to L. Folkersen at Nucleus and Tobias Nyholm Wistisen at Novo Nordisk for technical discussions. The research reported in this publication was supported in part by Novo Nordisk Fonden under grant nos. NNF15CC0001 (to M.M.), NNF15OC0016692 (to the MicrobLiver Consortium), NNF15OC0016544 (to T.H.), NNF14CC0001 and NNF21SA0072102 (to S.R.), NNF20OC0059393 (to M.T.), NNF18SA0034956 (to C.E.F.) and NNF20SA0067242 (to L.A.H.); Innovationsfonden under grant no. 0603-00484B (to T.H.); Horizon 2020 Framework Programme under grant no. 668031 (to the GALAXY Consortium); Region Zealand Health and Medical Research Foundation under grant no. R32-A1191 (to C.E.F.); and the Danish Heart Foundation under grant no. PhD2023009-HF (to L.A.H.). The funders had no role in study design, data collection and analysis, decision to publish or preparation of the manuscript.

## Author contributions

L.N., S.E.S., T.H. and M.M. conceptualized the study. L.N. designed the experimental and computational workflow and performed the proteomics experiments, bioinformatic and statistical analyses, data interpretation, visualizations and manuscript drafting. S.S. selected samples and assisted in designing the computational pipeline. L.C. and J.M. assisted in designing the computational pipeline. L.A.H., M.A.V.L., C.E.F. and H.B.J. curated and advised on the use of clinical metadata. J.F. performed Mendelian randomization analysis. M.T. and A.K. provided clinical data for the adult replication cohort. J.-C.H. provided clinical data for the HOLBAEK Study. S.R. supervised the computational pipeline and revised the manuscript, as did T.H. with genomics data. L.N. and M.M. wrote and edited the manuscript with input from all coauthors.

## Funding

## Competing interests

M.M. is an indirect investor in Evosep. L.N. and J.F. are employees of Novo Nordisk; however, this work was conducted while L.N. was a full-time employee at the University of Copenhagen. M.T. is a co-founder and board member of Evido. She is also a board member of the non-governmental organization Alcohol & Society. She receives speaker fees from Siemens Healthcare, Echosens, Norgine, Madrigal, Takeda and Tillotts Pharma as well as advisory fees from Boehringer Ingelheim, Astra Zeneca, Novo Nordisk and GSK. A.K. is a co-founder and board member of Evido. He has served as a speaker for Novo Nordisk, Norgine and Siemens, and has participated in advisory boards for Siemens, Boehringer Ingelheim and Novo Nordisk. Additionally, he receives research support from Astra Zeneca, Siemens, Nordic Bioscience and Echosense, all outside the submitted work. The other authors declare no competing interests.

## Additional information

**Extended data** is available for this paper at https://doi.org/10.1038/s41588-025-02089-2.

**Correspondence and requests for materials** should be addressed to Simon Rasmussen, Torben Hansen or Matthias Mann.

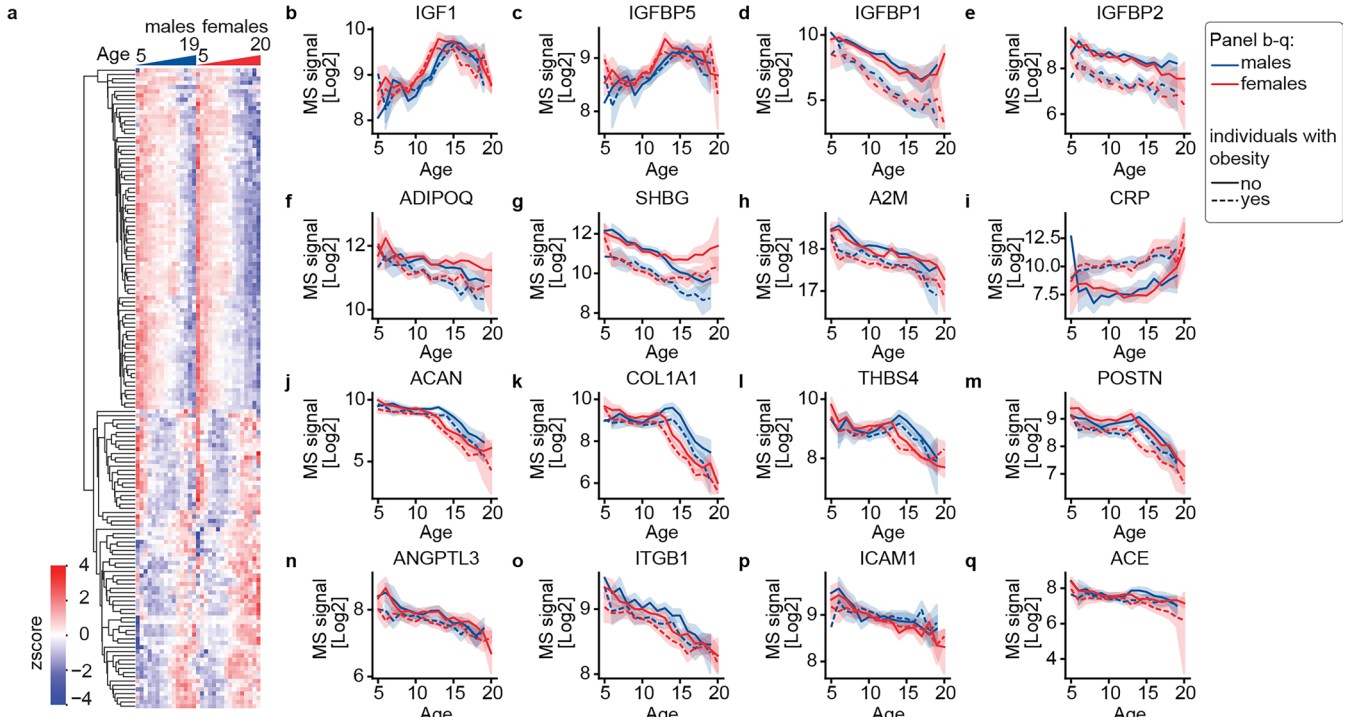

**Extended Data Fig. 1 | Sex- and obesity-dependent temporal plasma proteome profiles. a**, Hierarchical clustering dendrogram of proteins that are significantly associated with age. The heat maps display z-scored median intensities across age for girls (n = 1,170) and boys (n = 958). **b-q**, Temporal trajectories of representative proteins stratified by sex and obesity status in Panel (**a**). Mean values along the age axis and 95% confidence intervals are shown. Obesity status is classified based on BMI-SDS, with 'yes' indicating a BMI-SDS ≥ 1.28 (above the 90th percentile) according to Danish reference values, and 'no' otherwise. The number of participants in each trajectory group is as follows: males with obesity (n = 538), females with obesity (n = 641), males without obesity (n = 420), and females without obesity (n = 529). MS: mass spectrometry; BMI-SDS: body mass index standard deviation score.

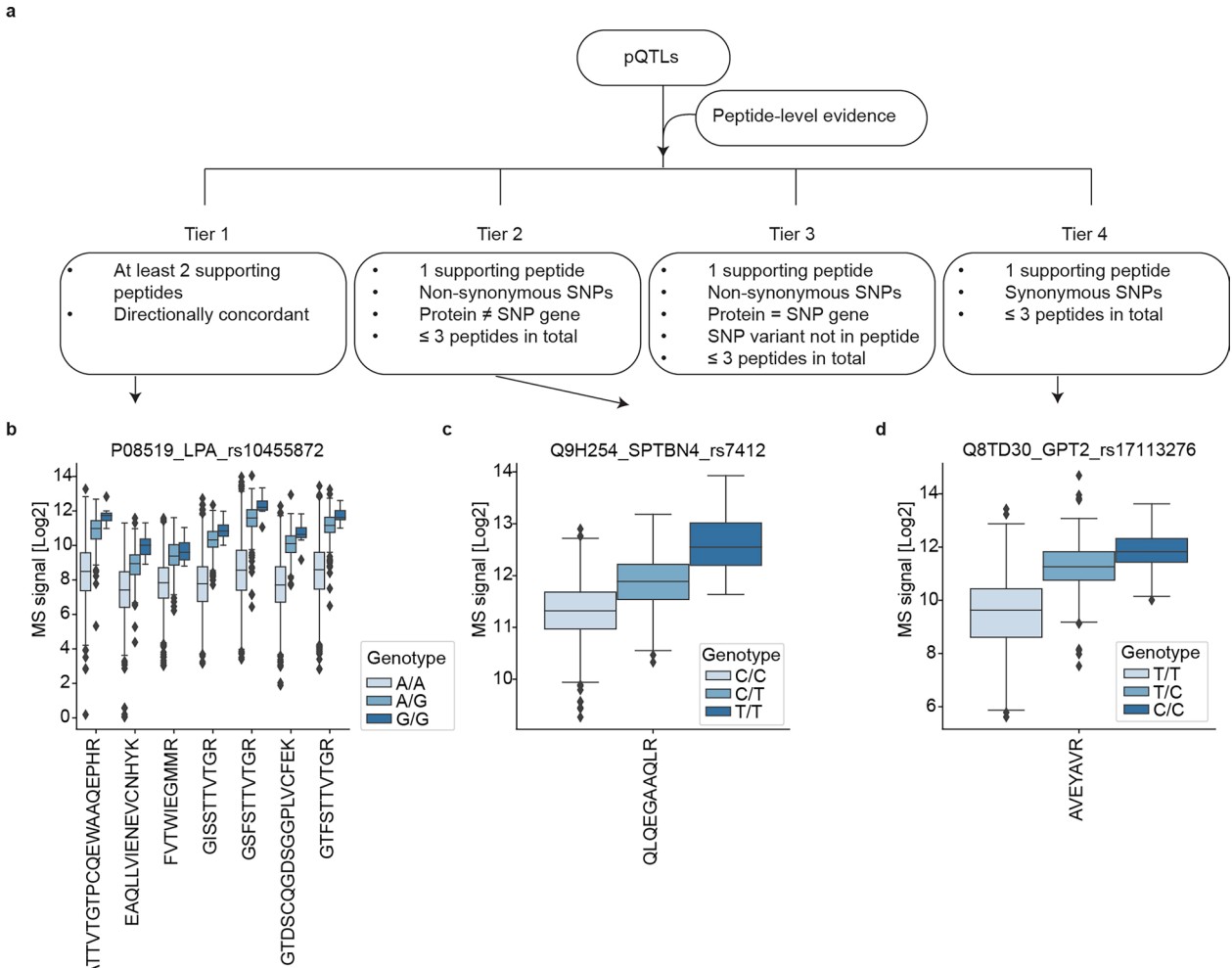

**Extended Data Fig. 2 | Classification of pQTLs based on peptide-level data.**
**a**, Classification scheme and criteria. **b**, Example of a Tier 1 pQTL. The number of biological replicates per genotype in the order of A/A, A/G, G/G for each peptide from left to right is: 1,162/281/8, 572/274/8, 1,030/285/8, 1,242/284/8, 1,305/287/8, 1,240/283/8, 1,236/286/8. **c**, Example of a Tier 2 pQTL. The number of biological replicates per genotype in the order of C/C, C/T, and T/T is 1,455/255/9. **d**, Example of a Tier 4 pQTL. The number of biological replicates per genotype

in the order of T/T, T/C and C/C is 1,371/391/23. For Panels (**b**), (**c**), and (**d**), the gray line in the middle of the box is the median, the top and bottom of the box represent the upper and lower quartile values of the data and the whiskers represent the upper and lower limits for consideration of outliers (Q3 + 1.5 × IQR, Q1 − 1.5 × IQR). IQR represents the interquartile range (Q3 − Q1). pQTL: protein quantitative trait locus; SNP: single nucleotide polymorphism; MS: mass spectrometry.

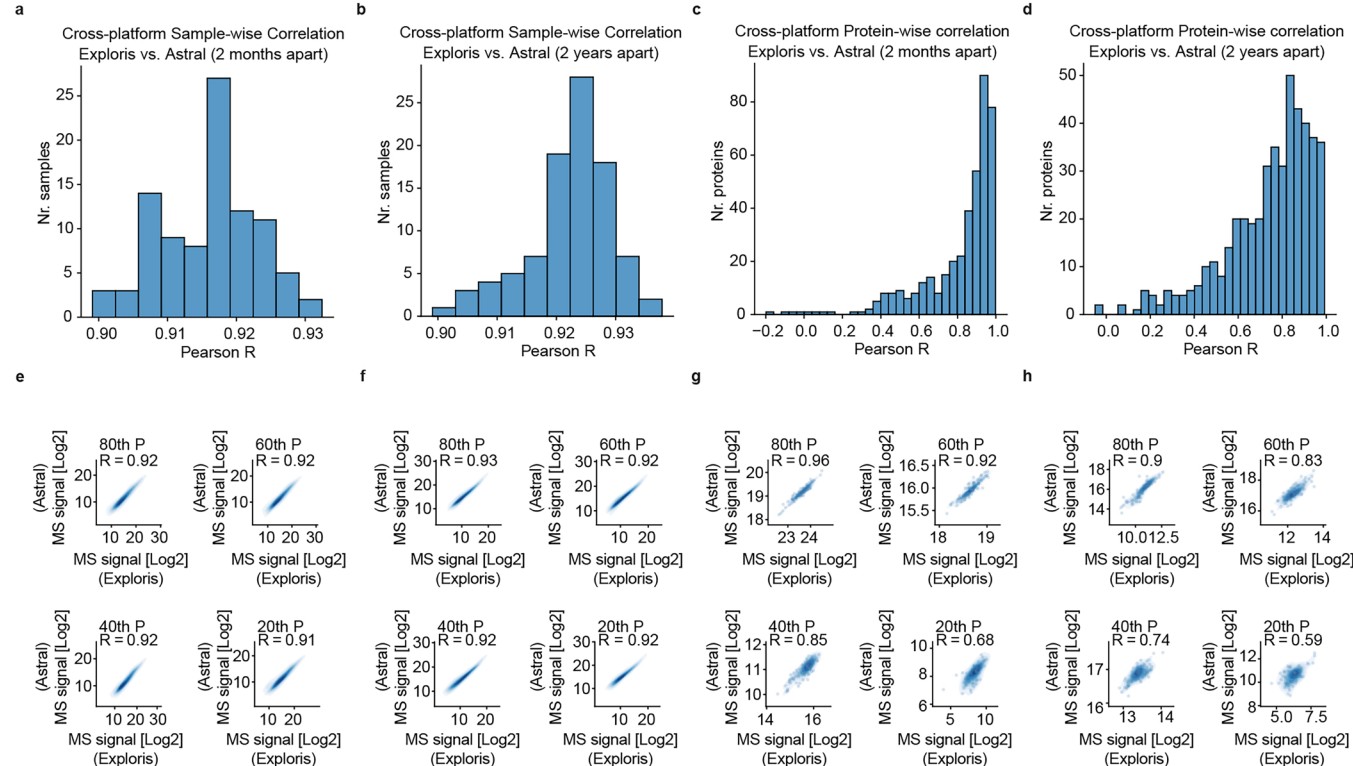

**Extended Data Fig. 3 | Quantification consistency between old and new instrumentation and the effect of delayed measurements. a-b**, Distribution of sample-wise proteome Pearson correlation coefficients (*n* = 94 samples) between old and new instruments with a 2-month (**a**) and 2-year (**b**) measurement gaps. **c-d**, Distribution of protein-wise Pearson correlation coefficients between old and new instruments with (**c**) 2-month gap (408 overlapping proteins) and

(**d**) 2-year gap (465 overlapping proteins). **e-f**, Density distribution of pair-wise proteomes for samples at 80th, 60th, 40th, and 20th percentile correlation coefficients from Panel (**a**) and (**b**), respectively. **g-h**, Density distribution of protein values at 80th, 60th, 40th, and 20th percentile correlation coefficients from Panel (**c**) and (**d**), respectively. MS: mass spectrometry; P: percentile.

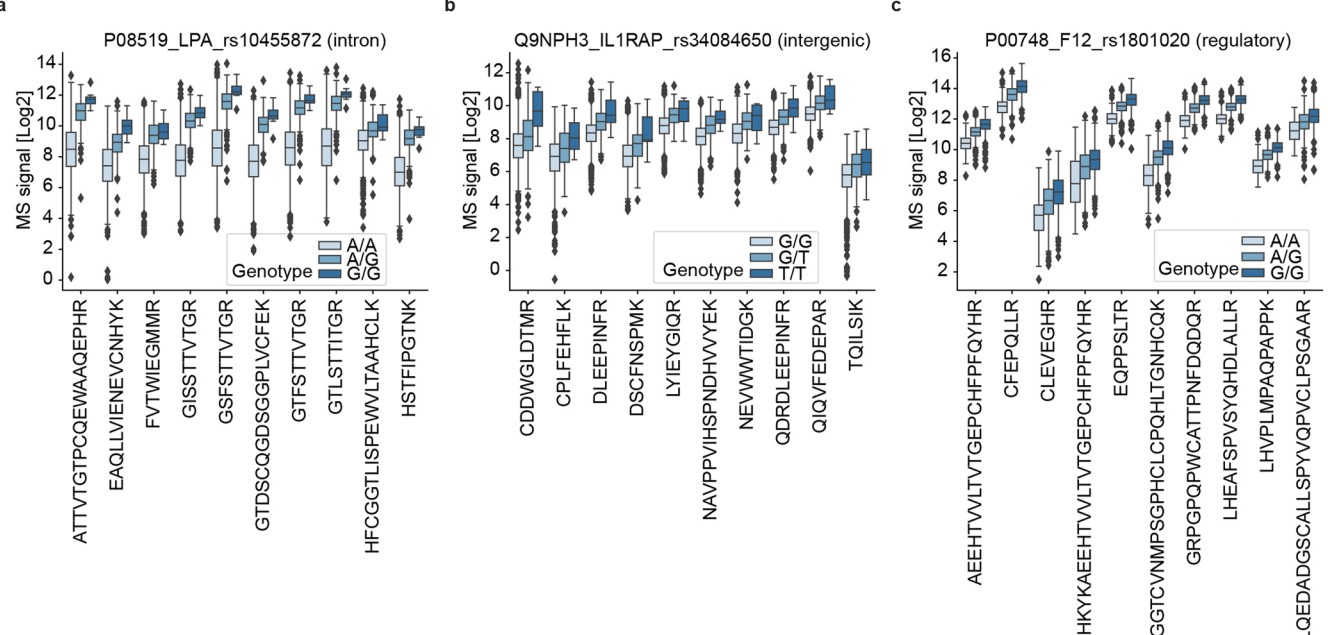

**Extended Data Fig. 4 | Examples of Genetic Variants and Peptide Quantity.**
**a-c**, Peptide quantities by genotype illustrating common regulation of peptide quantities for examples of variants located in the (**a**) intron region, (**b**) intergenic region and (**c**) regulatory region. Up to ten peptides per protein are displayed. The number of biological replicates per genotype in the order of A/A, A/G, and G/G for peptides from left to right in Panel (**a**) is as follow: 1,162/281/8, 572/274/8, 1,030/285/8, 1,242/284/8, 1,305/287/8, 1,240/283/8, 1,236/286/8, 1,156/286/8, 906/277/8, 1,088/286/8. The number of biological replicates per genotype in the order of G/G, G/T, T/T for peptides from left to right in Panel (**b**) is as follow: 1,191/161/6, 1,358/174/7, 1,596/199/8, 1,015/149/5, 1,610/203/8,

1,557/200/6, 1,565/191/8, 1,701/204/8, 1,705/204/8, 1,397/189/8. The number of biological replicates per genotype in the order of A/A, A/G, G/G for peptides from left to right in Panel (**c**) is as follow: 119/718/1,084, 119/718/1,084, 83/593/994, 101/672/1,043, 119/718/1,083, 112/696/1,072, 119/718/1,084, 119/718/1,084, 119/718/1,084, 119/718/1,084. For Panels (**a**), (**b**) and (**c**), the gray line in the middle of the box is the median, the top and bottom of the box represent the upper and lower quartile values of the data and the whiskers represent the upper and lower limits for consideration of outliers (Q3 + 1.5 × IQR, Q1 − 1.5 × IQR). IQR represents the interquartile range (Q3 − Q1). MS: mass spectrometry.

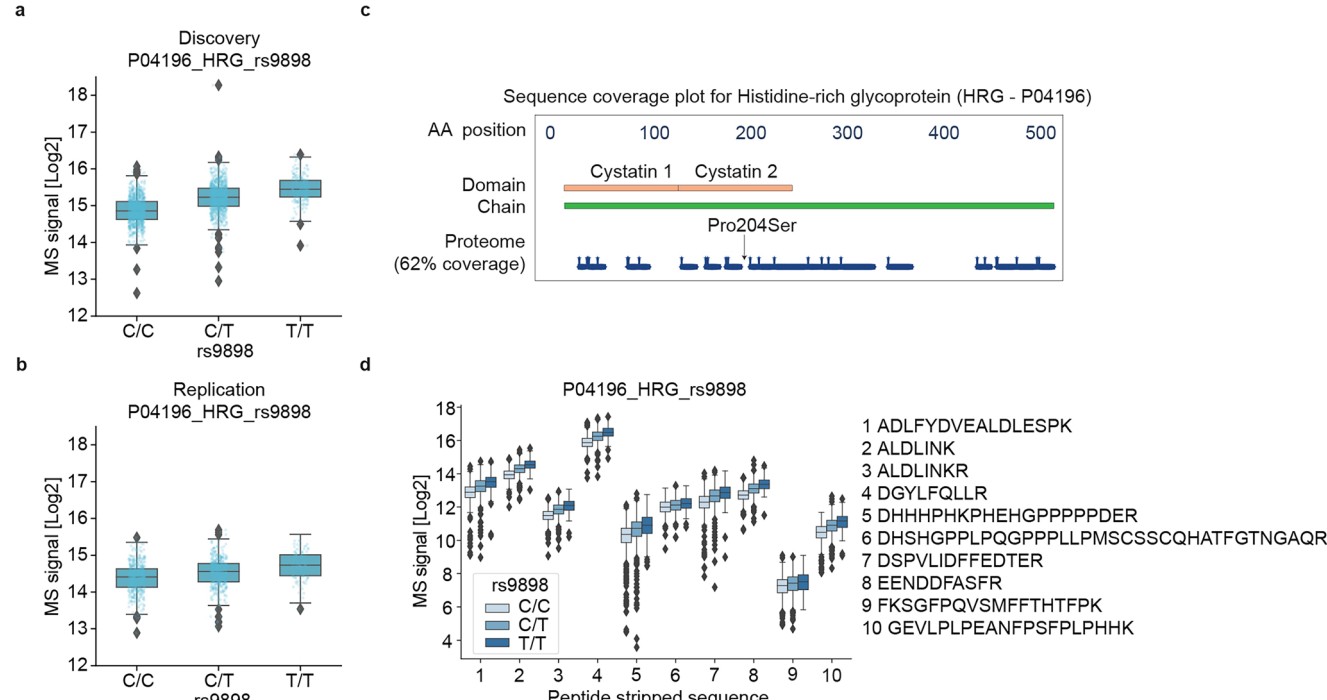

**Extended Data Fig. 5 | Peptide level evidence for the association between rs9898 and circulating HRG levels. a-b**, Increasing protein levels with the number of T alleles at rs9898 in the discovery (**a**) and children replication (**b**) cohorts. The number of biological replicates per genotype in the order of C/C, C/T and T/T in Panels (**a**) and (**b**) is 868/828/213 and 454/421/115, respectively. **c**, Sequence coverage of the HRG protein. **d**, Increasing peptide levels with the number of T alleles at rs9898. Up to ten HRG-derived peptides are displayed. The number of biological replicates per genotype in the order of C/C, C/T, and

T/T for the peptides from left to right is as follows: 871/836/214, 871/836/214, 871/836/214, 871/836/214, 855/826/209, 871/836/214, 871/836/214, 871/836/214, 752/723/190, 869/835/214. For Panels (**a**), (**b**) and (**d**), the gray line in the middle of the box is the median, the top and bottom of the box represent the upper and lower quartile values of the data, and the whiskers represent the upper and lower limits for consideration of outliers (Q3 + 1.5 × IQR, Q1 − 1.5 × IQR). IQR represents the interquartile range (Q3 − Q1). MS: mass spectrometry.

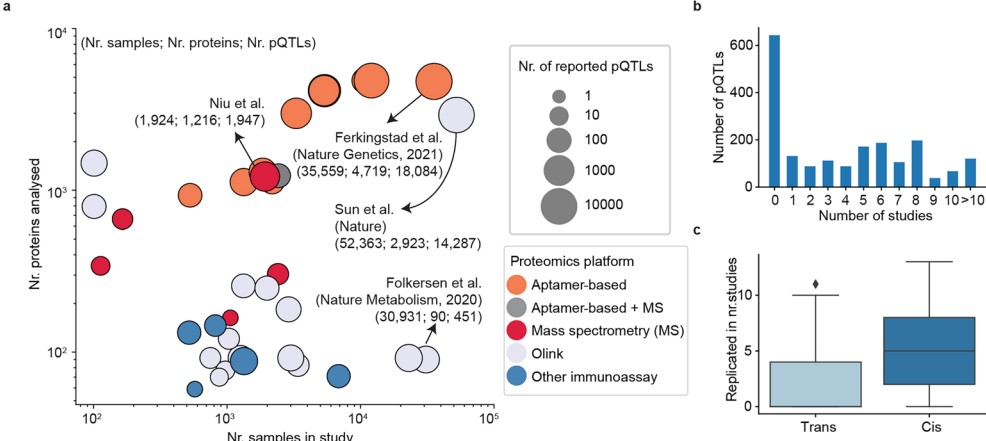

**Extended Data Fig. 6 | Comparison of pQTLs to previous plasma or serum studies. a**, Comparison of the number of proteins analyzed and the number of samples analyzed by 35 previous studies. **b**, The number of protein quantitative trait loci (pQTLs) replicated in previous studies. **c**, Number of published studies in which the cis- ($n = 1,206$) and trans-pQTLs ($n = 741$) are replicated. The gray line in the middle of the box is the median, the top and bottom of the box represent the upper and lower quartile values of the data, and the whiskers represent the upper and lower limits for consideration of outliers (Q3 + 1.5 × IQR, Q1 − 1.5 × IQR). IQR represents the interquartile range (Q3 − Q1). MS: mass spectrometry.

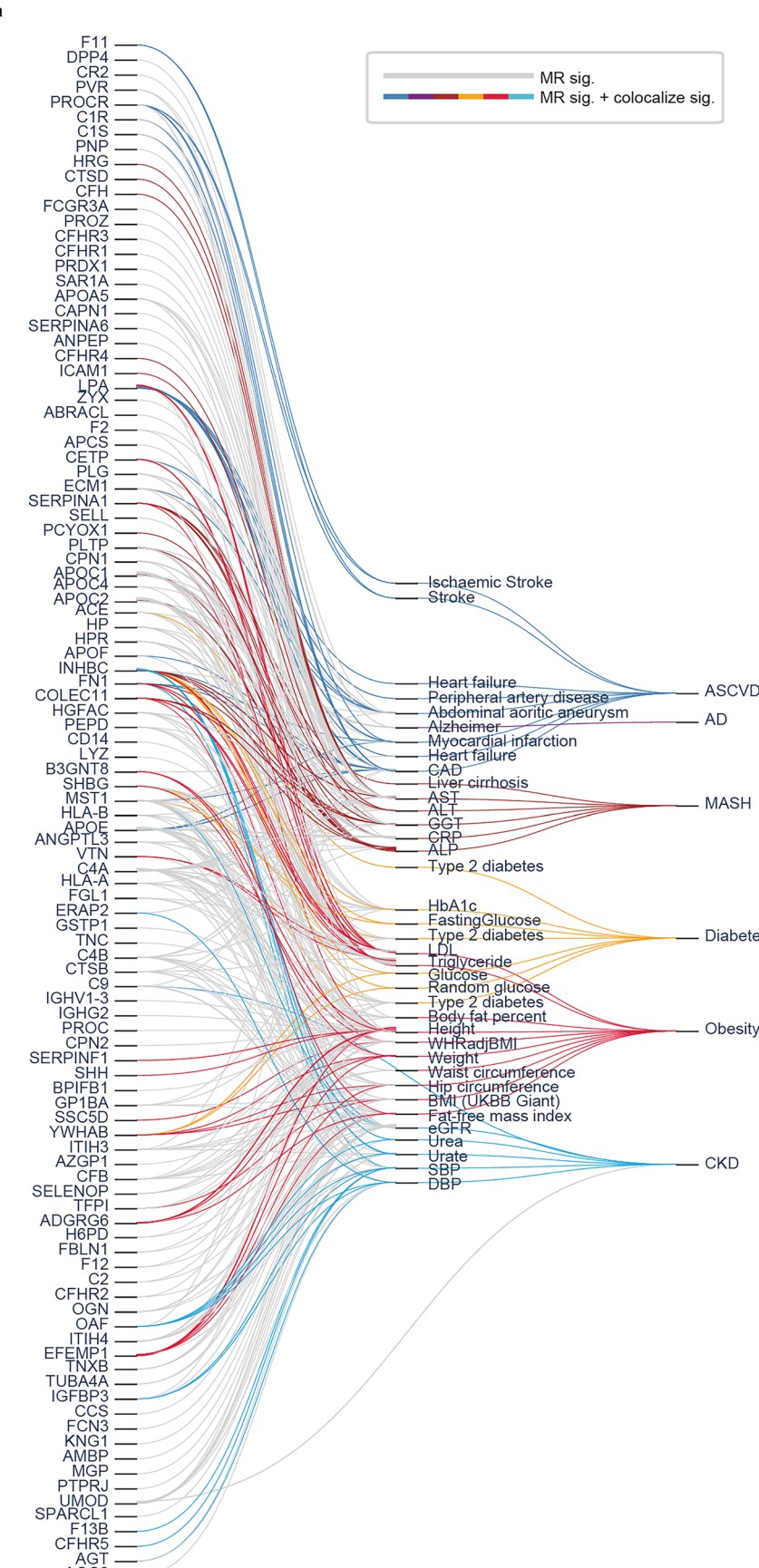

**Extended Data Fig. 7 | See next page for caption.**

**Extended Data Fig. 7 | Mendelian randomization identifies potential causal genes for cardiometabolic traits and diseases. a**, Gene-trait causal relationships for six diseases (two-sided Wald-ratio test implemented in the twoSampleMR package with significance defined as $p < 2.5e^{-6}$ (correcting for the number of protein coding genes). MR: Mendelian randomization; CAD: coronary artery disease; ALT: alanine aminotransferase; AST: aspartate aminotransferase; CRP: C-reactive protein; ALP: Alkaline phosphatase; GGT: gamma glutamyl transferase; HbA1c: hemoglobin A1c; LDL: low-density lipoprotein; eGFR: estimated glomerular filtration rate; SBP: systolic blood pressure; DBP: diastolic blood pressure; BMI: body mass index; WHRadjBMI: waist-to-hip ratio adjusted for body mass index. ASCVD: atherosclerotic cardiovascular disease; MASH: metabolic dysfunction-associated steatohepatitis; AD: Alzheimer's disease; CKD: chronic kidney disease.

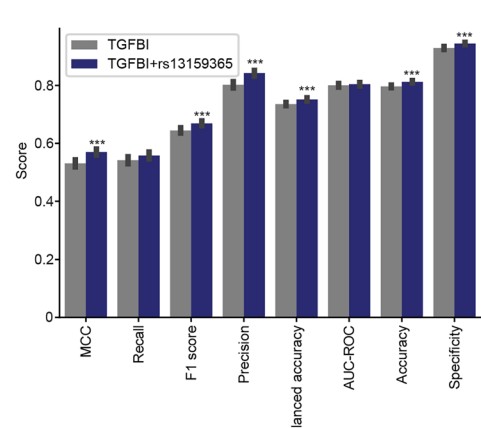

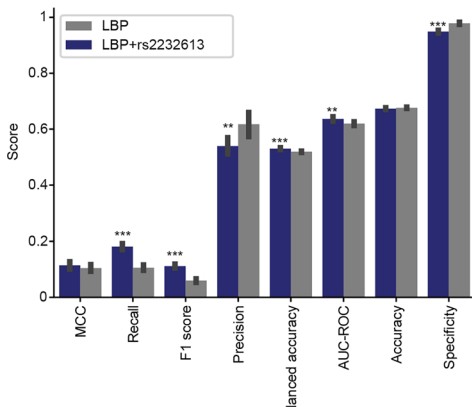

**Extended Data Fig. 8 | Incorporating pQTLs affect biomarker performance.**
**a**, Classification performance metrics of TGFBI and TGFBI+rs13159365 for identifying significant fibrosis in the adult replication cohort. Significance levels from two-sided two-sample independent t-test are indicated (*$p < 0.05$, **$p < 0.01$, ***$p < 0.001$). The exact P-value for each parameter from left to right is as follows: $2.4e^{-6}$, 0.06, $4.1e^{-4}$, $4.4e^{-7}$, $2.8e^{-4}$, 0.43, $2.7e^{-6}$, and $5.4e^{-6}$. **b**, Classification

performance metrics of LBP and LBP+rs2232613 at identifying any steatosis. The exact P-value for each parameter from left to right is as follows: 0.32, $5.4e^{-20}$, $1.1e^{-21}$, 0.006, $3e^{-6}$, 0.002, 0.15, $1.7e^{-17}$. The number of iterations and hence technical replicates is 100 in both Panel (**a**) and (**b**). Data are presented as mean values with 95% confidence interval. MCC: Matthews's correlation coefficient; AUC-ROC: area under the receiver operating characteristic curve.

# Reporting Summary

## Statistics

For all statistical analyses, confirm that the following items are present in the figure legend, table legend, main text, or Methods section.

| n/a | Confirmed | |
|---|---|---|
| ☐ | ☒ | The exact sample size (*n*) for each experimental group/condition, given as a discrete number and unit of measurement |
| ☐ | ☒ | A statement on whether measurements were taken from distinct samples or whether the same sample was measured repeatedly |
| ☐ | ☒ | The statistical test(s) used AND whether they are one- or two-sided<br>*Only common tests should be described solely by name; describe more complex techniques in the Methods section.* |
| ☐ | ☒ | A description of all covariates tested |
| ☐ | ☒ | A description of any assumptions or corrections, such as tests of normality and adjustment for multiple comparisons |
| ☐ | ☒ | A full description of the statistical parameters including central tendency (e.g. means) or other basic estimates (e.g. regression coefficient) AND variation (e.g. standard deviation) or associated estimates of uncertainty (e.g. confidence intervals) |
| ☐ | ☒ | For null hypothesis testing, the test statistic (e.g. *F*, *t*, *r*) with confidence intervals, effect sizes, degrees of freedom and *P* value noted<br>*Give P values as exact values whenever suitable.* |
| ☐ | ☒ | For Bayesian analysis, information on the choice of priors and Markov chain Monte Carlo settings |
| ☒ | ☐ | For hierarchical and complex designs, identification of the appropriate level for tests and full reporting of outcomes |
| ☐ | ☒ | Estimates of effect sizes (e.g. Cohen's *d*, Pearson's *r*), indicating how they were calculated |

*Our web collection on statistics for biologists contains articles on many of the points above.*

## Software and code

Policy information about availability of computer code

| | |
|---|---|
| Data collection | The commercial software Spectronaut (v18.4) was used to perform protein identification and quantification. |
| Data analysis | The software used in this study can be accessed here: EAGLE2 (v2.0.5): https://alkesgroup.broadinstitute.org/Eagle/; Spectronaut (v15.4): https://biognosys.com/software/spectronaut/; Python (v3.9.0 and 3.8.11): https://www.python.org/; Combat (v0.3.2): https://pypi.org/project/combat/; Gseapy (v1.1.1): https://gseapy.readthedocs.io; Seaborn (v0.12.2): https://seaborn.pydata.org/; QLattice implemented in the Feyn Python module (v3.0.1): https://www.abzu.ai/qlattice/; Compute-lambda.py (v2.0): https://github.com/pgxcentre/lambda; Bcftools (v1.14): https://samtools.github.io/bcftools/; Gemma (v0.98.5): https://github.com/genetics-statistics/GEMMA; Plink (v1.90b6.24): https://www.cog-genomics.org/plink/; GCTA-COJO (v1.93.3): https://yanglab.westlake.edu.cn/software/gcta; Variant effect predictor was performed online on March 02, 2024 with the RefSeq transcript database (http://grch37.ensembl.org/Homo_sapiens/Tools/VEP); Scikit-learn (v1.0): https://scikit-learn.org/stable/whats_new/v1.0.html; INT-transformation (v2019.08.21): https://github.com/edm1/rank-based-INT; Scipy (v1.7.1): https://scipy.org/; Pingouin (v0.4.0 and v0.5.4): https://pingouin-stats.org/build/html/index.html; Statsmodels (v0.13.0): https://www.statsmodels.org/stable/index.html; Pyliftover (v0.4): https://pypi.org/project/pyliftover/; GeneLocator (v1.1.2): https://pypi.org/project/GeneLocator/; Hyprcoloc (v1.0.0): https://github.com/cnfoley/hyprcoloc; TwoSampleMR (v0.5.6): https://mrcieu.github.io/TwoSampleMR; The customed scripts can be accessed at github.com/llniu/pQTL_HolbaekStudy. |

For manuscripts utilizing custom algorithms or software that are central to the research but not yet described in published literature, software must be made available to editors and reviewers. We strongly encourage code deposition in a community repository (e.g. GitHub). See the Nature Portfolio guidelines for submitting code & software for further information.

## Data

The GWAS summary statistics generated in this study have been uploaded to the GWAS Catalog (https://www.ebi.ac.uk/gwas/), under accession IDs GCST90452968 to GCST90454170, and will be available upon publication. Accession IDs or download links for publicly available GWAS summary statistics datasets used in this study are listed in Supplementary Note 7. The canonical human reference proteome database (2023.05 release) was downloaded from the European Bioinformatics Institute database (https://ftp.ebi.ac.uk/pub/databases/reference_proteomes/); Tissue specificity annotation of proteins was downloaded from the Human Protein Atlas database (https://www.proteinatlas.org/about/download). The GWAS Catalog (v1.0.2) was downloaded at https://www.ebi.ac.uk/gwas/docs/file-downloads; Transcription start site of proteins was extracted from BioMart (accessed on November 23, 2023) (https://grch37.ensembl.org/info/data/biomart/index.html); The gene sets used for mapping biological processes and functions can be accessed via Gseapy (v1.1.1) using the identifiers 'MSigDB_Hallmark_2020' and 'GO_Biological_Process_2023'. All analysis results are available as supplementary tables. Searchable results are publicly accessible at proteomevariation.org. The study protocol is also available upon request to Jens-Christian Holm, jhom@regionsjaelland.dk. Due to GDPR regulations, individual-level clinical metadata, genomics and proteomics data generated in this study cannot be made publicly available but are available upon request to the corresponding authors. The time frame for response to requests from the authors is within a 1-month period. When processing data, certain restrictions apply: (1) a data processing agreement must be signed between the data controller and processor; (2) data must not be processed for purposes other than statistical and scientific studies; and (3) personal data must be deleted, anonymized and destroyed at the end of investigation and must not be passed on to a third party or individuals who are not authorized to access the data.

## Human research participants

| | |
|---|---|
| Reporting on sex and gender | Information on sex was self-reported and checked using X-chromosome data using the Plink software. Boys and girls were well-balanced in the discovery cohort (55% girls and 45% boys). Sex-based analysis was performed in testing the effects of sex on the plasma proteome. |
| Population characteristics | Discovery cohort: n=2,147; 45% males; ages 5-20, median age of 12; median BMI of 21.7. Distribution of covariates: 65% in pubertal/post-pubertal stage (tanner stage 2-5); sample storage time 1.4-12.4 years (median 7.4). Replication cohort in children: n=1,000 matched by age, sex and overweight status for replication (58% girls and 42% boys). Replication cohort: n=558; 73% males; ages 19-82, median age of 56; median BMI of 27. Distribution of covariates: fibrosis stage F0/1/2/3/4: 249/118/102/26/63, inflammatory activity I0/1/2/3/4/5: 290/90/78/50/28/22, steatosis S0/1/2/3: 373/79/70/36, 35% abstinent upon inclusion, 17% had treatment of statin prior to inclusion. Note that individuals who were not biopsied due to low liver stiffness as measured by FibroScan (<6.0 kPa) were considered as healthy. |
| Recruitment | The participants from the discovery cohort were recruited from 1) the Children's Obesity Clinic, Centre of Obesity Management offering the multidisciplinary childhood obesity management program at Copenhagen University Hospital Holbæk and 2) a population-based cohort recruited from schools in 11 municipalities across Zealand, Denmark in a cross-sectional study design. Both groups were enrolled between January 2009 and April 2019. Eligibility criteria for the children in the obesity clinic group were an age of 5–20 years and a BMI above the 90th percentile (BMI SDS >=1.28) according to Danish reference values. Exclusion criteria for this study for both groups are 1) age at recruitment younger than 5 years or older than 20 years; 2) diagnosed type 1 diabetes; 3) diagnosed type 2 diabetes; 4) treatment with medications including insulin, liraglutide, and/or metformin; 5) meeting type 2 diabetes criteria based on the blood sample taken for this study (fasting plasma glucose > 7.0 mmol/L and/or hemoglobin A1c (HbA1c) > 48 mmol/mol.<br><br>The study cohort consists solely of individuals of European ancestry. This limitation may affect the generalizability of findings to individuals from other ancestry groups, and further studies in more diverse cohorts would be beneficial for broader applicability. The participants' ages follow a normal distribution with a mean of 12, but there is limited representation (n<30) at the younger (age 5) and older (age 20) extremes. This may affect the accuracy of the age-dependent protein abundance trajectories at these age extremes. |
| Ethics oversight | The study protocol for the discovery cohort was approved by the ethics committee for the Region Zealand (protocol no. SJ-104) and is registered at the Danish Data Protection Agency (REG-043-2013). The HOLBAEK Study including the obesity clinic cohort and the population-based cohort are also registered at ClinicalTrials.gov (NCT00928473).<br><br>The study protocol for the GALAXY replication cohorts was approved by the ethics committee for the Region of Southern Denmark (nos. S-20160006G, S-20120071, S-20160021 and S-20170087) and is registered with both the Danish Data Protection Agency (nos. 13/8204, 16/3492 and 18/22692) and Odense Patient Data Exploratory Network (under study identification nos. OP_040 and OP_239 (open.rsyd.dk/OpenProjects/da/openProjectList.jsp)). |

Note that full information on the approval of the study protocol must also be provided in the manuscript.

# Field-specific reporting

Please select the one below that is the best fit for your research. If you are not sure, read the appropriate sections before making your selection.

☒ Life sciences  ☐ Behavioural & social sciences  ☐ Ecological, evolutionary & environmental sciences

For a reference copy of the document with all sections, see nature.com/documents/nr-reporting-summary-flat.pdf

# Life sciences study design

All studies must disclose on these points even when the disclosure is negative.

| Sample size | n=2,147 for the discovery cohort and n=1,000 for replication in children, n=558 for replication in adults. No sample size calculations were done prior to inclusion of the study participants. The sample size was determined partly based on the throughput of the proteomics workflow and the instrument capacity available in the laboratory at the time. The chosen sample size for discovery was within the range of existing pQTL studies, i.e. between 100s and 10000s. Sample size in the replication cohort was limited by data availability (genotype data, proteomics data and covariates). |
|---|---|
| Data exclusions | All samples were kept in the analysis unless relevant variables in the corresponding analysis had missing values. |
| Replication | For plasma proteome profiling in the discovery cohort, 94 pooled plasma samples were analyzed to assess the total technical variability of the proteomics workflow. The study included an additional cohort of 1,000 children and adolescents from the same HOLBAEK Study, as well as an independent adult cohort to replicate the identified pQTLs. Notably, for all biological samples, plasma proteome measurements were performed only once, without additional technical replicates, to capture a single-point measurement per sample. |
| Randomization | Plasma samples were randomized prior to proteomics sample preparation and data acquisition. Sample storage time was used as a covariate to control for potential systematic bias related to sample storage. Batch correction was further applied to correct for batch effects related to proteomics sample preparation. |
| Blinding | The investigators who performed proteomics sample preparation and data acquisition were blinded to the participants' genotype information but not age, BMI and sex as randomization in proteomics data generation was anyway performed to avoid systematic bias during the measurement and thus blinding is not relevant. |

# Reporting for specific materials, systems and methods

We require information from authors about some types of materials, experimental systems and methods used in many studies. Here, indicate whether each material, system or method listed is relevant to your study. If you are not sure if a list item applies to your research, read the appropriate section before selecting a response.

### Materials & experimental systems

| n/a | Involved in the study |
|---|---|
| ☒ | ☐ Antibodies |
| ☒ | ☐ Eukaryotic cell lines |
| ☒ | ☐ Palaeontology and archaeology |
| ☒ | ☐ Animals and other organisms |
| ☐ | ☒ Clinical data |
| ☒ | ☐ Dual use research of concern |

### Methods

| n/a | Involved in the study |
|---|---|
| ☒ | ☐ ChIP-seq |
| ☒ | ☐ Flow cytometry |
| ☒ | ☐ MRI-based neuroimaging |

## Clinical data

Policy information about clinical studies

All manuscripts should comply with the ICMJE guidelines for publication of clinical research and a completed CONSORT checklist must be included with all submissions.

| Clinical trial registration | NCT00928473 |
|---|---|
| Study protocol | Study protocol for the discovery cohort and replication cohort in children is available upon request to Jens-Christian Holm, jhom@regionsjaelland.dk. Study protocol for the replication cohort is available upon request to Odense Patient Data Exploratory Network (open@rsyd.dk) with reference to project ID OP_040. |
| Data collection | Subjects in the discovery cohort were recruited between January 2009 and April 2019 from 1) an obesity clinic cohort recruited from a multidisciplinary childhood obesity management program at Copenhagen University Hospital Holbæk and 2) a population-based cohort recruited from schools in 11 municipalities across Zealand, Denmark in a cross-sectional study design. Demographics, clinical data and blood plasma samples were collected upon recruitment. |

| Outcomes | Not applicable |

