## [Peer Review File · Nature Genetics]

Plasma Proteome Variation and its Genetic Determinants in Children and Adolescents

Corresponding Author: Professor Matthias Mann

Version 0:

Decision Letter:

5th Jul 2023

Dear Professor Mann,

Your Article, "Plasma Proteome Variation and its Genetic Determinants in Children and Adolescents" has now been seen by 2 referees. You will see from their comments copied below that while they find your work of considerable potential interest, they have raised quite substantial concerns that must be addressed. In light of these comments, we cannot accept the manuscript for publication, but would be very interested in considering a revised version that addresses these serious concerns.

We hope you will find the referees' comments useful as you decide how to proceed. If you wish to submit a substantially revised manuscript, please bear in mind that we will be reluctant to approach the referees again in the absence of major revisions.

To guide the scope of the revisions, the editors discuss the referee reports in detail within the team, including with the chief editor, with a view to identifying key priorities that should be addressed in revision and sometimes overruling referee requests that are deemed beyond the scope of the current study. In this case, we would like to encourage you to address Reviewers' comments in full and particularly points 1. and 2. from Reviewer#2. We hope that you will find the prioritised set of referee points to be useful when revising your study. Please do not hesitate to get in touch if you would like to discuss these issues further.

If you choose to revise your manuscript taking into account all reviewer and editor comments, please highlight all changes in the manuscript text file. At this stage we will need you to upload a copy of the manuscript in MS Word .docx or similar editable format.

*2) If you have not done so already please begin to revise your manuscript so that it conforms to our Article format instructions, available here. Refer also to any guidelines provided in this letter.

*3) Include a revised version of any required Reporting Summary: <https://www.nature.com/documents/nr-reporting-summary.pdf>

Please be aware of our guidelines on digital image standards.

Link Redacted

If you wish to submit a suitably revised manuscript we would hope to receive it within 6 months. If you cannot send it within this time, please let us know. We will be happy to consider your revision so long as nothing similar has been accepted for publication at Nature Genetics or published elsewhere. Should your manuscript be substantially delayed without notifying us in advance and your article is eventually published, the received date would be that of the revised, not the original, version.

Nature Genetics is committed to improving transparency in authorship. As part of our efforts in this direction, we are now requesting that all authors identified as 'corresponding author' on published papers create and link their Open Researcher and Contributor Identifier (ORCID) with their account on the Manuscript Tracking System (MTS), prior to acceptance. ORCID helps the scientific community achieve unambiguous attribution of all scholarly contributions. You can create and link your ORCID from the home page of the MTS by clicking on 'Modify my Springer Nature account'. For more information please visit <http://www.springernature.com/orcid>.

Thank you for the opportunity to review your work.

Sincerely,
Chiara

Chiara Anania, PhD
Associate Editor
Nature Genetics
<https://orcid.org/0000-0003-1549-4157>

Referee expertise:

Referee #1:

Referee #2: proteomics, system biology, cancer proteomics

Referee #3: plasma proteomics

Reviewers' Comments:

Reviewer #2:

Remarks to the Author:

Niu et al. presented a potentially interesting study using mass spectrometry (MS) based DIA proteomics in analyzing plasma proteome variation and its genetic determinants in children and young people. A similar scale of human cohorts has been reported previously (Xu et al, Nature Communications 2023, 2,958 Han Chinese participants, Ref 14) using a similar DIA-MS technique for a similar goal of identifying pQTLs in plasma, but not in young people's samples. The technical performance of the present study from the Mann lab seemed to derive a better proteome coverage than Xu et al. (420 proteins vs. 304 proteins). Therefore, while we feel like this is a high-quality dataset, we also think that this study missed a few opportunities to distinguish itself from previous studies (including Xu et al., other MS-based or non-MS based). The current manuscript also had some issues in presenting itself as an integrative analysis (see below) that would otherwise significantly improve our understanding of the plasma proteome variability and its genetic control.

Major:

1. The authors placed emphasis on their unique sample set, specifically children and adolescents, in the title and introduction. However, their analysis does not adequately consider or highlight this uniqueness. The results show the identification of similar pQTLs compared to previous studies. No validation experiments or specific analyses were conducted to establish the unique or significant importance of these findings in children. Moreover, their replication cohort (Table S1) consisted of senior citizens (age=56+-10) rather than children and adolescents. The authors should address this issue by exploring potential connections between the age of participants in previous studies and the current results. Due to the natural aging process, the sample set of young individuals should have better statistical power in detecting genetic control. The authors should provide a more comprehensive presentation and discussion in this regard. Furthermore, they may consider removing the replication cohort demonstration in Figure 1 and replacing it with "analysis from other datasets" to reflect the need for a more appropriate comparison.

2. The authors mentioned that more than half of their samples were obtained from children with obesity. They observed a strong association between inflammatory proteins and BMI SDS. However, this trait was regarded as "non genetic" and largely omitted in the analysis of "genetic effects". Relevant questions are, e.g., whether their investigation on "genetic effects" would be biased towards obesity (even the replication set was obesity enriched)? Whether individual age-dependent (or pre- and post-pubertal stage dependent) plasma markers that would predict the obesity risk during development? Their figure S2 entirely missed this component and opportunities – a similar analysis could be nicely performed for BMI high and BMI regular groups.

3. Non-genetic and genetic effects. This point might be challenging to address using an integrative analysis. But the current manuscript presented and discussed Non-genetic and genetic effects in a mutually distinctive manner. How would the non-genetic and genetic effects jointly shape the plasma protein concentrations and variability? For a specific protein marker for kids, what is the relative importance of non-genetic and genetic effects? Could the genetic effects be used as an additional information piece to improve the linear regression model in Figure 2a and improve the prediction results in Figure 2l? While we understand these analyses are challenging, the authors should strengthen this aspect, minimally through discussions, to improve the coherence of the manuscript.

Minor:

1. Abstract: why the authors commented that the extensive genetic impact on plasma levels is "unexpected"? Also, does this imply that the author would expect less impact on other proteomes, such as tissues? Please revise.
2. Introduction: Could the authors provide additional comments on current protein markers that could vary during the developmental stages of children? Also, discuss the importance of studying obesity in children.
3. Line 83-84: Please add "based on the current dataset and our modeling" to clarify the context. Would the authors expect a significant change, such as the contribution percentage of BMI, if another sample set were used?
4. Line 99: A relevant question is which proteins, if any, appeared to be exclusive to obesity.
5. Line 107-108 and Figure S3: Forty common proteins were included for predicting age and BMI. Is this expected? It would also be helpful to highlight the best predictive proteins (e.g., top 5 or 10) and their power for further studies.
6. Line 116: We found the statement that "Half of these proteins decrease with age" to be risky and potentially related to data normalization or processing. Please respond or revise accordingly.
7. Line 130: It would be useful to indicate that the "genetic effects" studied by the authors are solely related to individual SNPs (e.g., not gene copies or genetic interactions).
8. Line 132: Please add references for how to define a primary pQTL.
9. Line 140: Please add a comment and some references since the finding of non-coding regions seems consistent with previous studies.
10. LINE 172: What if the peptide-level data could support a pQTL, whereas the protein level does not? Since DIA-MS essentially measures peptides, it might be necessary to perform a peptide-level analysis and compare the results to the protein level.
11. Line 200: "of which" - Please revise the text to clarify whether "which" refers to proteins or pQTLs.
12. Figure 5B: It would be helpful to provide a supplementary table for Figure 5B and check if the most common pQTLs among studies have larger effect sizes.
13. Line 222-242: Are most samples in the primary dataset and replication dataset from Denmark? It might be worth providing a remark about this in the Discussion.
14. Methods Line 707: Were the 17 missing values due to a survey failure or the unwillingness to disclose gender?
15. Line 714: We do not understand what "72,792 additional sequences" refers to. Please clarify.
16. Line 760: Were the two covariates (a) sufficient and (b) useful in the model?
17. Line 850: The short description of effective size determination can be moved to the Results section as well.
18. Define what a primary pQTL is and what information is included in the "SNP ID." Effect size vs. protein variability.
19. The authors used the ratio of mean protein abundance between zygote types to calculate the effect size. This might be too simple – How would the author exclude the contribution of the basic protein variability due to the non-genetic effects and technical measurements?

Reviewer #3:

Remarks to the Author:

In "Plasma Proteome Variation and its Genetic Determinants in Children and Adolescents", Niu et al. present the use of mass spectrometry (MS) to determine the levels of circulating proteins and study their associations with genetic variation (pQTLs). This study provides a valuable complementary contribution in a field dominated by affinity-based methods applied to large biobanks. It expands the list of available MS-based pQTL studies in sample size and provides insights into the challenge of assigning peptide-centric pQTLs. The manuscript is well-written, concise and does not present overly controversial findings. The established website is very helpful when browsing the data.

Especially for a non-proteomics audience, the authors should clarify the use of proteins versus peptides in their analysis. How was the peptide information used to determine protein levels? Do all peptides detected per protein provide concordant and interchangeable information? Did the number of peptides detected per protein influence the assignment of pQTLs? Could a better pQTLs coverage derive from a higher analytical coverage of proteins (= more data points per protein)?

The authors mention that specificity is a crucial advantage of MS over affinity proteomics. Considering the above, this is

effective via the number of peptides detected per protein. The authors should, however, further clarify how genetic variation affects the different peptides detected from a protein. Ideally, corresponding examples for the most dominant annotations (introns, intergenic and regulatory regions) should be presented.

In the provided web interface, the top-ranked pQTL is rs9898 for HRG. Interestingly, a recent study presented the challenge when detecting HRG (DOI: 10.1016/j.mcpro.2023.100585) with its non-synonymous substitutions (highlighting rs9898). HRG is an abundant and glycosylated multi-domain protein. As mentioned by the authors, the chosen MS approach detects specific chains or subunits of a protein rather than the protein in its complete shape (and together with its interaction partners). The authors should therefore take this opportunity to showcase these advantages, and maybe HRG can serve as a relevant example.

The authors correctly applied a stringent selection scheme for the downstream analysis instead of artificially inflating the reported number of proteins. However, many of the remaining proteins are abundant in the circulation. Hence they will be easier to detect and likely offer more feasible targets to obtain a large number of unique peptides that support a higher precision. The statement that MS offers better precision for all circulating proteins cannot be answered without a direct head-to-head comparison. Growing evidence shows that low-abundant targets are more difficult to detect reliably due to sensitivity issues, fewer detected peptides, and a higher frequency of missing values. Please discuss and refine the statements considering their effects on the pQTL analysis.

For this reviewer, it remained unclear if there is a relationship between age-dependent markers and pQTLs. Given the biology of adolescence, did the authors find any specific associations in younger participants that they could not replicate in the older cohort? Did the authors consider using their pQTLs as instruments in Mendelian randomization (MR) - e.g. for childhood obesity?

The authors present a few proteins to validate their observations. However, as shown for the protein IGFALS, could these be due to lifestyle, nutrition, and activity at a young age rather than age itself? Please conduct interaction analyses between age, sex, and BMI to understand their dependencies in adolescence better. The authors present CD5L as a sex-associated protein. However, longstanding evidence exists that this protein is a surrogate of IgM (top hit). Do the authors have additional evidence for ongoing infections that could assist in explaining their observations (e.g., AST, CRP, clinical records)? In addition, A2M was found as a sex and BMI-associated protein. Please explain this dual assignment and maybe stratify the discovery cohort in Table 1 by sex (+ add p-values of comparative analysis).

The clustering analysis is interesting. Please add the clinical traits from Table 1 to this analysis. This might assist in explaining some of the biology these clusters represent. Did the authors see any relationship between these biologies and the frequency of reported pQTLs?

It would be helpful to dissect the analysis into the organ of origin to add some more physiology to the manuscript. Since many abundant and detected proteins derive from the liver by secretion, it would be very interesting to learn what the other (non-liver) proteins are and which biologies they can represent regarding genetic variance, age, sex, and BMI.

The study is likely underpowered to determine many trans-pQTLs. However, considering the observed trans-pQTLs, did the authors find any "usual suspects", such as the ABO locus, or did they identify other trans-pQTLs? Could these be MS-specific and linked to enzymatic cleavage (see KLKB1)? Please consider this recent preprint during the revision (DOI: 10.1101/2023.04.20.537640v1). Lastly, the authors should extract some biology from the novel, top-ranked, or most interconnected cis- and trans-pQTLs.

Other comments:

Line 242 states "case-by-case basis". Does this imply that there is no systematic approach to solving such cases?

Please provide QQ-plots for all association analyses as supplementary information, and include all variables from the discovery cohort (e.g., ALT, AST, insulin, cholesterol, ...)

The online tool is very helpful. Please explain the meaning of the "lower" volcano in the BMI plot. Please reorder the pQTL table to move the p-value and MAF columns further to the left for ranking simplicity. It would be helpful if this pQTL table could be searchable for features of interest.

Sex is listed among the non-genetic effects. Please clarify, and explain the rationale behind removing the sex-specific chromosomes from the analysis.

Version 1:

Decision Letter:

18th Jul 2024

Dear Professor Mann,

Your Article, "Plasma Proteome Variation and its Genetic Determinants in Children and Adolescents" has now been seen by 2 referees. You will see from their comments below that while they appreciate how the paper improved in revision and find your work of interest, some important points on the new analysis are raised. We are interested in the possibility of publishing your study in Nature Genetics, but would like to consider your response to these concerns in the form of a revised manuscript before we make a final decision on publication.

To guide the scope of the revisions, the editors discuss the referee reports in detail within the team, including with the chief editor, with a view to identifying key priorities that should be addressed in revision and sometimes overruling referee requests that are deemed beyond the scope of the current study. In this case, we ask you to address Reviewers' comments in full. Please do not hesitate to get in touch if you would like to discuss these issues further.

We therefore invite you to revise your manuscript taking into account all reviewer and editor comments. Please highlight all changes in the manuscript text file. At this stage we will need you to upload a copy of the manuscript in MS Word .docx or similar editable format.

*2) If you have not done so already please begin to revise your manuscript so that it conforms to our Article format instructions, available

[here](http://www.nature.com/ng/authors/article_types/index.html).

*3) Include a revised version of any required Reporting Summary: <https://www.nature.com/documents/nr-reporting-summary.pdf>

Link Redacted

We hope to receive your revised manuscript within four to eight weeks. If you cannot send it within this time, please let us know.

Sincerely,
Chiara

Chiara Anania, PhD
Associate Editor
Nature Genetics
<https://orcid.org/0000-0003-1549-4157>

Referee expertise:

Referee #1:

Referee #2:

Referee #3:

Reviewers' Comments:

Reviewer #2:

Remarks to the Author:

In the revised manuscript, the authors provided a complete remeasurement of the entire cohort and additionally included substantial replication samples using the latest mass spectrometry platform, despite previous positive reviewer comments. This effort perhaps explains why the Mann lab has been one of the major leading labs driving mass spectrometry application for many years and should be highly applauded. The revised study is at the forefront, comprehensive, and inspiring in many aspects. Its significance includes: a) the best large-scale plasma proteomic dataset of its kind so far; b) the first application to a large number of children; c) a comprehensive resource of pQTLs (both protein and peptide level, which is not possible via affinity-based approaches). It should definitely be published in Nature Genetics in our opinion. Before it is accepted, however, due to the new dataset and profound new analysis presented in the revision, we have the following suggestions and questions.

Major:

1. The tissue origin and specificity of the plasma proteins.

Given that >1000 proteins were identified in the neat plasma, the authors sought to perform some annotation to understand the tissue derivation of these proteins. They only annotated to an mRNA dataset, not a proteomic dataset, using ambiguous descriptive items such as "tissue enhanced" or "group enriched" etc. We don't think this analysis (Figure 2a & Figure 3i) is necessary or precise. Notably, this is a very challenging scientific topic that requires additional analysis of tissue proteomes (see Malmström et al., 2016). Also, although >1000 proteins were analyzed, this is certainly not the largest proteomic dataset in plasma so far. The fact that the brain was ranked 4th among all tissues is very suspicious due to the existence of the BBB and is risky to publish. This reviewer therefore strongly suggests that the authors remove this analysis, as it is not very relevant to the study's topic, either. Otherwise, they should perform this analysis more properly, addressing all the above points, and adding relevant discussion.

2. Figure 4 b-d and pQTLs across age groups.

We are still struggling to understand how aging should impact calling pQTLs (re. Figure 4b-d). Suppose there are contributions from environmental factors, the aging process, and emerging diseases, the chance of detecting genetic control via pQTLs should be affected by aging to some extent? Would it be possible to compare the absolute portion % rather than the correlation? Relating to this, some analysis and discussion around aging and pQTLs or event eQTLs would be beneficial since this study focuses on children, as compared to previous studies.

Minor:

1. Line 152: Mention how data is normalized in the result texts (plasma volume or total MS signals).

2. Supplementary Figure 2b-q: Add more legend text to explain the variables and numbers of boys and girls and yes and no.

3. Line 153: What controversies? How are they resolved?

4. Line 268: Explain "a held out set of 639 individuals."

5. Line 337: Explain "(5x10⁻⁸/1,216 proteins)," since the same p-value was used to infer 443 proteins in the same paragraph.

6. Line 358: The argument of Pro204Ser is weak unless it is really detected. Right now, is it not detected because of database searching?

7. Line 362: 62% of "discovered pQTLs"? Since a paragraph was inserted above now.

8. Figure 3i: Again, it makes no sense to conclude that pQTLs largely mirror tissue origination or "enhanced".

9. Figure 4a: A supplementary table of Figure 4a would be nice, as the font is too small for the x-axis.

10. Supplementary Figure 7a: Highlight the authors' study here.

11. Line 856-861: The peptide analysis for excluding artefactual pQTLs seems to be an important note to the field and the authors even mentioned that in the abstract. We, therefore, think this result and "Reviewer Fig 2" should be included as a supplementary figure (at least) and presented as part of the Results, unless there is a reason not to.

12. Line 92: We suggest adding "due to the limitation of MS sensitivity and other factors" after the authors' comment on "low numbers of proteins" to give more credit to previous studies. Obviously, after 10 years, ">1000 plasma proteins by MS" will not be as impressive as it is now, and Astral will be outdated.

Reviewer #3:

Remarks to the Author:

Niu et al. have revised their manuscript: "Plasma Proteome Variation and its Genetic Determinants in Children and Adolescents Plasma Proteome Variation and its Genetic Determinants in Children and Adolescents" and responded to all

my questions and concerns. The study complements the ongoing efforts to map pQTLs of circulating proteins.

Beyond my previous comments, the authors used a novel MS instrument to generate new data reporting 1200 instead of 400 proteins (in all samples). Moreover, they added a new cohort of adults compared to the ones described in the first submission. Adding new data with the newest instruments is, of course, very good. However, please take this opportunity to compare data from overlapping proteins between the previous and new data. This helps the readers understand if other instruments will deliver concordant data for the overlapping targets. It may also offer technical validation and understanding if a later (delayed) analysis provides the same protein levels.

One observation was decreased p-values for BMI-SDS analysis (Fig 2f). Is this due to measuring other proteins?

Since the time of submission, other related work has been published. To provide a complete picture, please discuss and reference the two articles:

<https://doi.org/10.1038/s41467-024-45233-y> as a recent example of MS-derived pQTLs, covering some aspects discussed here, including the novel pQTLs.

2) <https://www.biorxiv.org/content/10.1101/2024.05.27.596028v1> as a recent example that directly compares MS and affinity-based pQTLs.

Reference # 11 is the preprint of # 61

Version 2:

Decision Letter:

Our ref: NG-A62522R1

20th Sep 2024

Dear Dr. Mann,

Thank you for submitting your revised manuscript "Plasma Proteome Variation and its Genetic Determinants in Children and Adolescents" (NG-A62522R1). It has now been seen by the original referees and their comments are below. The reviewers find that the paper has improved in revision, and therefore we'll be happy in principle to publish it in Nature Genetics, pending minor revisions to satisfy the referees' final requests and to comply with our editorial and formatting guidelines.

Congratulations!

Sincerely,
Chiara

Chiara Anania, PhD
Associate Editor
Nature Genetics
<https://orcid.org/0000-0003-1549-4157>

Reviewer #2 (Remarks to the Author):

The manuscript is acceptable now.
The abstract has a tiny typo "resulting in a q dataset...".

Reviewer #3 (Remarks to the Author):

As final minor remarks (or preferences), please

1) add the comparison of the older and newer data to the supplementary figure, and leave a comment in the text/discussion. Also, add a density distribution of the correlation of random protein pairs to show the intrinsic correlation structure.

2) add a correlation comparison of the 96 SAMPLE profiles (in addition to the one shown for proteins). Here, add a density distribution of correlation of random sample pairs correlations to show the intrinsic correlation structure.

3) provide the numerical outcomes of protein analyses in the supplementary data file (ST15).

Otherwise, the authors have substantially improved the manuscript and addressed all my comments or questions, so there is no need for another round of review from my side.

Jochen Schwenk

Point-by-point answers to reviewers' comments

We thank the reviewers for their thorough evaluation and valuable feedback on our manuscript “**Plasma Proteome Variation and its Genetic Determinants in Children and Adolescents**”. The editor’s suggestions for revision, including the importance of points 1. and 2. from Reviewer 2 have also been instrumental in guiding our revisions.

Executive Summary of the revision

Reviewers 2 and 3 were both generally very positive regarding our work, finding it of considerable potential interest as the editor mentions. At the same time, they raised important concerns:

Reviewer 2 raised concerns about 1) the marginal increase in proteome depth compared to the recent study by Xu et al. (2023), 2) the absence of a replication cohort with a similar demographic profile, 3) the lack of integrative analysis of the plasma proteome variations influenced by age, sex, BMI-SDS, and genetics, and 4) the scarcity of unique and significant findings in children and childhood obesity.

Reviewer 3 recommended 1) investigating technical factors in MS-based proteomics that affect pQTL detection, 2) incorporating peptide-level data in pQTL mapping, 3) enriching the physiological interpretation of findings, and 4) illustrating the advantages of MS-based proteomics over affinity-based methods for detecting pQTLs.

(Please note there was no Reviewer 1)

We have carefully addressed these major points, which involved a complete remeasurement of the entire existing cohort with novel instrumentation, not available upon first submission and adding a substantial cohort as well. This and the exhaustive new analyses are the reason why the revision took some time, which we are grateful to the editor for allowing. As the response is quite long and detailed, we here summarize the main points:

1. Enhanced Dataset and Technological Advances: In response to concerns about dataset depth, we now employed the newly released Orbitrap Astral Mass Spectrometer, the top end instrument by Thermo Fisher Scientific, which we had only received after first submission. With this new technology, we nearly tripled the proteome depth in neat plasma from 420 to 1,216 proteins, the number of primary protein-SNP associations from 1,116 to 1,947 and the number of proteins with pQTLs from 200 to 443 at genome-wide significance ($5e-8$). Compared to Xu et al (2023), this new dataset has a three-fold increase in terms of the number of total proteins profiled (1,216 vs. 304) and four-fold increase in the number of proteins identified with genetic associations (327 vs. 67) at a study-wide significance ($4e-11$). To our knowledge, this updated dataset is the most comprehensive MS-based neat plasma¹ proteomics dataset to date. It clearly provides deeper insights into the plasma proteome's complexity during pediatric development.

2. Replication Cohort and Methodological Enhancements: We included an additional 1,000 children and adolescents as a replication cohort alongside the previous validation cohort consisting of adults. We clarified our methodological choices, adjusting for BMI-SDS and obesity in our GWAS to minimize confounding effects, enhancing the demographic robustness of our findings. Our conditional and joint association analysis helped identify candidate causal SNPs that regulate plasma protein levels. Details are in our response to Reviewer 2, Major Point 1.3-1.4, Major Point 2, and Minor Point 8.

¹ plasma without depletion of highly abundant proteins or enrichment of lowly abundant proteins

3. Decomposition of variance in the pediatric plasma proteome: As requested by reviewer 2, for proteins with genetic associations, we now estimated the contributions of independent pQTLs, age, sex, BMI-SDS, and obesity to protein level variance. This analysis revealed that while genetic factors significantly impact the levels of most proteins, age, sex, and obesity are primary determinants in certain contexts. Further, by dividing the study cohort into three age groups, we observed consistent pQTL effects across pediatric development, highlighting the stable influence of genetics over these critical developmental stages. Details are in our response to Reviewer 2, Major Point 1.1.

4. Genome-wide Association Analysis on peptide level data and Methodological Improvements: Inspired by both reviewers, we performed a genome-wide association analysis on approximately 10,000 peptides specific to protein groups (see answers for details). We refined our methodology to further distinguish genuine pQTLs from artefactual ones, categorizing genuine pQTLs into tiers of confidence based on peptide-level evidence. As requested by reviewer 3, we also delved into the impact of technical factors and protein attributes, such as quantification precision and protein abundance, on detecting pQTLs. Note that our novel framework for using peptide-level data in pQTL mapping can be readily adopted by all subsequent MS-based studies, which we believe to be a significant contribution to the field. Details are in our response to Reviewer 2, Minor Point 10 and Reviewer 3, Major Point 1 and 3.

5. Physiological Relevance and Tissue Specificity: In this revision, we have enhanced the physiological relevance of our findings by detailing the tissue specificity and biological processes of the identified proteins, focusing on those that are associated with obesity and genetic variants and whose expression is specifically enriched in organs like the pancreas, kidney, skeletal muscle, and brain. Moreover, we have elaborated on known and novel proteomic changes related to IGF signaling, skeletal development and angiogenesis during pediatric growth, as well as sex-dependent trajectories and novel protein associations with childhood obesity. Details are in our response to Reviewer 2, Major Point 1.1 and Reviewer 3 Major Point 6.

6. Integrating pQTLs with known variant-trait associations using colocalization and Mendelian Randomization: This excellent suggestion revealed the involvement of >40 proteins in obesity, type 2 diabetes (T2D), metabolic dysfunction-associated steatohepatitis (MASH), atherosclerotic cardiovascular disease (ASCVD), chronic kidney disease (CKD), Alzheimer's disease (AD) and bone mineral density. Specifically, MR on cis-pQTLs resulted in 345 causal relationships between 106 proteins and 36 traits related to the mentioned diseases ($P_{MR} < 2.3e-6$). Among these 345 protein-trait pairs, 101 (29%) between 41 genes and 33 traits were further supported by colocalization evidence. Colocalization analysis on selected novel pQTLs identified proteins colocalizing with NAFLD, T2D and bone mineral density, suggesting the involvement of these proteins in disease development.

We hope that you agree that this comprehensive revision addresses the initial limitations, enhances our understanding of pediatric plasma proteome trajectories crucial for pediatric development, identifies novel disease-causing proteins through MR and colocalization analysis, and sets a new standard for pQTL discovery using MS-based proteomics.

To guide the revision, the text in red represents our response to the comment, while the text in blue indicates the changes we have made in the manuscript.

Reviewers' Comments:

Reviewer #2:

Remarks to the Author:

Niu et al. presented a potentially interesting study using mass spectrometry (MS) based DIA proteomics in analyzing plasma proteome variation and its genetic determinants in children and young people. A similar scale of human cohorts has been reported previously (Xu et al, Nature Communications 2023, 2,958 Han Chinese participants, Ref 14) using a similar DIA-MS technique for a similar goal of identifying pQTLs in plasma, but not in young people's samples. The technical performance of the present study from the Mann lab seemed to derive a better proteome coverage than Xu et al. (420 proteins vs. 304 proteins). Therefore, while we feel like this is a high-quality dataset, we also think that this study missed a few opportunities to distinguish itself from previous studies (including Xu et al., other MS-based or non-MS based). The current manuscript also had some issues in presenting itself as an integrative analysis (see below) that would otherwise significantly improve our understanding of the plasma proteome variability and its genetic control.

Authors' response: We thank the reviewer for their valuable feedback on our manuscript. We are happy to report that we have been able to address the concerns and suggestions of the reviewer as already outlined in our high-level response to the primary concerns regarding our study's novelty in comparison to the work of Xu et al. and our new approach to integrative analysis. The detailed amendments and findings are outlined in the subsequent sections.

Major points:

1.1 The authors placed emphasis on their unique sample set, specifically children and adolescents, in the title and introduction. However, their analysis does not adequately consider or highlight this uniqueness.

We agree and we have considerably expanded the description of our findings unique to pediatric populations. These include known and novel proteomic changes related to IGF signaling, skeletal development and angiogenesis during pediatric growth, as well as sex-dependent trajectories and novel proteins associated with childhood obesity. In addition, we now estimated the proportion of variance in plasma protein levels during pediatric development explained by independent pQTLs, age, sex, BMI-SDS and obesity, revealing distinct influences on the plasma proteome across these factors (**Fig. 4** of the revised manuscript).

Changes to the manuscript are as follows:

“At the pathway level, our dataset reveals significant insights into three critical biological processes during pediatric development. Firstly, in the insulin-like growth factor (IGF) signaling pathway, we observed an age-related increase in IGF receptor signaling, with IGF1 levels peaking at ages 12-13 in girls and 14-15 in boys, then declining into adulthood (**Supplementary Fig. 2b**), consistent with literature [PMID: 21921983]. IGFBP5 mirrors the trend of IGF1, whereas IGFBP1 and IGFBP2 levels declined from childhood through adolescence (**Supplementary Fig. 2c-e**). They were consistently reduced in children with obesity, mirroring adiponectin trends (**Supplementary Fig. 2f**), substantiating the involvement of IGFBP1 and IGFBP2 in childhood obesity. Similarly, SHBG, A2M and CRP all showed obesity-dependent levels (**Supplementary Fig. 2g-i**). Secondly, puberty is a period of rapid change in skeletal development, followed by the cessation of longitudinal growth. In skeletal development, our data capture declines after onset of puberty in essential bone development proteins such as ACAN, COL1A1, COL1A2, THBS4,

COMP, and POSTN, reflecting sex-specific growth plate closure differences (**Supplementary Fig. 2j-m**). Aggrecan (ACAN) is a major proteoglycan in cartilage, mutation of which causes early growth cessation, resulting in a severely reduced adult stature [PMID: 28804204]. Our data documents a more than ten-fold decline in ACAN levels starting at age 12-13 for girls and 13-14 for boys, providing new insights into early diagnostics for growth disorders. Thirdly, proteins involved in angiogenesis and cellular adhesion, such as ANGPTL3, CDH5, ITGB1, ICAM1, VCAM1, and ACE, showed an age-related decline (**Supplementary Fig. 2n-q**). We found that levels of ANGPTL3, a potent regulator of blood triglyceride levels, are inversely correlated with BMI-SDS (**Supplementary Table 2**), helping to resolve previous controversies [PMID: 34422408; PMID: 29695708]. We also discovered novel proteins associated with childhood obesity, including SHBG, A2M, PON3, ADAMTSL4, HSPG2, and MAGEB6B, all of which were decreased in children with obesity.”

“To understand the relative contribution of variance in plasma protein levels by genetic variation and other biological factors during pediatric development, we performed variance decomposition analysis (**Methods**). This revealed that independent pQTLs accounted for 1% to 66% of variance in protein levels (average 11%) (**Fig. 4a**). Remarkably, for 63% of the proteins, the pQTLs were a more significant contributor than age, sex, BMI-SDS and obesity combined. For some proteins, however, other factors than pQTLs contribute most to the overall variance. For example, SHBG levels are primarily affected by age and obesity, PRG4 levels by obesity and IGF1 and RBP4 level by age. To address the important question of how stable the genetic influences are across pediatric development, we segmented the cohort into 5-10, 10-15, and 15-20 year olds. This revealed a remarkable stability in genetic influences, with Pearson correlations between 0.95 to 0.98 (**Fig. 4b-d**). Our results establish that MS-based proteomics can provide unique insights into factors determining protein level variance throughout pediatric development.”

New Manuscript Fig. 4. Variance in plasma protein levels explained by various factors. (a) Proportion

of variance explained by conditionally independent pQTLs, age, sex, obesity, and BMI-SDS (summed variance from BMI-SDS and its interaction with obesity status). Proteins are ordered by decreasing variance attributable to independent pQTLs. **(b-d)** Pairwise comparisons of variance explained by independent pQTLs across three age groups: 5-10, 10-15, and 15-20 years, including the Pearson correlation coefficient.

1.2 The results show the identification of similar pQTLs compared to previous studies. No validation experiments or specific analyses were conducted to establish the unique or significant importance of these findings in children.

In the revised manuscript, we have specifically highlighted the novel pQTLs. Further, we have integrated pQTLs with known variant-trait associations using Mendelian Randomization and colocalization analysis (see also answer just above for unique findings in children).

In our new data a third of the identified pQTLs are novel compared to existing large-scale pQTL studies including the Sun et al., 2023 study in *Nature* (PMID 37794186). We highlight previously undetected associations for proteins like PRG4, SPTBN4, and HTR1D, showcasing the capability of MS-based proteomics to uncover new plasma pQTLs even in a much smaller sample size. This attests to the specificity and quantitative accuracy of MS-based proteomics, especially of our refined workflow.

Furthermore, we performed two-sample Mendelian Randomization (MR) and colocalization to integrate pQTLs with known variant-trait associations.

“In addition, we performed a systematic two-sample Mendelian Randomization (MR) using top cis-pQTLs from 267 proteins on 47 cardiometabolic GWAS outcomes spanning obesity, diabetes, atherosclerotic cardiovascular disease, metabolic dysfunction-associated steatohepatitis, Alzheimer’s disease, and chronic kidney disease therapeutic areas. This analysis reported 345 causal relationships between 106 proteins and 36 traits related to six of the most highly prevalent diseases ($P_{MR} < 2.3e-6$) (**Supplementary Fig. 8** and **Supplementary Table 10**). Of these 345 protein-trait pairs, 101 (29%) between 41 genes and 33 traits were further supported by colocalization evidence, defined as a posterior probability above 70% (**Methods, Fig. 5g-h** and **Supplementary Table 11**). Interestingly, our data causally connect sonic hedgehog (SHH) to height, which may be connected to its crucial roles in embryonic development.”

New Manuscript Supplementary Fig. 8. Mendelian Randomization identifies potential causal genes for cardiometabolic traits and diseases. a, gene-trait causal relationships ($P_{MR} < 2.3e-6$) for six diseases.

Additionally, we performed colocalization on selected novel pQTLs with mapped GWAS traits and identified proteins whose genetic association signal colocalize with those of NAFLD, T2D and bone

mineral density. Our results suggest the involvement of these proteins in disease development, providing hypotheses for causal testing in drug target and biomarker discovery.

The following paragraph replaces the corresponding text in the original manuscript.

“Individuals having the bone mineral density-lowering allele at rs13469-T within the gene *POLDIP2* had lower circulating levels of SPP2, a bone matrix protein regulating bone metabolism and an attractive therapeutic target for bone healing (PMID: 25339413). Similarly, children having the T2D risk increasing allele rs529565-C within the *ABO* locus have higher levels of MASP1 in our data, in line with MASP1 levels being significantly higher among patients with T2D compared with healthy controls (PMID: 28318015; PMID: 27344311). Colocalization analysis was consistent with all the examples above (Supplementary Table 11). “

In addition, we highlighted novel trans-pQTLs within the *APOE* gene for brain-expressed protein SPTBN4. Although we did not find colocalization evidence for SPTBN4 and AD, the involvement of SPTBN4 in AD is supported by existing literature and published omics datasets. In the revised manuscript we take advantage of the increased plasma proteome depth leading to the following new results:

“The levels of 24 proteins were associated with missense variants in a different gene, 11 of these 24 proteins with variants in *APOE*, which has been extensively studied in the context of Alzheimer's disease (AD) and cardiovascular disease. Our data reveal novel protein associations with *APOE* variants. In particular, lower plasma levels of SPTBN4 were associated with the rs7412-C and the rs429358-C alleles, a combination known as the *APOE*- ϵ 4 allele, which strongly predisposes to Alzheimer's disease. SPTBN4 is expressed in the brain, pathogenic variants in SPTBN4 cause neurological disorders including intellectual disability and developmental delay (PMID: 33772159; PMID: 32672909). Given that both proteins have strong connections to neurodegenerative disorders, we hypothesized that SPTBN4 is downregulated in patients with AD. Indeed, both SPTBN4 downregulation and epigenetic silencing has been reported (PMID: 31207390, 24030951).

Furthermore, the *APOE* ϵ 4 allele is significantly associated with depression (PMID: 30834074) and we found the rs7412-C allele associated with higher protein levels of HTR1D in the plasma. This protein is a target of FDA approved drugs for depressive and anxiety disorders, indicating that the association between *APOE* ϵ 4 allele and increased depression risk may be through elevated HTR1D levels. These observations offer new hypotheses for understanding disease mechanisms and potentially new therapeutic strategies.”

1.3 Moreover, their replication cohort (Table S1) consisted of senior citizens (age=56+-10) rather than children and adolescents. The authors should address this issue by exploring potential connections between the age of participants in previous studies and the current results. Due to the natural aging process, the sample set of young individuals should have better statistical power in detecting genetic control. The authors should provide a more comprehensive presentation and discussion in this regard.

Our study is the first large-scale pQTL study in the pediatric populations, underscoring both the novel nature of our work and the lack of comparable data for this demographic. Based on the reviewer's comments we have now obtained samples from an additional 1,000 children and adolescents which we measured for this revision (Fig. 6a of the revised manuscript and Reviewer Fig. 1 below). After quality control on the proteomics and genomics data independently for the discovery and replication sets, approximately 90% of the primary pQTLs could be tested in this children's replication set. Of these, we successfully replicated 97% with nominal significance ($p < 0.05$, and 76% after applying a Bonferroni corrected p-value of $2.9e-5$), showing highly concordant effect sizes and a Pearson correlation coefficient of 0.98 between the beta estimates of the discovery and replication sets. We are delighted by this excellent validation and thank the

reviewer for spurring us to do this. The new results are incorporated in the revised manuscript (see figure below which is **Fig. 6a** and **Supplementary Fig. 9a** in the revised manuscript).

The idea of comparing the effectiveness of pQTL detection between children and adults is indeed fascinating. Our results clearly indicate that the large majority of pQTLs are shared between children and adults. To pick up potential pQTLs that are specific to younger participants, would likely require large datasets of comparable size and the same medical condition or health status, which is not the case in our cohorts. Given the current lack of directly comparable datasets, we have decided against making this comparison to avoid drawing inaccurate conclusions. We acknowledge this limitation in our discussion and emphasize it as an important avenue for future research.

Reviewer Fig. 1, Replication of pQTLs in 1,000 children and adolescents. a, Miami plot showing genetic signals for all pQTLs in the discovery cohort and those replicated. **b,** Correlation of beta coefficient for replicated pQTLs in the children replication cohort with the Pearson correlation coefficient indicated.

1.4 Furthermore, they may consider removing the replication cohort demonstration in Figure 1 and replacing it with "analysis from other datasets" to reflect the need for a more appropriate comparison.

Since we now do have an appropriate replication cohort for the children, we would like to keep the adult replication cohort, too.

2. The authors mentioned that more than half of their samples were obtained from children with obesity. They observed a strong association between inflammatory proteins and BMI SDS. However, this trait was regarded as "non genetic" and largely omitted in the analysis of "genetic effects". Relevant questions are, e.g., whether their investigation on "genetic effects" would be biased towards obesity (even the replication set was obesity enriched)? Whether individual age-dependent (or pre- and post-pubertal stage dependent) plasma markers that would predict the obesity risk during development? Their figure S2 entirely missed this component and opportunities – a similar analysis could be nicely performed for BMI high and BMI regular groups.

Thank you for these comments. In the revised manuscript we further adjusted for obesity status as a binary variable in addition to BMI-SDS in the proteomics data standardization pipeline prior to the genome-wide association analysis. We also present protein abundance trajectories differentiated by sex and obesity status in **Fig. S2** of the revised manuscript.

In more detail, our decision to adjust for BMI-SDS as a covariate by regressing out its effects on protein levels before conducting genome wide association analysis was integral to our study's methodology. This approach was purposely chosen to control the potential confounding effects that obesity-associated genetic variants might exert on protein levels. By doing so, our analysis aimed to minimize bias towards obesity, ensuring a more accurate exploration of the genetic effects on protein levels irrespective of the high prevalence of obesity within our sample population. In this revision, in addition to BMI-SDS, we further included binary obesity status as a covariate.

We recognize the desirability of investigating the potential of age-dependent plasma markers in predicting obesity risk during development. However, such an analysis requires a prospective study design, encompassing plasma protein measurements at an early life stage followed by the development of obesity as an outcome measure at a later stage. Our baseline plasma proteomics data collection anticipates this need, with ongoing long-term follow-up to provide the requisite temporal dimension. However, the cross-sectional design of our current study inherently limits our capacity to predict the risk of developing obesity.

3. Non-genetic and genetic effects. This point might be challenging to address using an integrative analysis. But the current manuscript presented and discussed Non-genetic and genetic effects in a mutually distinctive manner. How would the non-genetic and genetic effects jointly shape the plasma protein concentrations and variability? For a specific protein marker for kids, what is the relative importance of non-genetic and genetic effects? Could the genetic effects be used as an additional information piece to improve the linear regression model in Figure 2a and improve the prediction results in Figure 2I? While we understand these analyses are challenging, the authors should strengthen this aspect, minimally through discussions, to improve the coherence of the manuscript.

Thank you for your excellent suggestion. We have estimated the proportion of variance explained by various factors which improve our understanding of the variation of plasma protein levels in pediatric population. Please refer to our response to your Major Point 1 and see the new Fig. 4 which is also reproduced there.

Minor points:

1. Abstract: why the authors commented that the extensive genetic impact on plasma levels is "unexpected"? Also, does this imply that the author would expect less impact on other proteomes, such as tissues? Please revise.

Thank you for the chance to clarify. Landmark pQTL studies including those by Sun et al., 2018, Ferkingstad et al., 2021 and Sun et al., 2023 (PMID: 29875488; 34857953; 37794186) have traditionally relied on beta estimate (regression slope) derived from the association analysis to gauge effect size. However, the beta value for the genotype effect is an imperfect proxy for true biological effect size of the pQTL, as the input data for pQTL mapping are heavily normalized to ensure robustness of the regression model results. We therefore calculated allelic fold change applied to protein levels prior to inverse normal transformation which is preferred over beta value for biologically interpretable effect size estimates

(<https://doi.org/10.1038/s43586-022-00188-6>). This approach unveiled substantial genetic effects for certain pQTLs exceeding 30-fold changes, leading to our reference to these findings as "unexpected." Such biologically interpretable effect sizes might have been overlooked in previous studies, underscoring the contribution of our analysis. We believe highlighting these findings in our manuscript is crucial for a deeper understanding of the genetic influences on circulating protein levels. Acknowledging potential confusion, we have refined our language to specify "large genetic effects in terms of allelic fold change..." to ensure clarity.

2. Introduction: Could the authors provide additional comments on current protein markers that could vary during the developmental stages of children? Also, discuss the importance of studying obesity in children.

Thank you for this suggestion. We have provided additional comments on current protein markers that are known to vary during the developmental stages of children. In addition, we have discussed the importance of studying obesity in children, as below:

"During children's development, including puberty, the concentrations of several proteins in the blood are known to vary due to rapid growth and development, hormonal and metabolic fluctuations, immune system maturation, and tissue development and remodeling. Examples include Insulin-like growth factor 1 (IGF-1), leptin, growth hormone, sex hormones, insulin, and C-reactive protein [PMID: 28437784]."

"Global prevalence of pediatric obesity has increased markedly over the past four decades [PMID: 29029897]. Children and adolescents with obesity are more likely to develop a range associated complications including prediabetes, metabolic syndrome, asthma and fatty liver disease. Studying childhood obesity is vital for understanding its health consequences and formulating effective prevention and treatment strategies [PMID: 37202378]."

3. Line 83-84: Please add "based on the current dataset and our modeling" to clarify the context. Would the authors expect a significant change, such as the contribution percentage of BMI, if another sample set were used?

That is an astute point and we have added this as requested.

Obesity is associated with low-grade chronic inflammation, and significantly impacts the plasma proteome. We hypothesized that a cohort enriched in obesity would overestimate the number of plasma proteins associated with BMI-SDS compared to a sample set representative of the general population. To test this, we included an interaction term between the binary obesity group and BMI-SDS to determine if the association with BMI-SDS differed significantly between the two subgroups. This analysis identified 163 BMI-SDS-associated proteins unique to the obesity group and 32 proteins with differing association strengths between the obesity group and the general population. This suggests that a cohort representative of the general population would likely show a lower percentage of BMI-SDS-associated proteins. We have incorporated these results in the revised manuscript (see **Fig. 2f** and **Supplementary Table 2**)."

"To assess the impact of age, sex, and BMI-SDS on the plasma proteome, we performed multiple linear regression analysis (**Methods**). In total, 58% of quantified plasma proteins were associated with at least one of the three factors (about 40% with age, 32% with sex, and 22% with BMI-SDS) and 8% with all three factors based on the current dataset and our modeling (**Fig. 2c**)."

“As per the study design, we included an interaction term between BMI-SDS and binary obesity status. This allowed us to examine if the association with BMI-SDS differed between the two subgroups. Among the 240 proteins associated with BMI-SDS, 163 were unique to the obesity group, and 32 had differing association strengths. This suggests that a cohort representative of the general population would likely show a lower percentage of BMI-associated proteins.”

4. Line 99: A relevant question is which proteins, if any, appeared to be exclusive to obesity.

Please see our response to your **Minor Point 3** just above.

5. Line 107-108 and Figure S3: Forty common proteins were included for predicting age and BMI. Is this expected? It would also be helpful to highlight the best predictive proteins (e.g., top 5 or 10) and their power for further studies.

Our study identified many plasma proteins associated with age and BMI-SDS, leading us to hypothesize that these proteins could in turn predict both age and BMI. It is noteworthy that there are no prior studies on this topic encompassing the specific age range we investigated, which limits direct comparisons. However, existing research focused on adults has revealed that several hundred plasma proteins predicted age [PMID: 31806903; PMID: 33031577]. Considering this, we initially anticipated that a similar number of proteins would be necessary to model age in this study. Remarkably, we found that even with a considerably smaller set of proteins, high predictive accuracy was attainable. This unexpected finding underscores the potency and selectivity of MS-based plasma proteomics in age and BMI prediction in this age range. In our revised manuscript, we have elaborated on this aspect and specifically highlighted the most predictive proteins, as pasted below.

Proteins are in **Supplementary Table 4**. In addition, we added the following text:

“Proteins best predicting age included those regulating the IGF receptor signaling pathway, cartilage and skeletal development, fibroblast growth factor response and cell-cell adhesion. The top five and ten proteins alone predicted age almost as well, with mean absolute error of 1.5-1.6 years vs. 1.2 for all 50.

BMI was predicted by known markers of obesity and proteins linked to obesity, including adiponectin, C-reactive protein (CRP), IGFBP1, IGFBP2, PRG4, SHBG, as well as apolipoproteins (APOA4, APOF) and proteins of the inflammatory response (A2M, APCS, LBP, HSPG2, HP, AOC3, ITGB1, VNN1).”

6. Line 116: We found the statement that "Half of these proteins decrease with age" to be risky and potentially related to data normalization or processing. Please respond or revise accordingly.

We have replaced these results with biological processes that show age-dependent abundance trajectories.

7. Line 130: It would be useful to indicate that the "genetic effects" studied by the authors are solely related to individual SNPs (e.g., not gene copies or genetic interactions).

We have revised the section title to “Effects of SNPs on the plasma proteome”.

8. Line 132: Please add references for how to define a primary pQTL.

We have added references for how to define a primary pQTL. We in addition performed approximate conditional analysis using GCTA-COJO to estimate conditionally independent loci.

“We defined a primary pQTL as the most significant variant in linkage disequilibrium (LD) ($r^2 > 0.2$) within a region (± 1 Mb) that was associated with plasma levels of a protein (**Methods**) [PMID: 30111768; PMID: 32628676].”

In Results: “Approximate conditional analysis revealed 733 conditionally independent pQTLs for 443 proteins (**Methods and Supplementary Table 6**)”

In **Methods**: “We first identified LD-independent signal clusters for pQTLs by LD-based clumping implemented in PLINK (1.90b) [PMID: 7701901] with the following settings: `--clump-r2 0.2 --clump-kb 1000 --clump-p1 5e-8`. We then performed approximate conditional analysis using GCTA-COJO (1.93.3) to identify conditionally independent loci, which we refer as conditionally significant pQTLs [PMID: 22426310].”

9. Line 140: Please add a comment and some references since the finding of non-coding regions seems consistent with previous studies.

“The dominance of non-coding variants among identified pQTLs is consistent with previous studies which reported a percentage ranging between 86% and 98% [PMID: 34239129; PMID: 30111768].”

10. LINE 172: What if the peptide-level data could support a pQTL, whereas the protein level does not? Since DIA-MS essentially measures peptides, it might be necessary to perform a peptide-level analysis and compare the results to the protein level.

We thank the reviewer for the opportunity to elaborate on our approach regarding peptide level vs. protein level to discover pQTLs. This point was also raised by Reviewer #3 in their Major Point 1.

A key advantage of bottom-up MS-based proteomics is that it provides multiple measurements at the tryptic peptide level for a single protein, potentially offering greater granularity on protein quantification. Peptides may map uniquely to a single protein or be shared by multiple proteins, such as proteoforms resulting from genetic mutations, alternative splicing, and PTMs. Therefore, how much a peptide is associated with a protein may vary across biological conditions, especially if the peptide is not unique to that protein. Note, however, that this only happens in a small minority of cases.

If a genetic association was only seen with a peptide but not supported by protein-level data, it would be challenging to determine if it is coincidental, for instance this would be very likely if only one out of ten unique peptides appears to carry a genetic signal. Therefore, we prioritize protein-level quantification for pQTL discovery, using only protein group-specific peptides to minimize interference from non-unique peptides. However, we do further validate pQTLs with peptide-level evidence, assessing the consistency and effect direction of significant peptides supporting a pQTL.

Altogether, we performed GWAS for ~10,000 quality-controlled protein group-specific tryptic peptides, that is, peptides unique to a protein group in the context of the canonical human protein sequence database. After adjusting for multiple tests by a Bonferroni adjusted p value of $5.6e-12$, one would identify 56 additional proteins whose peptides had genetic associations. However, when looking at all peptides associated with these proteins, only 15% met the significance threshold, compared to 66% of peptides belonging to proteins identifying pQTLs using protein-level data. Furthermore, only seven of these proteins

had more than one peptide with a directionally consistent genetic signal. While these seven proteins may well be false negatives, we decided against including them in the final set of reported pQTLs to be conservative and to maintain consistency in our pQTL identification strategy. Nonetheless, we used peptide data in a targeted approach to refine the pQTLs, categorizing them into four categories as follows (now included in **Supplementary Table 5**):

- Tier 1: a variant-protein association with at least two supporting peptides showing directionally concordant associations.
- Tier 2: a variant-protein association with only one supporting peptide, the SNP is non-synonymous and does not reside within the protein-coding gene.
- Tier 3: a variant-protein association with only one supporting peptide, the SNP is non-synonymous and resides within the protein-coding gene. However, the SNP-encoded single amino acid variant should not be part of the supporting peptide.
- Tier 4: a variant-protein association with only one supporting peptide, and the SNP is synonymous.

For Tier2-4, the total number of peptides available for testing must not exceed three, as the SNP-protein associations in these tiers are supported by only one peptide.

With this strategy, 77% of the reported pQTLs have at least two supporting peptides. Please also see the figure just below which will be **Supplementary Fig. 3** in the revised manuscript.

Reviewer Fig. 2, Manuscript Supplementary Fig. 3. Classification of pQTLs based on peptide-level data. a, Classification scheme and criteria. **b-d,** Example of a Tier 1, 2 and 4 pQTLs. **e,** Proportion of Tier1-4 pQTLs.

We added this text to the revised Discussion:

“Our peptide-based framework excluded artefactual pQTLs for four proteins: IL6ST (rs2228043), FLT4 (rs34221241), LYZ (rs1800973) and OSMR (rs34675408). Notably, the association signal between FLT4 and missense mutation rs34221241 was only supported by only one peptide, which flanks the single amino acid variation (N/D) at position 149 due to the missense mutation. Similarly, only the peptide flanking the SNV (T/N) at position 88 supported the association between LYZ and the missense mutation rs1800973.

These findings led us to explore whether our data could help eliminate artefactual pQTLs identified using affinity-based proteomics. This indeed proved to be the case. Three of the above four proteins were also quantified in a recent affinity-based pQTL study [PMID: 37794186], which reported pQTLs (either the exact variant or in LD with $r^2 > 0.8$) for two proteins that our peptide-level evidence deemed artefactual (rs11739016 for IL6ST and rs34221241 for FLT4). Additionally, our data helps resolve disagreements in pQTL direction of effect. Discrepancies between MS-based and affinity-based proteomics have been reported, including for APOE. Specifically, the T-allele of rs7412 was negatively associated with APOE levels in an aptamer-based pQTL study of the Icelandic population (PMID: 35078996). However, a positive association was observed by MS-based proteomics in a pQTL study of the Han Chinese population (PMID: 36797296), which aligns with our data.”

11. Line 200: "of which" - Please revise the text to clarify whether "which" refers to proteins or pQTLs.

Thank you for the opportunity to clarify this. We have revised the sentence to clarify that “which” refers to proteins.

12. Figure 5B: It would be helpful to provide a supplementary table for Figure 5B and check if the most common pQTLs among studies have larger effect sizes.

Inspired by this comment, we compared the effect sizes between replicated and non-replicated pQTLs in our study’s two replication cohorts and have incorporated the results in the revision (**Reviewer Fig. 3a** below, new **Fig. 6c** and **Supplementary Fig. 7c**). We found that pQTLs that replicated in both children and adult cohorts indeed exhibited larger effect sizes. However, for replication results in published studies, there was no clear linear relationship between effect size and the number of studies in which pQTLs were replicated. We did observe that cis-pQTLs were replicated in more studies than trans-pQTLs, suggesting greater evidence and higher agreement for cis-pQTLs among existing pQTL studies.

We have incorporated the insights into the revised manuscript, like so:

“...with cis-pQTLs replicated in more studies than trans-pQTLs, suggesting greater evidence and higher agreement for cis-pQTLs among existing pQTL studies (**Supplementary Fig. 7c**)”

Reviewer Fig. 3. Comparison of effect size between replicated and non-replicated pQTLs. a, Distribution of absolute beta coefficient of replicated and non-replicated pQTLs in our children and adult replication cohorts. **b,** Distribution of absolute beta coefficients of pQTLs as a function of number of existing studies the pQTLs are replicated in. **c,** Number of published studies in which the cis- and trans-pQTLs are replicated.

13. Line 222-242: Are most samples in the primary dataset and replication dataset from Denmark? It might be worth providing a remark about this in the Discussion.

Indeed, all samples in the primary and replication dataset are from Denmark. We have now provided a remark about this in the Discussion, also pasted below.

“By focusing on the Danish population, we benefit from its homogeneity which reduces confounding factors associated with population stratification, genetic admixture, and environmental variation, thereby potentially enhancing the identification and replication of pQTLs. In the future, it would be interesting to extend similar studies to other ethnicities.”

14. Methods Line 707: Were the 17 missing values due to a survey failure or the unwillingness to disclose gender?

These were due to a survey failure not unwillingness to disclose gender, which has been added to the revised manuscript. “17 had missing values due to survey failure”

15. Line 714: We do not understand what "72,792 additional sequences" refers to. Please clarify.

These additional sequences refer to the isoform sequence data as part of the human reference proteome database. In the revised version, we chose to use only protein group specific peptides for protein quantification to reduce interference from peptides that could derive from other proteins. Consequently, we used only the canonical proteome database. Changes have been incorporated in the revised manuscript.

16. Line 760: Were the two covariates (a) sufficient and (b) useful in the model?

We included ‘time to analysis’ as a covariate due to observed variation in plasma proteome driven by platelet marker levels, which correlate with sample storage time (Reviewer Fig. 4a-b below). Unequal distribution of platelet marker levels across the sample collection period (Reviewer Fig. 4c below) suggests a systematic bias related to storage duration, though sampling related technical factors cannot be ruled out. Thus, we added ‘time to analysis’ as a covariate, an approach that was also used in the latest UKB-PPP pQTL study (Sun et al., Nature 2023). We also adjusted for PC1 to eliminate potential confounding. Additionally, we included puberty status as a covariate due to protein abundance changes around puberty onset. This is described in the Methods section and we now include the figure below as well.

We chose to include these most relevant covariates for simplicity. For their potential interactions see response to reviewer 3 below (Reviewer 3, Major Point 6)

Reviewer Fig. 4, Plasma proteomics quality control. **a**, Principal component analysis for the plasma proteome profile of all samples in the discovery cohort, showing the first and second principal component. Samples are color-coded according to time to proteomics analysis (or sample storage time). **b**, Loading plot for the PCA in panel (a) with platelet markers highlighted. **c**, Abundance of TLN1 by year of sampling, with samples being collected in more recent years having higher levels of TLN1.

17. Line 850: The short description of effective size determination can be moved to the Results section as well.

We agree and have moved it to the Results section.

18. Define what a primary pQTL is and what information is included in the "SNP ID." Effect size vs. protein variability.

Please see answer to Minor Point 8 above. SNP ID information has been incorporated in the column header of the supplementary tables. As stated in our previously submitted version, in addition to beta statistics derived from the association test, we determined the effect size of the pQTLs by calculating the ratio of mean protein abundance in heterozygotes (1/0) and homozygotes (1/1) to that of the wild type (0/0) without any data transformation.

19. The authors used the ratio of mean protein abundance between zygote types to calculate the effect size. This might be too simple – How would the author exclude the contribution of the basic protein variability due to the non-genetic effects and technical measurements?

While we use the allelic fold change as a measure of effect size, we also provide the beta coefficient, which takes into account the variability due to those effects. For effect size the allelic fold change is superior in our opinion. As a third measure of effect size, we estimated the contribution of independent pQTLs in plasma protein level variance relative to other measurable factors such as age, sex, BMI-SDS, obesity and the interaction term between obesity and BMI-SDS. Please refer to our response to your Major Point 1 and 3.

Reviewer #3:

Remarks to the Author:

In “Plasma Proteome Variation and its Genetic Determinants in Children and Adolescents”, Niu et al. present the use of mass spectrometry (MS) to determine the levels of circulating proteins and study their associations with genetic variation (pQTLs). This study provides a valuable complementary contribution in a field dominated by affinity-based methods applied to large biobanks. It expands the list of available MS-based pQTL studies in sample size and provides insights into the challenge of assigning peptide-centric pQTLs. The manuscript is well-written, concise and does not present overly controversial findings. The established website is very helpful when browsing the data.

Authors' response: We thank the reviewer for the thorough and positive assessment of our manuscript as well as the highly relevant questions related to mass spectrometry-based proteomics. We have accommodated these suggestions as detailed below.

Major points:

1.1 Especially for a non-proteomics audience, the authors should clarify the use of proteins versus peptides in their analysis. How was the peptide information used to determine protein levels?

Please also see answer to reviewer 2 Minor Point 10 above. In our study, we leveraged both protein and peptide-level data to enhance the precision in identifying plasma pQTLs. Bottom-up proteomics is the most commonly used and most powerful workflow in MS-based proteomics. In bottom-up proteomics, peptide fragments resulting from enzymatic cleavage of proteins are analyzed instead of intact proteins. The identified peptides are then mapped to individual proteins based on a provided protein-sequence database, which are then further aggregated to protein groups in the absence of unique peptides that can distinguish between two proteins. In DIA-MS (data independent acquisition mass spectrometry), peptide quantity is typically calculated based on MS2 level, which records the fragment ion intensity deriving from a specific peptide precursor. Protein quantity is then aggregated from peptide quantities depending on the summarization strategy used. In the revised introduction, we now state:

“In bottom-up MS-based proteomics the signals of peptides are measured and quantified. These peptides are assembled into proteins based on the reference proteome database. Depending on the number of peptides mapping to each protein, this generally yields several data points on the identity and quantitative changes of the proteins in the plasma proteomes of the study participants. These can be used for better protein level estimates or removal of outliers. In contrast, binder-based methods typically probe a single epitope of plasma proteins or two closely spaced one in the case of the Olink proximity extension assay.”

Furthermore, we go into more detail in the Methods:

“For this study, protein quantification is performed at the MS2 level using the Quant 2.0 algorithm within Spectronaut, to minimize cross-interference, our quantification exclusively relied on peptides unique to protein groups.”

This is also a point of central relevance for our treatment of peptide vs. protein level evidence for pQTL identification. The rationale for utilizing protein-level values stems from the methodological advantages of aggregating peptide measurements into a unified protein quantity, which is standard practice in bottom-up proteomics. Benefits include diminishing technical variability, reducing the occurrence of missing values, and lowering the number of hypotheses tested—thereby augmenting statistical power (DOI: 10.1021/acs.jproteome.1c00894 for in-depth technical justification).

Motivated by this and reviewer 2’s insightful feedback, we have refined our analysis to incorporate peptide-level evidence for pQTL verification, categorizing them into four distinct groups (see also answer to Minor Point 10 above).

Changes to the revised manuscript include introduction to protein quantification in bottom-up proteomics for non-MS audience, our rationale of adhering to protein-level quantification for pQTL mapping and the framework for categorizing pQTLs into tiers based on peptide-level data, as below.

1.2 Do all peptides detected per protein provide concordant and interchangeable information?

With the dual-level approach for pQTL identification outlined just above, we observed that 77% of the identified pQTLs were supported by at least two peptides. Notably, within this subset, 94% of the pQTLs demonstrated perfect concordance in directional trends across all supporting peptides, aligning with the protein-level quantitative findings.

1.3 Did the number of peptides detected per protein influence the assignment of pQTLs? Could a better pQTLs coverage derive from a higher analytical coverage of proteins (= more data points per protein)?

Thank you for raising this highly relevant point about “abundance bias”. Indeed, our analysis confirmed that the proportion of proteins being identified with genetic associations increases with increasing number of peptides detected per protein, higher protein abundance, and increased quantification precision (please see figure below). This underscores the inherent challenges in detecting pQTLs for proteins that are present in low abundance. We have incorporated these observations in Results.

“We hypothesized that higher abundance of proteins in the plasma would manifest in increased number and signal of identifying peptides, which would in turn increase the chances of finding genetic associations. Indeed, when grouping proteins into four increasing abundance buckets, we observed a marked increase in the proportion of proteins with genetic associations, and the same trend was observed with increasing technical reproducibility and number of quality-controlled peptides per protein (Fig. 3 j-l)”

In Discussion:

“Given the positive association of abundance with the chance to find genetic associations, we predict that ongoing instrumental developments in MS-based plasma proteomics will lead to increased number of genetic findings even in the same studies.”

Reviewer Fig. 5, New Main Fig. 3 j-l. Technical factors and protein characteristics affecting pQTL identification. a-c, Proportion of proteins with genetic associations when stratifying proteins into buckets based on technical variation (a), median abundance (b), and number of identified peptides after quality control per proteins(c).

2. The authors mention that specificity is a crucial advantage of MS over affinity proteomics. Considering the above, this is effective via the number of peptides detected per protein. The authors should, however, further clarify how genetic variation affects the different peptides detected from a protein. Ideally, corresponding examples for the most dominant annotations (introns, intergenic and regulatory regions) should be presented.

The specificity advantage of MS over affinity proteomics comes from the fact that MS measures the mass of peptides and their fragments down to several parts per million (ppm). This is principally different from affinity proteomics where specificity comes from the three dimensional fit of antibody to target and how unique this is. Furthermore, there are typically several peptides identifying the same protein which adds additional specificity and quantitative accuracy. Importantly, genetic variations normally affect the expression level of the entire protein. Therefore, all peptides identifying the same protein should generally

show the same fold change between genotypes. This shown for examples of introns, intergenic and regulatory regions in the figure below.

We have added this sentence to the introduction:

“The specificity advantage of MS over affinity proteomics comes from the fact that MS measures the mass of peptides and their fragments down to several parts per million (ppm).”

We also mention and show that different genetic variants should affect the expression level of the entire protein and therefore the expression level changes of the proteins should be consistent.

“Genetic variations normally affect the expression level of the entire protein. Therefore, all peptides identifying the same protein should generally show the same fold change between genotypes. Our peptide-level framework for classifying pQTLs showed that 77% of reported pQTLs had at least two supporting peptides (**Fig. 3d and Supplementary Table 5**). Notably, in 94% of these cases, all peptides exhibited the same direction of effect, indicating highly consistent quantitative information at the peptide level (**Supplementary Fig. 4**).”

Reviewer Fig. 6, New Supplementary Fig. 4. Examples of Genetic Variants and Peptide Quantity. a-c, Peptide quantities by genotype illustrating common regulation of peptide quantities for examples of variants located in the (a) intron, (b) intergenic region and (c) regulatory region (c). Up to ten peptides per protein are displayed.

3. In the provided web interface, the top-ranked qQTL is rs9898 for HRG. Interestingly, a recent study presented the challenge when detecting HRG (DOI: 10.1016/j.mcpro.2023.100585) with its non-synonymous substitutions (highlighting rs9898). HRG is an abundant and glycosylated multi-domain protein. As mentioned by the authors, the chosen MS approach detects specific chains or subunits of a protein rather than the protein in its complete shape (and together with its interaction partners). The authors should therefore take his opportunity to showcase these advantages, and maybe HRG can serve as a relevant example.

We thank the reviewer for suggesting this excellent example to showcase the advantage of MS-based proteomics over affinity-based approaches and have incorporated this example in the revised manuscript.

The mentioned study underscores the necessity of considering HRG's polymorphic nature in proteomic studies, especially as these mutations may affect HRG's abundance (DOI: 10.1016/j.mcpro.2023.100585). Previous work has indeed highlighted the limitations of affinity-based proteomics in accurately quantifying protein variants affected by amino acid substitutions in HRG (DOI: 10.26508/lsa.202000817). Specifically, researchers in this later study demonstrated differential selective affinities toward different HRG variants between two antibodies (DOI: 10.26508/lsa.202000817). This phenomenon occurs when the HRG-binding epitope is in close proximity to the variant amino acid residue. This highlights the challenges associated with quantifying abundance differences between protein variants using affinity-based proteomics, particularly in the presence of protein-altering variants. We have added the text below in the Results section just behind the text from the last point and provide a new supplementary figure:

“...highly consistent quantitative information at the peptide level (**Supplementary Fig. 4**). The peptide-level data also helps quantify protein variants affected by amino acid substitutions, a limitation in affinity-based proteomics as highlighted in previous work for the case of rs9898's effect on HRG abundance (DOI: 10.26508/lsa.202000817). Our data included the association between rs9898 and circulating HRG levels, which was successfully replicated in both the children and adult cohorts, with a 62% protein sequence coverage and 26 supporting peptides (**Supplementary Fig. 5**). Importantly, protein quantification was unaffected by the missense mutation (Pro204Ser), which was not detected but would have been an outlier if it had. This example illustrates an important advantage of MS-based proteomics in navigating the complexities of protein quantification across variants.”

Reviewer Fig. 7, New Supplementary Fig. 5. Peptide level evidence for the association between rs9898 and circulating HRG levels. a-b, Increasing protein levels with the number of T alleles at rs9898 in the discovery (a) and children replication (b) cohorts. **c,** Sequence coverage of the HRG protein. **d,** Increasing peptide levels with the number of T alleles at rs9898. Up to ten HRG-derived peptides are displayed.

4. The authors correctly applied a stringent selection scheme for the downstream analysis instead of artificially inflating the reported number of proteins. However, many of the remaining proteins are abundant in the circulation. Hence they will be easier to detect and likely offer more feasible targets to obtain a large

number of unique peptides that support a higher precision. The statement that MS offers better precision for all circulating proteins cannot be answered without a direct head-to-head comparison. Growing evidence shows that low-abundant targets are more difficult to detect reliably due to sensitivity issues, fewer detected peptides, and a higher frequency of missing values. Please discuss and refine the statements considering their effects on the pQTL analysis.

We thank the reviewer for agreeing with our data filtering process. We fully concur with their points regarding the challenge in detecting low-abundance proteins due to sensitivity issues (although this is a moving target due to increasingly powerful technology). This is already reflected in Fig. S1c in our previously submitted version, which shows in general the lower protein abundance the lower quantification reproducibility. Please also refer our response to their Major Point 1, in which we have further investigated how protein abundance and number of peptides detected per protein affect the identification of pQTLs and added the results to the revised manuscript (**New Main figure 3 j-l**).

We agree with the reviewer that whether MS offers better precision for all circulating proteins cannot be answered without a direct head-to-head comparison (please see our response to their Major Point 1).

5.1 For this reviewer, it remained unclear if there is a relationship between age-dependent markers and pQTLs.

According to our data and modeling, 51% of the proteins identified with genetic associations exhibited an age-dependent pattern, with notable examples including IGF1, ITGA2, and RBP4. As described in our high-level executive summary, to gain an integrative understanding of the factors affecting protein abundance during childhood and adolescence, we estimated the proportion of variance explained by various factors studied in this work. This analysis revealed that protein levels in the pediatric population are influenced by age and genetics to varying extents. Please also refer to our response to Reviewer 2, Major Point 1, pasting the relevant content below.

“To understand the relative contribution of variance in plasma protein levels by genetic variation and other biological factors during pediatric development, we performed variance decomposition analysis (**Methods**). This revealed that independent pQTLs accounted for 1% to 66% of variance in protein levels (average 11%) (**Fig. 4a**). Remarkably, for 63% of the proteins, the pQTLs were a more significant contributor than age, sex, BMI-SDS and obesity combined. For some proteins, however, other factors than pQTLs contribute most to the overall variance. For example, SHBG levels are primarily affected by age and obesity, PRG4 levels by obesity and IGF1 and RBP4 level by age. To address the important question of how stable the genetic influences are across pediatric development, we segmented the cohort into 5-10, 10-15, and 15-20 year olds. This revealed a remarkable stability in genetic influences, with Pearson correlations between 0.95 to 0.98 (**Fig. 4b-d**). Our results establish that MS-based proteomics can provide unique insights into factors determining protein level variance throughout pediatric development.

New Manuscript Fig. 4. Variance in plasma protein levels explained by various factors. (a) Proportion of variance explained by conditionally independent pQTLs, age, sex, obesity, and BMI-SDS (summed variance from BMI-SDS and its interaction with obesity status). Proteins are ordered by decreasing variance attributable to independent pQTLs. **(b-d)** Pairwise comparisons of variance explained by independent pQTLs across three age groups: 5-10, 10-15, and 15-20 years, including the Pearson correlation coefficient.

5.2 Given the biology of adolescence, did the authors find any specific associations in younger participants that they could not replicate in the older cohort?

The idea of comparing the effectiveness of pQTL detection between children and adults is indeed fascinating. Our results clearly indicate that the large majority of pQTLs are shared between children and adults. To pick up potential pQTLs that are specific to younger participants, would likely require large datasets of comparable size and the same medical condition or health status, which is not the case in our cohorts. Given the current lack of directly comparable datasets, we have decided against making this comparison to avoid drawing inaccurate conclusions. We acknowledge this limitation in our discussion and emphasize it as an important avenue for future research.

after high replication results “The high replication rate in the adult cohort also indicates that the vast majority of detected pQTLs is not specific to life stage. To detect such potential pQTLs would likely require larger cohorts with similar sizes and health status.”

5.3 Did the authors consider using their pQTLs as instruments in Mendelian randomization (MR) - e.g. for childhood obesity?

We thank the reviewer for this suggestion, prompted by which we performed a systematic two-sample Mendelian Randomization (MR) using top cis-pQTLs from 267 proteins on 47 cardiometabolic GWAS outcomes spanning obesity, diabetes, atherosclerotic cardiovascular disease, metabolic dysfunction-associated steatohepatitis, Alzheimer's disease, and chronic kidney disease therapeutic areas. We have added the results to the revised manuscript (see also our response to Reviewer 2 Major Point 1.2) (**New Supplementary Fig. 8, pasted below**).

“In addition, we performed a systematic two-sample Mendelian Randomization (MR) using top cis-pQTLs from 267 proteins on 47 cardiometabolic GWAS outcomes spanning obesity, diabetes, atherosclerotic cardiovascular disease, metabolic dysfunction-associated steatohepatitis, Alzheimer's disease, and chronic kidney disease therapeutic areas. This analysis reported 345 causal relationships between 106 proteins and 36 traits related to six of the most highly prevalent diseases ($P_{MR} < 2.3e-6$) (**Supplementary Fig. 8** and **Supplementary Table 10**). Of these 345 protein-trait pairs, 101 (29%) between 41 genes and 33 traits were further supported by colocalization evidence, defined as a posterior probability above 70% (**Methods, Fig. 5g-h** and **Supplementary Table 11**). Interestingly, our data causally connect sonic hedgehog (SHH) to height, which may be connected to its crucial roles in embryonic development.”

New Manuscript Supplementary Fig. 8. Mendelian Randomization identifies potential causal genes for cardiometabolic traits and diseases. a, Gene-trait causal relationships ($P_{MR} < 2.3e-6$) for six diseases.

6.1 The authors present a few proteins to validate their observations. However, as shown for the protein IGFALS, could these be due to lifestyle, nutrition, and activity at a young age rather than age itself?

Please conduct interaction analyses between age, sex, and BMI to understand their dependencies in adolescence better.

We agree with the reviewer that we cannot exclude the possibility that the age-associated protein trajectories are influenced by factors such as lifestyle, nutrition and physical activity rather than age itself. However, the current analysis lacks the necessary metadata to directly account for these factors in the regression model. We have conducted a supplementary analysis that includes these interaction terms. The results of this analysis are presented in **Supplementary Table 3**, enabling interested readers to explore these additional dimensions.

“Further exploring the interaction effects between age, sex and BMI-SDS, we found for 24 proteins there was interaction between age and BMI-SDS, 18 between sex and BMI-SDS and 149 between age and sex, including PZP, AGT, SHBG and the above-mentioned age-associated proteins that are related to bone development (**Supplementary Table 3**).”

6.2 The authors present CD5L as a sex-associated protein. However, longstanding evidence exists that this protein is a surrogate of IgM (top hit). Do the authors have additional evidence for ongoing infections that could assist in explaining their observations (e.g., AST, CRP, clinical records)?

We thank the reviewer for this interesting remark. Indeed, in our analysis CD5L and IGHM are both highly associated with sex. This specially makes sense, as a recent study has found that in blood serum, IgM exclusively exists as a complex of J-chain-containing pentamers covalently bound to the protein CD5L (DOI: 10.1101/2023.05.27.542462), supporting our observation that CD5L and IGHM are highly correlated to each other in plasma. Additional analysis of our data revealed that IGHM and J-chain are the two proteins that correlate the most with CD5L in plasma (Pearson correlation coefficient of 0.93 and 0.86, respectively) supporting the redefinition of the composition of circulatory IgM as a J-chain containing pentamer, always in complex with CD5L by Oskam et al from the Heck group.

In the revised text, we have added a sentence to this effect:

“The fact that we found CD5L and IGHM to be the top two hits associated with sex makes particular sense not only because females have higher levels than males (PMID: 7414242) but also in light of a recent report showing that CD5L is an obligate member of IgM in circulation (PMID: 38055740).”

Reviewer Fig. 8. CD5L and IGHM are highly correlated. a-b, Correlation between CD5L and IGHM (a) and JCHAIN (b) with Pearson correlation coefficient indicated.

6. 3 In addition, A2M was found as a sex and BMI-associated protein. Please explain this dual assignment and maybe stratify the discovery cohort in Table 1 by sex (+ add p-values of comparative analysis).

We found A2M to be independently associated with both sex and BMI. We have further stratified the discovery cohort in Table S1 by sex along with the p-values for comparative analysis in Supplementary Note 1. Liver enzymes (ALT, AST and GGT), glucose and insulin levels are significantly different between males and females in both study groups, which is expected given that the pediatric reference levels of ALT, AST, glucose and insulin are known to differ between boys and girls [PMID: 36131081; PMID: 25376219].

7. The clustering analysis is interesting. Please add the clinical traits from Table 1 to this analysis. This might assist in explaining some of the biology these clusters represent. Did the authors see any relationship between these biologies and the frequency of reported pQTLs?

An interesting suggestion, however, when we tried this, we did not observe the clinical variables to fall into any of the clusters of the age dependent protein trajectories. Therefore, we decided to not include them.

8. It would be helpful to dissect the analysis into the organ of origin to add some more physiology to the manuscript. Since many abundant and detected proteins derive from the liver by secretion, it would be very interesting to learn what the other (non-liver) proteins are and which biologies they can represent regarding genetic variance, age, sex, and BMI.

This is the first study describing the top 1,200 proteins identified by MS-based proteomics from neat plasma in an entire cohort. Prompted by the reviewer, we have included two sub-figures illustrating the tissue specificity and biological processes represented by the identified proteins. The Human Protein Atlas (HPA) classifies genes into five categories based on transcriptomics analysis across all major organs and tissue types in the human body: tissue enriched, group enriched, tissue enhanced and not detected [PMID: 24648543]. (*The Human Protein Atlas (HPA) defines “tissue enriched”, “group enriched”, and “tissue enhanced” proteins with at least 500% higher mRNA levels in a specific tissue compared to all other tissues, at least 500% higher mRNA levels in a group of 2–7 tissues compared to the rest, and at least 500% higher mRNA levels in the tissue compared to average levels in all tissues, respectively.*)

According to this classification, 10%, 18% and 37%, and 35% of the proteins are “tissue enriched”, “group enriched”, “tissue enhanced” and “low tissue specificity”. Liver has the most proteins assigned, followed by lymphoid tissue and intestine, comprising proteins involved in acute inflammatory response and immune response. Interestingly, the brain ranks fourth featuring proteins involved in nervous system development, such as neuronal cell adhesion molecules (NCAM1, NCAM2, NRCAM), neuronal growth regulator (NEGR1), and GABA transporter SLC6A1.

Above, we discussed the physiological roles of age-associated proteins in response to Reviewer #1, Major Point 1.2; highlighted brain-specific proteins and their pQTLs (Reviewer 1, Major Point 1.2) and emphasized non-liver proteins associated with obesity, as outlined below. Moreover, we added tissue specificity annotation to proteins associated with age, sex, BMI-SDS and pQTLs in **Supplementary Table 2 and 5**.

New text in results:

“As this is the first MS-based proteomics study on neat plasma with substantial depth (> 1000 proteins), we investigated the tissue origin as classified by tissue specific transcriptomics (ref. Protein Atlas). According to this classification, 10%, 18% and 37%, and 35% of the proteins are “tissue enriched”, “group enriched”, “tissue enhanced” and low tissue specificity. Liver has the most proteins assigned, followed by lymphoid tissue and intestine, comprising proteins involved in acute inflammatory response and immune response (Fig. 2a). Interestingly, the brain ranks fourth featuring proteins involved in nervous system development, such as neuronal cell adhesion molecules (NCM1, NCAM2, NRCAM), neuronal growth regulator (NEGR1), and GABA transporter SLC6A1. Reflecting the important roles of plasma proteins in immune response, blood clotting, and transportation, complement, coagulation cascades, metabolism and inflammatory response are among the biological processes with the most proteins assigned (Fig. 2b).”

Reviewer Fig. 9, new Manuscript Fig. 2 a-b. Tissue and functional characterization of all identified proteins. **a**, Characterization of tissue specificity according to the Human Protein Atlas (HPA) classification for all identified proteins. **b**, Biological processes represented by all identified proteins.

The following text has been added to discussion:

“The liver dominated the tissue-specific proteins, but we also identified several non-liver proteins linked with obesity. These include the well-known adipokine ADIPOQ, which is elevated in obesity, as well as PRG4 and ITLN1 from adipose tissue, which show increased and decreased levels in obesity, respectively. Interestingly, uromodulin (UMOD), a kidney-specific and the most abundant protein in normal urine, is reduced in obese children. The physiological role of serum uromodulin remains elusive, but it is inversely associated with metabolic syndrome components and type 2 diabetes in the elderly population [PMID: 31505464; PMID: 30892596]. Our MR results further showed that UMOD is causally linked to hip circumference and diastolic blood pressure. Serum UMOD may have anti-inflammatory effects [PMID: 18495803]. Given systemic inflammation in obesity, perhaps lower UMOD levels contribute to increased inflammation in this condition. Additionally, other non-liver specific proteins, including brain-specific OLFM1, heart muscle-specific NEBL, and proteins enriched in the intestine/pancreas (ANPEP) and muscle (FHL3), also decreased in obesity.”

9. The study is likely underpowered to determine many trans-pQTLs. However, considering the observed trans-pQTLs, did the authors find any “usual suspects”, such as the ABO locus, or did they identify other trans-pQTLs? Could these be MS-specific and linked to enzymatic cleavage (see KLKB1)? Please consider this recent preprint during the revision (DOI: 10.1101/2023.04.20.537640v1). Lastly, the authors should extract some biology from the novel, top-ranked, or most interconnected cis- and trans-pQTLs.

Our previous dataset was indeed arguably underpowered in terms of sample size and proteome depth to identify subtle pleiotropic genetic effects on plasma protein levels. Our new dataset almost triples the proteome depth and this turned out to be crucial to find pleiotropic effects. We now added the following to the manuscript:

“We replicated the known pleiotropy of the ABO locus for 13 different proteins and the APOE locus for 10 different proteins. The proteins associated with variants in the ABO locus play important roles in maintaining homeostasis. These include von Willebrand factor (VWF) which is essential for platelet adhesion and blood coagulation, angiotensin-converting enzyme (ACE) that regulates blood pressure and several proteins that facilitate cell-cell adhesion, such as PTPRM, ICAM2, CDH1, CDH5, and CDH13. Additionally, proteins like MBL2 and MASP1 are involved in the lectin pathway of complement activation. Among these are novel pQTLs in the ABO locus for PTPRM, CDH13 and MASP1. In addition, we identified a novel pleiotropic locus regulating multiple proteins involved in the inflammatory processes and immune response. Specifically, the variant rs56294298 in AKNA is associated with ORM1, ORM2, S100A8 and S100A and TRAV17, of which all but ORM1 are novel associations.”

Please refer to Reviewer 2 Major Point 1.2 regarding highlighting biological insights from the novel pQTLs.

Other comments:

1. Line 242 states “case-by-case basis”. Does this imply that there is no systematic approach to solving such cases?

By “case-by-case basis” we meant for different diseases. We have rephrased this sentence to clarify.

2. Please provide QQ-plots for all association analyses as supplementary information, and include all variables from the discovery cohort (e.g., ALT, AST, insulin, cholesterol, ...)

We have provided QQ-plots for all association analyses as supplementary information (**Supplementary note 2**).

3. The online tool is very helpful. Please explain the meaning of the “lower” volcano in the BMI plot. Please reorder the pQTL table to move the p-value and MAF columns further to the left for ranking simplicity. It would be helpful if this pQTL table could be searchable for features of interest.

The ‘upper’ and ‘lower’ volcano plot reflects differential association strength with BMI-SDS between the obesity group and the general population for some proteins. We have revised the figure legend to indicate the results from BMI-SDS and its interaction with obesity (manuscript **Fig. 2f**). We have made changes on the webpage and will release it upon publication.

4. Sex is listed among the non-genetic effects. Please clarify, and explain the rationale behind removing the sex-specific chromosomes from the analysis.

We agree and have revised the section titles to “Effect of SNPs on plasma protein levels” and “Effect of demographic and health-related factors on plasma protein levels”. Although, sex chromosome exclusion has been commonly done in GWAS [PMID: 37267899], we acknowledge that there are specialized methods available to address the unique challenges associated with sex chromosome analysis. However, our study

focused on autosomal chromosomes. We aim to incorporate specialized methods to analyze sex chromosomes in subsequent projects, building on the foundation established in the current study.

Saturday, August 31, 2024

Point-by-point answers to reviewers' comments

We thank the reviewers for their very positive feedback on our manuscript “**Plasma Proteome Variation and its Genetic Determinants in Children and Adolescents**”. To guide the revision, the text in red represents our response to the comment, while the text in blue indicates the changes we have made in the manuscript.

Reviewers' Comments:

Reviewer #2:

Remarks to the Author:

In the revised manuscript, the authors provided a complete remeasurement of the entire cohort and additionally included substantial replication samples using the latest mass spectrometry platform, despite previous positive reviewer comments. This effort perhaps explains why the Mann lab has been one of the major leading labs driving mass spectrometry application for many years and should be highly applauded. The revised study is at the forefront, comprehensive, and inspiring in many aspects. Its significance includes: a) the best large-scale plasma proteomic dataset of its kind so far; b) the first application to a large number of children; c) a comprehensive resource of pQTLs (both protein and peptide level, which is not possible via affinity-based approaches). It should definitely be published in Nature Genetics in our opinion. Before it is accepted, however, due to the new dataset and profound new analysis presented in the revision, we have the following suggestions and questions.

Authors' response: We thank the reviewer for their valuable feedback which improved our manuscript.

Major:

1. The tissue origin and specificity of the plasma proteins.

Given that >1000 proteins were identified in the neat plasma, the authors sought to perform some annotation to understand the tissue derivation of these proteins. They only annotated to an mRNA dataset, not a proteomic dataset, using ambiguous descriptive items such as “tissue enhanced” or “group enriched” etc. We don't think this analysis (Figure 2a & Figure 3i) is necessary or precise. Notably, this is a very challenging scientific topic that requires additional analysis of tissue proteomes (see Malmström et al., 2016). Also, although >1000 proteins were analyzed, this is certainly not the largest proteomic dataset in plasma so far. The fact that the brain was ranked 4th among all tissues is very suspicious due to the existence of the BBB and is risky to publish. This reviewer therefore strongly suggests that the authors remove this analysis, as it is not very relevant to the study's topic, either. Otherwise, they should perform this analysis more properly, addressing all the above points, and adding relevant discussion.

We thank the reviewer for highlighting this important point. We initially included the tissue specificity analysis to provide context for the plasma proteins we identified. However, we agree that tissue specificity classification for plasma proteins is complex and potentially misleading. In response to the reviewer's suggestion, we have removed Figures 2a and 3i. While we recognize the limitations of the Human Protein Atlas (HPA) system for plasma proteins, it remains a widely accepted resource used in

recent literature, including a study on organ aging signatures in the plasma proteome published in *Nature* (PMID: 38057571). Therefore, we propose retaining the HPA tissue specificity annotations in the supplementary tables for readers who may find this information valuable.

2. Figure 4 b-d and pQTLs across age groups.

We are still struggling to understand how aging should impact calling pQTLs (re. Figure 4b-d). Suppose there are contributions from environmental factors, the aging process, and emerging diseases, the chance of detecting genetic control via pQTLs should be affected by aging to some extent? Would it be possible to compare the absolute portion % rather than the correlation? Relating to this, some analysis and discussion around aging and pQTLs or even eQTLs would be beneficial since this study focuses on children, as compared to previous studies.

This is a highly interesting project in its own right. To incorporate the reviewer's comments, we have added the following discussion. To address the impact of aging and related factors on pQTL detection, one approach would be to collect longitudinal data and compare molecular-QTL detection over time within the same population.

Our study focusses on children and adolescents. One may expect a decline in the proportion of genetic variants influencing molecule levels in later life as environmental factors and age-related processes become more prominent and challenging to adjust for. This aligns with Medawar's mutation accumulation theory which suggests that the genetic regulation system is more robust in early life due to strong selective pressures.

For example, a study on the impact of aging on eQTL (expression quantitative trait locus) detection found that the number of genes with detectable eQTLs in blood decreased by 4.7% when individuals aged 70 were resampled at 80 years old [PMID: 31684996]. Another study demonstrated that the impact of aging on the predictive power of eQTLs varies across different tissues [PMID: 36192477].

Regarding the reviewer's suggestion to compare absolute portion percentages, we found that the proportions of variance explained by pQTLs remained relatively stable across the age groups in our study (5-10, 10-15, and 15-20 years). This stability is likely due to the narrow age range of our cohort, which focuses on children and adolescents.

These findings highlight the importance of our study focusing on children and adolescents, as it provides insights into genetic regulation during a critical developmental period.

Minor:

1. Line 152: Mention how data is normalized in the result texts (plasma volume or total MS signals).

We have added the following to the results section.

“We performed sample preparation and LC-MS analysis using equal plasma volumes, followed by MS signal normalization at the precursor level across all experimental samples using the built-in method in Spectronaut (cross-run normalization) [PMID: 25724911].”

2. Supplementary Figure 2b-q: Add more legend text to explain the variables and numbers of boys and girls and yes and no.

We have added the following text to the figure legend of Supplementary Figure 2b-q.

“Obesity status is classified based on BMI-SDS, with 'yes' indicating a BMI-SDS ≥ 1.28 (above the 90th percentile) according to Danish reference values, and 'no' otherwise. The number of participants in each trajectory group is as follows: obese_boy (538), obese_girl (641), non-obese_boy (420), and non-obese_girl (529).”

3. Line 153: What controversies? How are they resolved?

We have rephrased and further expanded the sentence by adding the following paragraph.

“The relationship between ANGPTL3 and obesity or BMI has yielded mixed results [PMID: 34440242]. Some studies report elevated circulating levels of ANGPTL3 in patients with obesity or a positive correlation with BMI [PMID: 34422408; PMID: 33273510], while others show this is not the case in individuals with metabolically healthy obesity [PMID: 34440242] or even report the opposite trend [PMID: 29695708]. The reasons for these discrepancies remain unclear, though differences in underlying metabolic conditions have been suggested as potential explanations. Our data add new observations in children and adolescents, an underrepresented population, contributing to the ongoing exploration of the complex relationship between ANGPTL3 and BMI.”

4. Line 268: Explain “a held out set of 639 individuals.”

We have added the following text to Line 268.

“... a held out set of 639 individuals that were not exposed to model training and used to evaluate prediction performance of the model.”

5. Line 337: Explain “($5 \times 10^{-8}/1,216$ proteins),” since the same p-value was used to infer 443 proteins in the same paragraph.

We reported pQTLs at both the genome-wide significance level (5×10^{-8}) and the study-wide significance level ($5 \times 10^{-8}/1,216$), with the latter adjusting for the number of proteins tested in the genome-wide associations. At genome-wide significance level, we identified genetic associations for 443 proteins. We have clarified this in the text.

“Applying a study-wide significance level of $p < 4.1 \times 10^{-11}$ adjusting for the number of proteins tested in the genome-wide associations ($5 \times 10^{-8}/1,216$ proteins) resulted in 1,252 primary pQTLs for 327 proteins (Supplementary Table 5).”

6. Line 358: The argument of Pro204Ser is weak unless it is really detected. Right now, is it not detected because of database searching?

We appreciate the reviewer’s question. This case actually highlights an advantage of our MS-based approach rather than a limitation. Protein quantification is not affected by mismatches in the protein sequence database. In all cases (homozygous for Pro or Ser, and heterozygous), protein quantification can be accurately inferred from other peptides covering 62% of the protein sequence. The tryptic peptide containing the Pro204Ser variant is likely not detected because it would only be four amino acids long

(NCPR, see sequence below). Such a short peptide is difficult to identify by mass spectrometry as it cannot be uniquely assigned to a protein.

HRG_HUMAN|P04196 Sequence 190-210

GGEGTGYFVDFSVR|NCPR|HHFPR|

We have added the following to the manuscript text right after the sentence “Importantly, protein quantification was unaffected by the missense mutation (Pro204Ser), which was not ~~detected~~ identified but would have been an outlier if it had.”

“The peptide containing the Pro204Ser variant was not identified, likely because it would only produce a four-amino-acid sequence (NCPR).”

7. Line 362: 62% of “discovered pQTLs”? Since a paragraph was inserted above now.

Indeed, we have added “discovered pQTLs” in that sentence.

8. Figure 3i: Again, it makes no sense to conclude that pQTLs largely mirror tissue origination or “enhanced”.

We have removed Figure 3i and related statement from the revised manuscript.

9. Figure 4a: A supplementary table of Figure 4a would be nice, as the font is too small for the x-axis.

Thank you for this excellent suggestion. We have now added a supplementary table (Supplementary Table 7 in the revised manuscript) for Figure 4a.

10. Supplementary Figure 7a: Highlight the authors’ study here.

We have highlighted our study in Figure 7a (also pasted just below).

Supplementary Fig. 7. Comparison of pQTLs to previous plasma or serum studies. a, Comparison of the number of proteins analyzed and the number of samples analyzed by 35 previous studies. Studies were color-coded based on the proteomics platforms used to generate the proteomics data. **b,** The number of pQTLs replicated in previous studies. **c,** Number of published studies in which the cis- and trans-pQTLs are replicated.

11. Line 856-861: The peptide analysis for excluding artefactual pQTLs seems to be an important note to the field and the authors even mentioned that in the abstract. We, therefore, think this result and “Reviewer Fig 2” should be included as a supplementary figure (at least) and presented as part of the Results, unless there is a reason not to.

Thank you for recognizing the importance of this analysis. Reviewer Fig 2 was already included as a supplementary figure (Manuscript Supplementary Fig. 3) in the first revision.

12. Line 92: We suggest adding “due to the limitation of MS sensitivity and other factors” after the authors’ comment on “low numbers of proteins” to give more credit to previous studies. Obviously, after 10 years, “>1000 plasma proteins by MS” will not be as impressive as it is now, and Astral will be outdated.

Thank you for the suggestion. We have added the sentence to the revised manuscript.

Reviewer #3:

Remarks to the Author:

Niu et al. have revised their manuscript: “Plasma Proteome Variation and its Genetic Determinants in Children and Adolescents Plasma Proteome Variation and its Genetic Determinants in Children and Adolescents” and responded to all my questions and concerns. The study complements the ongoing efforts to map pQTLs of circulating proteins.

Authors’ response: We thank the reviewer for their valuable feedback which improved our manuscript.

Beyond my previous comments, the authors used a novel MS instrument to generate new data reporting 1200 instead of 400 proteins (in all samples). Moreover, they added a new cohort of adults compared to the ones described in the first submission. Adding new data with the newest instruments is, of course, very good. However, please take this opportunity to compare data from overlapping proteins between the previous and new data. This helps the readers understand if other instruments will deliver concordant data for the overlapping targets. It may also offer technical validation and understanding if a later (delayed) analysis provides the same protein levels.

Thank you for this insightful suggestion, the results of which will interest many readers. We now performed a comprehensive analysis to compare data from overlapping proteins between the previous and new measurements, which we have now added as Supplementary Note3. Despite changes in our methodology and instrumentation, we found strong consistency in our results:

1. We successfully replicated 82% of the pQTLs for the overlapping proteins (77%) in the new dataset at genome-wide significance ($5e-8$), despite modifications in our search and quantification strategy.

- Using a subset of 96 samples measured three months apart on old and new instrumentation, we observed a high median Pearson correlation coefficient of 0.89 among 408 overlapping proteins (**Supplementary Note 3 Fig. 1a**).
- To assess the impact of longer-term storage, we analyzed another set of 96 samples measured 2 years apart. Despite this extended delay, we still found a robust median Pearson correlation of 0.79 among 465 overlapping proteins (**Supplementary Note 3 Fig. 1b**).

These results demonstrate that our methodology produces highly consistent protein quantification across different time points and instrumentations. This consistency supports the robustness of our findings and suggests that our approach can reliably detect biological variation even with technological advancements or delays in sample analysis.

In addition, we have added the following paragraph in Results:

“Comparison with our previous dataset, generated using an older generation of mass spectrometer [<https://doi.org/10.1101/2023.03.31.23287853>], reveals that our methodology produces highly consistent protein quantification across different time points and instrumentations as detailed in **Supplementary Note 3**. This consistency supports the robustness of our findings and suggests that our approach can reliably detect biological variation even with technological advancements or delays in sample analysis.”

Supplementary Note 3 Fig. 1. Quantification consistency between old and new instrumentation and the effect of delayed measurements. **a**, Distribution of Pearson correlation coefficients for protein levels measured by the old and new instruments with a 2-month gap, showing results for 408 overlapping proteins. **b**, Distribution of Pearson correlation coefficients for protein levels measured by the old and new instruments with a 2-year gap, showing results for 465 overlapping proteins.

One observation was decreased p-values for BMI-SDS analysis (Fig 2f). Is this due to measuring other proteins?

This is due to the increased number of proteins in hypothesis testing and the modified model specification introduced in the first-round revision, where we added a binary term for obesity status and an interaction term between obesity and BMI-SDS.

Since the time of submission, other related work has been published. To provide a complete picture,

please discuss and reference the two articles:

<https://doi.org/10.1038/s41467-024-45233-y> as a recent example of MS-derived pQTLs, covering some aspects discussed here, including the novel pQTLs.

2) <https://www.biorxiv.org/content/10.1101/2024.05.27.596028v1> as a recent example that directly compares MS and affinity-based pQTLs.^[SEP]

We have added discussion related to the two articles as follows: Blue font indicates new insertions.

“The frequency of artefactual pQTLs may increase when using both protein and peptide-level data for discovery without verification compared to using protein-level data alone, especially when more protein-altering variants are typed or imputed. A prior pQTL study using exome sequencing and MS found that nearly half of independent associations based on combined protein and peptide-level data were technical artifacts [PMID: 30562114]. Our approach addresses this issue by using protein-level data for discovery and peptide-level data to eliminate artifacts, categorizing pQTLs into tiers of confidence. Alternatively, artefactual pQTLs can be minimized by modifying the protein sequence database prior to database searches, as shown in a recent study [PMID: 38307861]. This database approach involves generating and comparing results across multiple protein sequence databases. However, this approach may be limited to protein altering variants (PAVs) as it may miss artefactual pQTLs from SNPs in linkage disequilibrium with a PAV that isn't directly typed or imputed, and thus cannot be excluded from the sequence database. Our method complements these approaches by providing a comprehensive framework for pQTL validation.”

“A recent study comparing MS- and affinity-based proteomics found that up to one third of the affinity proteomics pQTLs may be affected by epitope effects [bioRxiv: doi: <https://doi.org/10.1101/2024.05.27.596028>]. It would now be interesting to study how many of the affinity-proteomics pQTLs can be validated by MS-based proteomics, accounting for various factors such as cohort characteristics, computational methods, sample size, proteome coverage, and variants analyzed. Our study, focusing on children and adolescents, provides a unique dataset for such comparisons and highlights the potential of MS-based approaches in validating and extending pQTL findings.”

Reference # 11 is the preprint of # 61

We have replaced reference #11 with #61 in the revised manuscript.